# Phase-separated NDF−FACT condensates facilitate transcription elongation on chromatin

Ziwei Li[1,18], Francesca Burgos-Bravo[2,3,4,5,18], Kevin Xu[1], Chen Li[3,5,6], Kelvin Y. Kwan[7], Alexander B. Tong [3,4,5,8], Zelin Shan[9], Huan Wang [10], Motoki Takaku[11], Joey Li[1], Zheng Shi [10,12], Dmitry Lyumkis[9,13,14], Carlos Bustamante [2,3,4,5,6,8,15,16,17] ✉ & Jia Fei [1,12] ✉

How the facilitates chromatin transcription (FACT) complex enables RNA polymerase II to overcome chromatin barriers in cells remains poorly understood−especially given the limited direct interactions of FACT with polymerases, DNA or nucleosomes. Here we demonstrate that phase separation, mediated by nucleosome destabilizing factor (NDF), is a key mechanism enabling the function of FACT during transcription elongation. Through biochemical and single-molecule assays, we found that NDF−FACT condensates create specialized biochemical environments that enhance transcription efficiency approximately 20-fold compared with FACT alone. These dynamic condensates form on transcribing RNA polymerase II and travel along chromatin, where they promote efficient nucleosome disassembly at barriers while retaining histones on DNA to preserve chromatin integrity. In human stem cells, disruption of these condensates leads to genome-wide transcriptional defects and chromatin instability, mirroring the effects of FACT depletion. By showing that phase separation enhances FACT function during transcription elongation, our study reveals a key mechanism that preserves chromatin integrity and transcriptional homeostasis in human stem cells.

Eukaryotic gene expression is tightly regulated, with nucleosomes serving as barriers that both prevent sequence-specific transcription factors from accessing regulatory regions and impede RNA polymerase II (Pol II) elongation through the transcribed genes, thereby controlling gene expression at multiple levels[1–4]. To overcome these chromatin barriers, cells have evolved various factors that assist Pol II in transcribing through chromatin[5,6].

Human nucleosome destabilizing factor (NDF) (also known as Glyr1, N-PAC or NP60) is one such elongation factor that facilitates Pol II progression through chromatin. NDF contains a conserved N-terminal PWWP domain that binds H3K36me3-modified histones[7,8], and a C-terminal dehydrogenase-like domain, which does not exhibit dehydrogenase activity[9]. Our previous studies showed that purified NDF destabilizes nucleosomes in an ATP-independent manner[10]. In addition, NDF directly binds Pol II and stimulates its transcription elongation[11,12]. Despite these findings, the precise cellular mechanisms by which NDF functions, particularly with other cellular factors, remain largely unexplored.

Another critical factor involved in overcoming nucleosome barriers is the human facilitates chromatin transcription (FACT) complex, composed of the SUPT16H (hereafter referred to as Spt16) and SSRP1 subunits[13]. FACT performs two key biochemical activities: disassembling nucleosomes to lower the energy barrier for Pol II and retaining histones to maintain chromatin integrity after Pol II passage[14–16].

---

Beyond transcription, FACT is also crucial for other chromatin-related processes, including replication and DNA repair (for reviews, please see refs. 17–19). Notably, targeting FACT in cancer cells using small molecules such as curaxin has shown significant efficacy in preclinical cancer models and is currently in clinical trials[20].

Despite their physiological importance, the exact mechanisms by which NDF and FACT operate in cells remain unclear[18,19]. This is particularly puzzling for FACT, as purified FACT shows limited interaction with DNA, RNA Pol II or intact nucleosomes[21–24]. Recent structural studies reveal that FACT bound to partially disassembled nucleosomes via regions on H2A/H2B dimers normally occluded by DNA[22,23,25], providing insights into histone retention during nucleosome disassembly. However, questions remain about FACT's initial chromatin recruitment, nucleosome barrier facilitation and how its substantial intrinsically disordered regions (IDRs, ~50% of the complex) contribute to its function.

In this study, we found that NDF-mediated phase separation is a key mechanism for FACT cellular functions. Phase separation enables proteins to form dynamic non-membrane-bound compartments that organize specialized biochemical reactions[26]. We found that NDF–FACT condensates form on transcribing Pol II, move along chromatin and enhance nucleosome disassembly and Pol II progression. In human stem cells, disrupting these condensates substantially reduces FACT chromatin occupancy, leading to defects similar to FACT depletion. These findings reveal that phase separation is critical for FACT function and gene regulation.

## Results

### NDF and FACT synergistically enhance Pol II transcription

To investigate the cellular function of NDF, we generated a HeLa cell line stably expressing GFP- and Flag-tagged human NDF and performed tandem affinity purification coupled with mass spectrometry. This approach identified several transcription-related proteins, notably the human FACT complex, as major NDF-associated partners (Fig. 1a). The interaction with FACT was particularly interesting given their overlapping functions in facilitating Pol II transcription through chromatin[10,13]. Co-immunoprecipitation confirmed the NDF–FACT interaction in cells (Fig. 1b). In vitro pull-down assays with purified recombinant proteins demonstrated direct interactions (Extended Data Fig. 1a,b), with additional experiments revealing that the interaction was primarily mediated by the Spt16 subunit (Fig. 1c).

We next sought to determine the functional significance of their physical interactions. Given their overlapping roles in Pol II elongation, we hypothesized that NDF and FACT might synergistically facilitate Pol II transcription on chromatin. We conducted in vitro transcription elongation assays using purified *Saccharomyces cerevisiae* Pol II and a positioned 601 nucleosome downstream of the transcription elongation start site (Fig. 1d). In this assay, Pol II transcribes/extends a fluorescently labelled RNA primer until encountering nucleosome barriers. Consistent with previous reports[27], Pol II exhibited major pauses at superhelical locations (SHL) −5/−4 and SHL −1 without additional factors. NDF specifically alleviated pausing near SHL −5/−4, while FACT reduced pausing at SHL −1. Remarkably, when both factors were present, all pauses before SHL −1 were eliminated, resulting in markedly increased full-length transcription product (Fig. 1e). Michaelis–Menten analysis showed that NDF and FACT together enhanced the transcription rate of Pol II more than either factor alone, with a fourfold decrease in Michaelis constant ($K_m$) and threefold increase in maximum velocity ($V_{max}$) compared with FACT alone (Fig. 1f). Monte Carlo simulations revealed that the combination of NDF and FACT increased transcription completion efficiency ($K_{cat}/K_m$) by 22.50-fold compared with FACT alone and 19.00-fold compared with NDF alone. This synergistic effect was not dependent on the order of factor addition, as mixing both factors before the reaction yielded similar results (Extended Data Fig. 1c). Together, these data support our hypothesis that NDF and FACT employ distinct mechanisms that synergistically facilitate Pol II progression through nucleosome barriers.

### NDF and FACT form phase-separated condensates

To investigate the mechanism behind the synergy, we attempted to solve the cryo-electron microscopy structure of the NDF–FACT complex but observed large molecular weight 'aggregates' during purifications, preventing structural analysis. Noting that both NDF and FACT contain substantial IDRs (Extended Data Fig. 2a), we hypothesized that they might form multivalent interactions leading to phase-separated condensates[26].

Indeed, while purified NDF–GFP or FACT–mCherry alone did not form condensates (Fig. 2a), mixing them under physiological salt conditions immediately produced micrometre-sized spherical droplets. Titration experiments showed that visible condensates could form at equimolar protein concentration as low as 0.125 µM without molecular crowding agents (Fig. 2b). Phase separation occurred in various buffers, except those with high ionic strength and detergent (Extended Data Fig. 2b). Notably, addition of Pol II elongation complex promoted condensates formation at even lower concentrations (Extended Data Fig. 2c), consistent with surface-enhanced condensation[28]. Control experiments showed that Spt4/5 did not form condensates with FACT (Extended Data Fig. 2d). We also confirmed condensates formation using sedimentation assay, where mixed NDF and FACT proteins condensed together while individual proteins remained in the supernatant (Fig. 2c).

Given that both FACT and NDF are highly conserved across eukaryotes, we tested whether condensate formation is also conserved by examining yeast homologues. Recombinant *S. cerevisiae* FACT (Spt16 + Pob3) and yeast NDF homologue (Pdp3) did not form condensates individually but readily assembled into droplets when mixed together (Extended Data Fig. 2e,f), mirroring our human protein results. This demonstrates that NDF–FACT condensate formation is evolutionarily conserved, supporting its biological importance.

Next, we investigated the biophysical properties of NDF–FACT condensates in vitro. Condensate formation was sensitive to ionic conditions, with higher salt concentrations inhibiting assembly (Extended Data Fig. 2g). Importantly, condensate formation was reversible, as pre-formed condensates could be disrupted by 0.4 M NaCl treatment (Fig. 2d). The condensates also showed sensitivity to other disruptive treatments, including partial disruption (~20%) by 1,6-hexanediol (1,6-Hex) (Extended Data Fig. 2h), a compound that disrupts weak hydrophobic interactions[29], and size reduction at elevated temperatures (Extended Data Fig. 2i). To explore the dynamics of NDF–FACT condensates, we performed a fluorescence recovery after photobleaching (FRAP) experiment. After photobleaching, only ~20–30% of the fluorescent signals recovered after 200 s (Fig. 2e,f). Varying the protein concentrations or their ratio affected the dynamics, though most showed low to medium recovery rates (Extended Data Fig. 2j). Additionally, NDF–FACT droplets did not fuse to form larger condensates within 10 min of incubation (Fig. 2g). Although these condensates exhibit lower material exchange rates than liquid droplets, their salt reversibility distinguishes them from solid protein aggregates, which are resistant to high salt treatment[30,31], positioning NDF–FACT condensates as gel-like assemblies.

Despite this limited material exchange, the gel-like condensates retained biochemical activity. First, pre-formed NDF–FACT condensates could actively recruit Cy2-labelled mononucleosomes within 10 s, distributing them evenly inside the condensates (Fig. 2h and Extended Data Fig. 2k–m). This rapid recruitment is particularly interesting as FACT alone shows limited interaction with intact nucleosomes in vitro[21], suggesting that phase separation creates a specialized environment that enhances interactions with nucleosomal substrates. Second, we analysed RNA transcripts associated with condensates during transcription by centrifuge-based sedimentation.

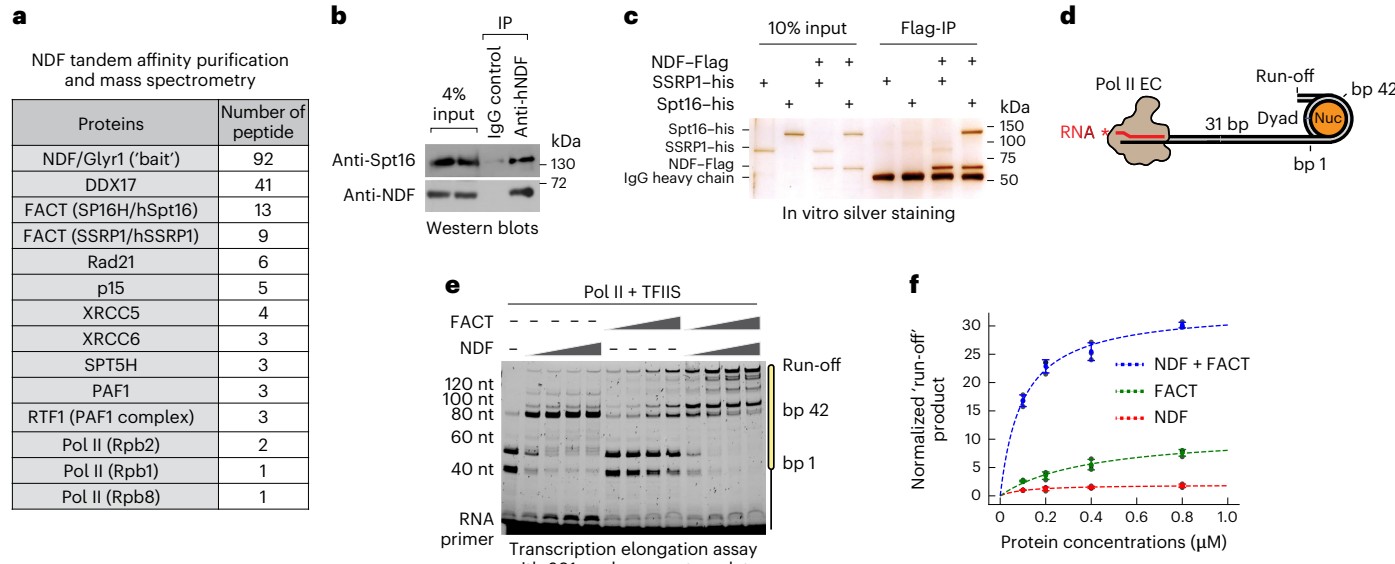

**Fig. 1 | NDF interacts with FACT and synergistically stimulates transcription.** **a**, Proteins identified by mass spectrometry following tandem affinity purification of NDF−Flag−GFP from HeLa cells, listed from three independent experiments. **b**, Co-immunoprecipitation (IP) of the endogenous NDF interaction with Spt16 in HeLa cells. **c**, Interaction between recombinant NDF−Flag and the FACT complex analysed by a FLAG pull-down assay. In **b** and **c**, representative results from three independent experiments are shown. **d**, A schematic of Pol II−nucleosome complexes for in vitro transcription elongation assays. The major pausing sites at nucleosome SHL −5 and SHL −1 are labelled as bp1 and bp42, respectively. **e**, A nucleosome transcription assay shows an increase in the full-length (run-off) product in the presence of 0.1 μM, 0.2 μM, 0.4 μM or 0.8 μM NDF or FACT and shows synergy in the presence of equimolar

amounts of both NDF and FACT. Transcription reactions were performed with 1.0 mM NTPs for 10 min at room temperature. RNA length and corresponding nucleosomal base pairs are indicated. **f**, Quantification of run-off products from the experiment shown in **e**, based on three biological replicates, normalized to run-off product from 0.1 μM NDF alone. Mean values and s.d. are displayed, with black dots representing individual data points. The reaction rates were calculated using Michaelis−Menten analysis: for NDF alone, $V_{max} = 1.97$, $K_m = 0.11$, goodness of fit ($R^2$) = 0.67; for FACT alone, $V_{max} = 11.22$, $K_m = 0.40$, $R^2 = 0.92$; and for equimolar NDF and FACT together, $V_{max} = 33.09$, $K_m = 0.10$, $R^2 = 0.94$. A $P$ value of $1.02 \times 10^{-9}$ was obtained from the one-way analysis of variance (ANOVA) test for the 0.8 μM concentration reactions.

Under suboptimal conditions that preserve certain levels of transcription pausing events during elongation (Fig. 2i, third lane), we found that nucleosome-specific pausing at SHL −4/−5 and −2 was notably reduced in condensates (Fig. 2i, fourth lane). To determine the spatial relationship between transcription and condensates, we performed RNase A protection assays and found that condensate-associated transcripts were more resistant to digestion than non-condensate transcripts (Fig. 2j). Super-resolution STORM microscopy further confirmed that transcripts localize within condensates rather than outside or at their surface (Fig. 2k,l). These results demonstrate that gel-like condensates actively facilitate transcription elongation by creating a membraneless reaction environment where Pol II can efficiently bypass nucleosome-specific pausing sites.

## NDF−FACT condensates aid transcription

To understand how these condensates function during transcription and observe their real-time behaviour, we employed single-molecule approaches to visualize both condensate formation and transcription dynamics simultaneously. We first used a high-resolution dual-trap optical tweezers[32,33] to determine how NDF and FACT affect Pol II dynamics during transcription through a single nucleosome (Fig. 3a). Without additional factors, Pol II exhibited pronounced pauses near SHL −5 and SHL −2 (Fig. 3b). In agreement with our biochemical results, individual factors alleviated some pausing sites and altered the overall transcriptional map (Extended Data Fig. 3a) but, remarkably, when both NDF and FACT were present, pauses at SHL −7 and −5 were greatly reduced and SHL −2 pauses shifted downstream (Fig. 3b,c). These data demonstrate that NDF and FACT together substantially alleviate all pausing events upstream of SHL −1.

To determine whether NDF−FACT condensates form on single Pol II−nucleosome complexes, we used a Lumicks C-Trap system,

which combines optical tweezers with confocal fluorescence microscopy (Fig. 3d and Extended Data Fig. 3b). We monitored condensate formation with fluorescently labelled proteins and simultaneously tracked Pol II transcription dynamics via optical tweezers. Pol II position was also determined using Cy5 fluorophores on the DNA handle attached to the polymerase. As transcription progresses, Pol II movement extends the upstream DNA, while Pol II, Cy5 fluorophores, and the right handle remain stationary under the detector. When individual factors were added, only faint NDF−GFP enrichment was detectable, and although some recruitment probably occurred (evidenced by changes in Pol II transcription dynamics; Extended Data Fig. 3e,f), fluorescent signals were too weak or photobleached too quickly for reliable detection. However, when both NDF−GFP and FACT−mCherry were added together, both fluorescent signals were clearly detected between the optical traps. Notably, the condensate fluorescence aligned with the Cy5 signal position, indicating NDF−FACT condensate formation in proximity to Pol II (Fig. 3e and Extended Data Fig. 3c,d). The condensate signal appeared when Pol II resumed transcription, before reaching the nucleosome, and remained associated throughout the process until Pol II dislodged from the template (Fig. 3e). Analysis of condensate size versus nucleosome crossing times revealed a trend where larger condensates were associated with faster transcription, though all condensates clearly enhanced efficiency compared to Pol II alone (Extended Data Fig. 3g,h).

To test whether condensates move with Pol II during transcription, we designed an asymmetric configuration using a 5.5 kb upstream DNA handle and a 3.5 kb downstream DNA handle tethered to Pol II. We ligated a 3.5 kb DNA template downstream of the Pol II elongation complex, followed by the nucleosome. This set up enabled tracking transcription over an extended duration, as transcribing the 3.5 kb template resulted in substantial bead separation (~1,200 nm extension).

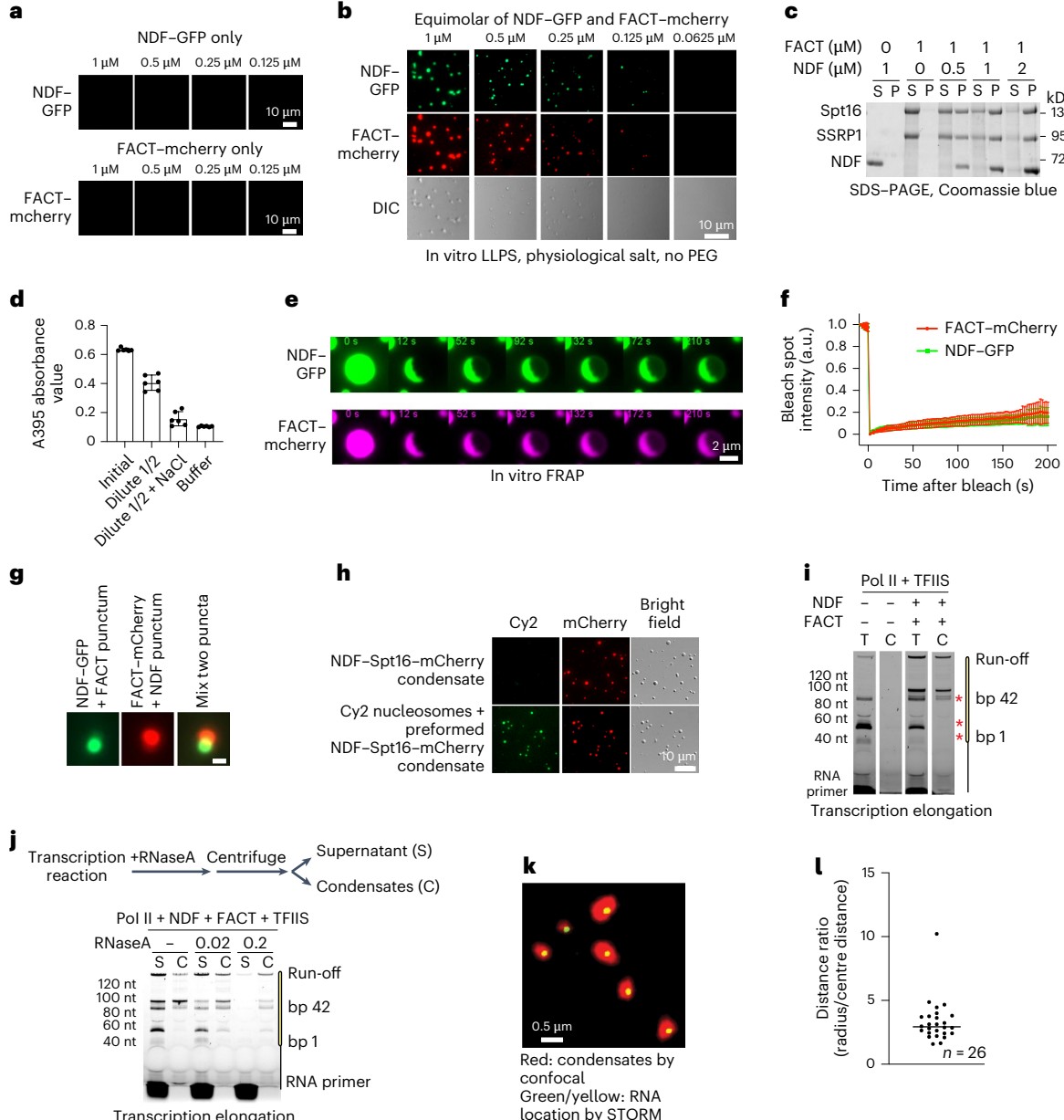

**Fig. 2 | NDF and FACT form phase-separated condensates. a**, Microscopic images show that neither NDF–GFP (monomeric GFP) nor FACT–mCherry (Spt16–mCherry + SSRP1) forms droplets at various concentrations in a 110 mM NaCl buffer at room temperature. **b**, Representative images of droplet formation when equimolar concentrations of NDF–GFP and FACT–mCherry are mixed at room temperature in a 110 mM NaCl buffer, without PEG (LLPS, liquid–liquid phase separation). Images taken with a Zeiss epifluorescence microscope. **c**, SDS–PAGE image from a sedimentation experiment illustrates the distribution of FACT and NDF in the aqueous (supernatant (S)) and condensed (pellet (P)) phases, mixed at specified concentrations in a 100 mM NaCl buffer without PEG. **d**, Normalized densitometry data from a reversibility assay where absorbance at 395 nm was measured for droplets formed with 3 μM FACT and 6 μM NDF, and after sequential 1:1 dilutions with and without added NaCl to 430 mM. Data represent mean ± s.d. from three independent experiments in technical triplicate. **e**, FRAP experiment images of NDF–GFP and FACT–mCherry condensates (equimolar at 0.8 μM) captured at specified times post-photobleaching (1 s). The panel labeled '0 s' corresponds to the pre-bleaching image; note that for quantitative FRAP analysis, time zero is defined as immediately after bleaching. **f**, Statistical FRAP recovery analysis shows the relative intensity of FACT–mCherry and NDF–GFP signals over time in in vitro formed droplets. Data are presented as average relative intensity ± s.d. ($n$ = 15 puncta, technical replicates). **g**, Partially overlapping but distinct NDF–FACT puncta do not merge

or mix. Mixed droplets of (NDF–GFP + FACT) with (NDF + FACT-mCherry) were incubated at room temperature for 10 min before imaging. Scale bar = 1.5 μm. **h**, Microscopic images display the incorporation of Cy2-labelled nucleosomes into pre-formed (NDF + Spt16–mCherry) condensates at room temperature. Protein concentrations were 1.2 μM. **i**, Distribution of transcription products in condensate phases. The transcription reactions were subjected to centrifugation to separate condensates (C) from the total reaction (T), and 0.2 μM NDF or FACT proteins were used for the reaction. RNA gel showing nucleosome-specific pausing at SHL −4/−5 and SHL −2 is reduced in the condensate fraction (lane 4). All lanes cropped from the same gel under identical conditions. **j**, Experimental scheme for an RNase A (ng μl$^{-1}$) protection assay with condensate-associated (C) and non-condensate transcripts (S) in the same reaction. In **a**–**c** and **g**–**j**, representative results from three independent experiments are shown. **k**, Overlay example image of STORM super-resolution microscopy of 6-FAM-labelled RNA positions (green/yellow) relative to condensate boundaries (red, captured by epifluorescence microscope, condensates were formed with purified NDF and FACT–mCherry) from 169 image acquisitions. Representative of two independent experiments. **l**, Quantitative analysis of RNA localization within condensates. The left axis shows the ratio of condensate radius to centre-to-centre distance (distance between condensate centre and RNA centre). Values >1 indicate RNA is located inside condensates, while values <1 would indicate RNA outside condensates ($n$ = 26 technical replicates).

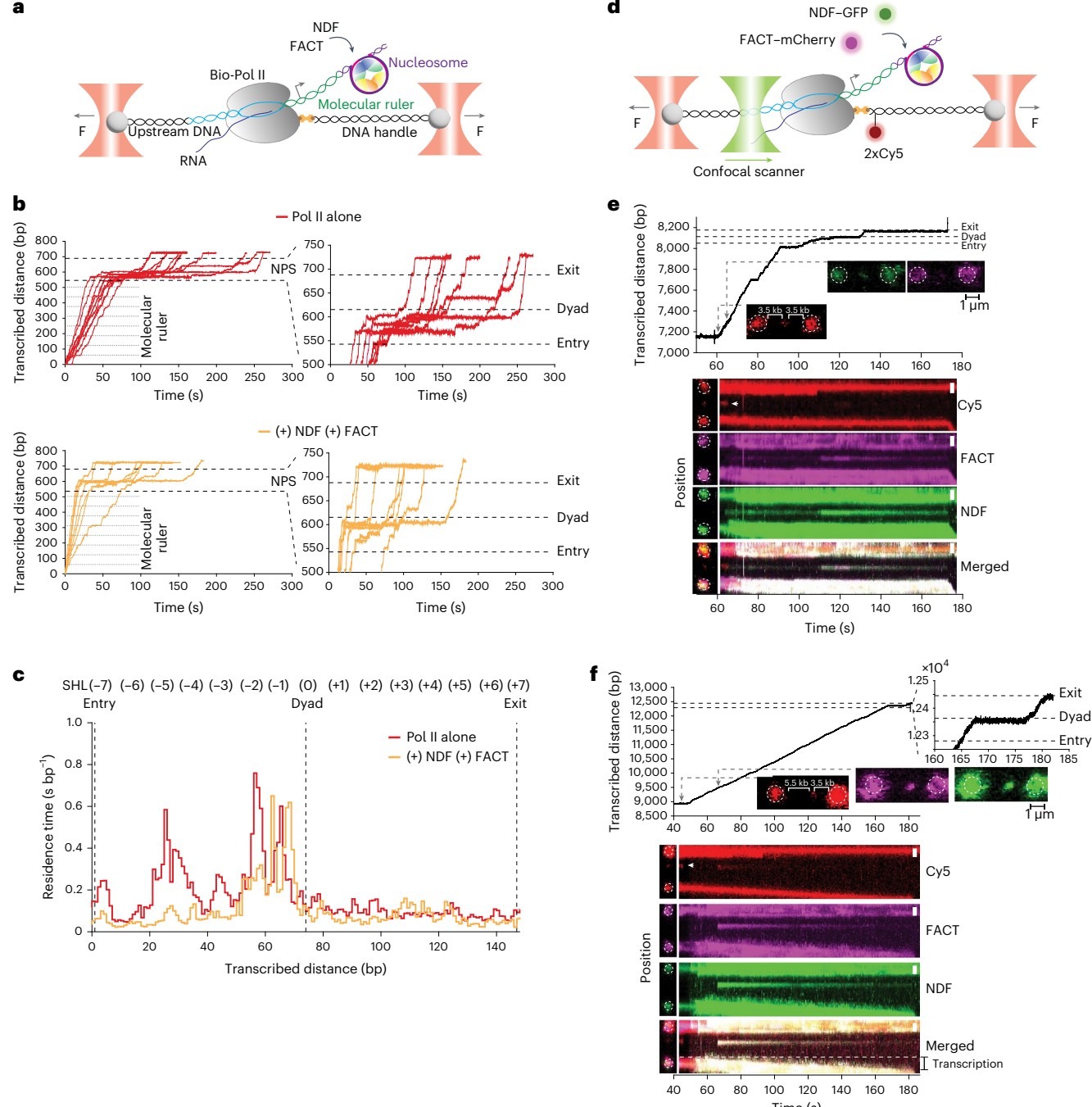

**Fig. 3 | NDF–FACT condensate forms on a single Pol II–nucleosome transcription elongation complex. a**, Experimental set-up for single-molecule transcription experiments. This assay involves forming a stalled biotinylated-Pol II (bio-Pol II) elongation complex, which is ligated downstream to a molecular ruler and a single human nucleosome and tethered between two beads optically trapped through two DNA handles: one attached to the polymerase and the other to the upstream DNA. The addition of ribonucleotide triphosphates induced the polymerase to resume transcription and to move forward on the template, causing the upstream DNA to get extended. The instrument is operated in constant force mode, which causes the beads to move apart by 1 bp for each bp that the enzyme advances on the template. This ensures that the tension remains constant, allowing the movement of the beads to reflect the progress of the polymerase. **b**, Representative trajectories of individual Pol II transcribing through the molecular ruler (dotted lines correspond to the eight pauses) and human NPS, either alone or in the presence of NDF and FACT. Zoomed-in views of Pol II molecules transcribing the nucleosomes are shown on the right. **c**, The median residence time of Pol II at each bp across the nucleosome. The mean

residence time is generated by averaging numerous single-molecule trajectories and serves as a quantitative description of the topography of the nucleosomal barrier. SHLs are shown. $N = 20$ biological replicates. **d**, A schematic of the C-trap experiments for simultaneous measurements of Pol II transcription in the optical tweezers channel and NDF–GFP and FACT–mCherry binding using a confocal microscope. Two Cy5 molecules in the SA–Biotin handle serve to mark the position of Pol II. **e,f**, Representative single-molecule traces of Pol II transcription (top) and fluorescence kymographs of NDF–GFP, FACT–mCherry and Cy5 signals of Pol II transcribing the molecular ruler and nucleosome template for the transcription experiment performed using a short template with 2 symmetrical 3.5 kb DNA handles (**e**) and transcription using a long template with asymmetric DNA handles (5.5 kb handle ligated to the upstream DNA and 3.5 kb handle tethered to Pol II) (**f**). The kymographs were obtained by continuously scanning the fluorescence lasers over the transcribing Pol II molecule until it reached the end of the template. The positions of the NPS entry, dyad and exit are indicated by dashed black lines. Scale bars, 1 μm.

Throughout this longer transcription process, the condensate was consistently detected at the Cy5 signal position and remained associated with the same location as the upstream DNA length and bead separation progressively increased (Fig. 3f). This demonstrated that the condensate maintains its position with Pol II, indicating dynamic association with the elongation complex rather than stationary binding on the template.

We next investigated whether condensates act on nucleosomes during transcription. We employed high-resolution optical tweezers to unwrap and rewrap nucleosomes by pulling on the DNA, mimicking the actions of Pol II through multiple cycles of transcription. The unwrapping signature allows us to distinguish between octasomes, hexasomes, tetrasomes and free DNA species, enabling real-time monitoring of nucleosome disassembly[33,34]. In control experiments without factors, nucleosomes disassembled into subnucleosomal species and free DNA after successive pulling cycles. NDF alone dramatically reduced octamers but produced excessive free DNA, indicating poor histone retention, while FACT alone increased hexasomes and reduced free DNA. Remarkably, combining both factors maintained ~60% hexasome-like species while reducing octamers by ~50%, with hexasome-like intermediates predominating after several cycles (Extended Data Fig. 3i). These results support our hypothesis that NDF–FACT condensates enhance nucleosome disassembly while maintaining chromatin integrity.

## NDF–FACT condensate, not protein interaction alone, drives synergistic enhancement of nucleosomal transcription

We then sought to determine whether phase separation itself, rather than just protein–protein interactions, is essential for enhanced transcriptional elongation. To address this, we first mapped the minimal interaction domains of NDF using truncated proteins (Extended Data Fig. 4a) and identified two regions involved in binding Spt16: the N-terminal PWWP domain and a second region between amino acids 160–180 (Extended Data Fig. 4b,c). Although both domains mediate protein–protein interaction with Spt16, only the second region is required for condensate formation (Extended Data Fig. 4d).

Focusing on this second region (Fig. 4a), alanine scanning mutagenesis pinpointed two conserved residues, K161 and R162, as crucial for the NDF–Spt16 interaction (Fig. 4b and Extended Data Fig. 4e,f). In a PWWP-less NDF construct, mutating either residue to alanine abolished its interaction with Spt16 (Fig. 4c). In full-length protein, NDF_K161A and NDF_R162A retained Spt16 interaction (Fig. 4d and Extended Data Fig. 4g) through the remaining PWWP domain (Extended Data Fig. 4b) but exhibited markedly reduced condensate formation capacity (Fig. 4e,f).

We then tested whether these mutants could achieve synergistic transcriptional enhancement. The purified NDF_K161A and NDF_R162A proteins retained normal catalytic activity in assisting Pol II transcription through nucleosomes (Extended Data Fig. 4h–j) but failed to show synergy with FACT (Fig. 4g,h). This loss of synergy correlated with impaired condensate formation rather than disrupted protein–protein interaction, suggesting that condensate formation is necessary for functional synergy.

Conversely, we tested the NDF_F1 truncate, which lacks the PWWP domain and exhibits notablyy reduced catalytic activity (Fig. 4i). Despite retaining the condensate formation ability with FACT (Extended Data Fig. 4d) and incorporating nucleosomes into these condensates (Extended Data Fig. 4k), NDF_F1 failed to exhibit synergistic transcription enhancement (Fig. 4i). Together, these results establish that both condensate formation and catalytic activity are necessary for synergistic transcription elongation.

## NDF–FACT condensates form in cells

To investigate the physiological relevance of our in vitro findings, we used CRISPR–Cas9 to tag endogenous NDF and FACT with fluorescent markers in human induced pluripotent stem (iPS) cell lines. We generated two single-knock-in lines (NDF–degron–GFP and Spt16–degron–GFP) and double knock-in lines (NDF–degron–GFP + Spt16–mScarlet and NDF–mScarlet + Spt16–degron–GFP) (Extended Data Fig. 5a). PCR and whole-genome sequencing analyses confirmed the correct knock-in (Extended Data Fig. 5b,c). Western blot analysis confirmed comparable expression levels to endogenous proteins (Extended Data Fig. 5d), and all cell lines retained the pluripotency marker Oct4 (Extended Data Fig. 5e) and differentiation capacity (Extended Data Fig. 5f).

Using these validated cell lines, fluorescence microscopy revealed that both NDF–mScarlet and Spt16–GFP formed nuclear puncta in live cells, with NDF forming distinct puncta and Spt16 showing additional diffuse nucleoplasmic signal. Importantly, NDF and Spt16 puncta largely colocalized, and these structures remained stable after formaldehyde crosslinking (Extended Data Fig. 6a).

To characterize these puncta further, we estimated their nuclear concentrations by measuring nuclear volume and quantifying protein amounts per cell (Extended Data Fig. 6b,c), determining concentrations of approximately 0.48 µM for NDF and 2.6 µM for Spt16 (Extended Data Fig. 6d). Despite Spt16 higher concentration, cytoskeleton (CSK) buffer treatment, which removes non-chromatin-bound nuclear proteins[35], eliminated 90% of Spt16 signal while leaving NDF largely unaffected (Extended Data Fig. 6e). This differential sensitivity revealed distinct chromatin association properties and improved puncta visualization for colocalization analysis. In double knock-in cells, after CSK buffer treatment, we observed substantial colocalization of NDF–mScarlet and Spt16–GFP puncta (Extended Data Fig. 6f). Owing to weak red fluorescent protein signals requiring prolonged exposures, we validated these observations using immunofluorescence with NDF antibody in Spt16–GFP cells (Fig. 5a and Extended Data Fig. 6g). Quantitative analysis yielded a Pearson correlation coefficient of $R = 0.70 \pm 0.06$ ($n = 20$), and three-dimensional imaging further corroborated colocalization across multiple focal planes (Extended Data Fig. 6h).

The substantial but incomplete overlap aligns with the diverse functional roles of FACT, including Pol I transcription in nucleoli and heterochromatin maintenance at the nuclear periphery[17–19]. Notably, cellular puncta exhibited heterogeneous sizes and less spherical morphologies compared with in vitro condensates, consistent with other endogenously expressed phase-separating proteins[36–38]. Additionally, our confocal microscopy with ~140 nm resolution probably overestimates smaller condensates while missing those below the detection threshold, as detailed in Methods section.

To test whether these puncta exhibit biophysical properties consistent with phase-separated condensates, we treated cells with 1,6-Hex, which significantly reduced both the number and normalized intensity of NDF and Spt16 puncta (Fig. 5b,c). While 1,6-Hex sensitivity supports condensate-like behaviour, we interpret these results cautiously, given potential effects on other cellular structures[39,40].

Having established the condensate-like properties of these puncta, we next investigated their functional relevance by examining their spatial relationship to transcription. RNA-FISH experiments revealed that both NDF and Spt16 puncta colocalize with actively transcribed genes, specifically the ACTB locus (Fig. 5d,e), demonstrating functional association with transcription sites. Immunofluorescence experiments revealed partial colocalization of NDF–FACT condensates with both Ser2- and Ser5-phosphorylated RNA Pol II condensates (Extended Data Fig. 6i–k), further supporting their association with transcriptionally active regions. This partial colocalization pattern is consistent with the temporal specificity of NDF–FACT function during transcription elongation.

Finally, we examined the dynamics of these cellular condensates through FRAP experiments. NDF puncta exhibited 20–30% recovery, similar to in vitro observations, while Spt16 puncta showed almost no

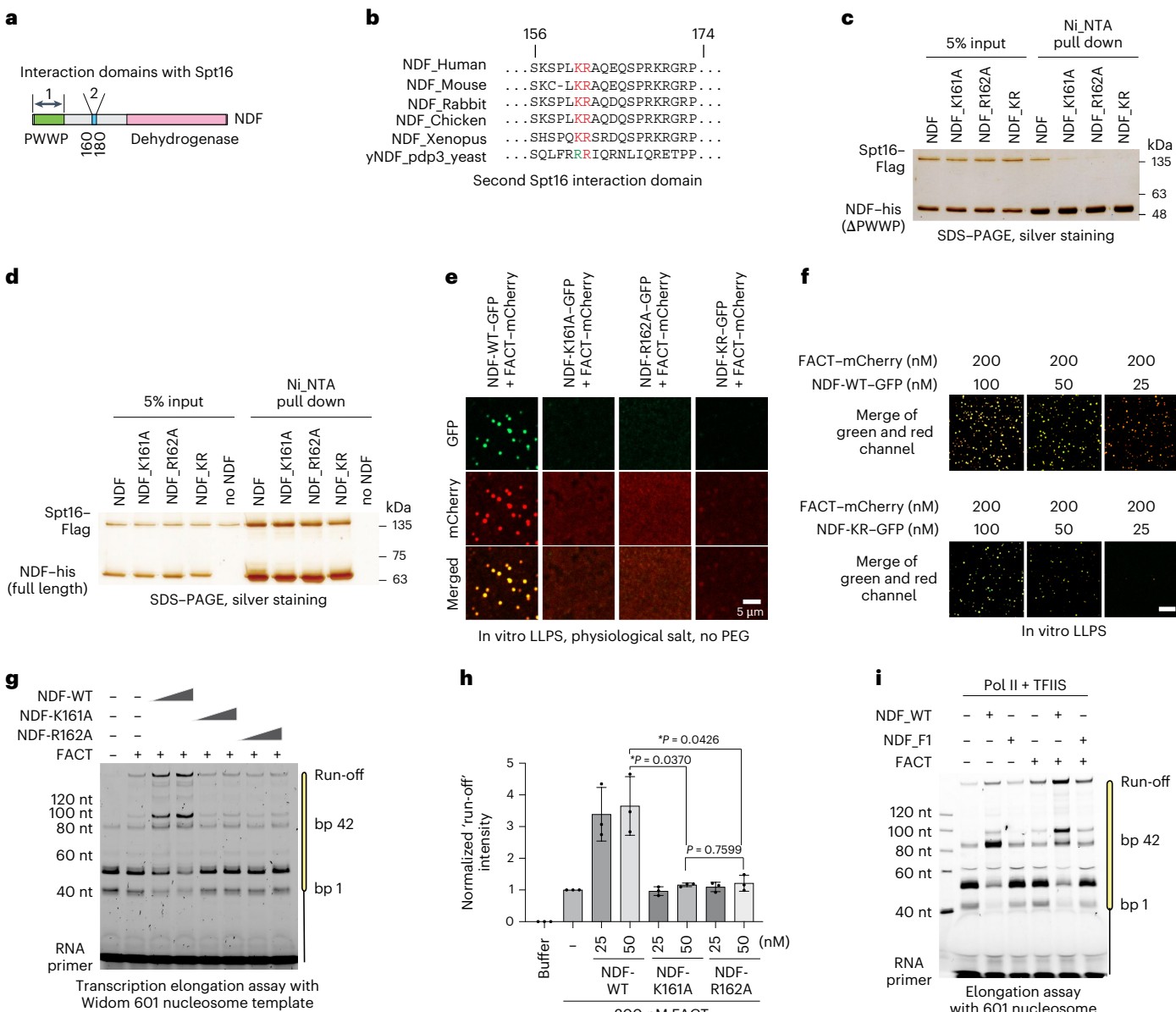

**Fig. 4 | NDF−FACT condensate formation, not merely protein interaction, is essential for synergistic transcription enhancement. a**, Domain scheme of the NDF protein and the focused regions that are required for interaction with Spt16. **b**, Amino acid sequence alignments around the second Spt16 interaction domain from different species. **c,d**, Interaction analysis between NDF mutants and Spt16 using Ni-NTA pull-down assays. PWWP-less NDF mutants (K161A and R162A) show abolished interaction with Spt16 (**c**), while full-length versions of the same mutants retain Spt16 binding due to intact PWWP domain interaction (**d**). **e**, Representative fluorescence microscopy images showing condensate formation between NDF−GFP variants (WT, K161A, R162A and KR; 25 nM each) and FACT−mCherry (0.2 µM) in droplet formation buffer at room temperature. **f**, Droplet formation between NDF-WT−GFP, NDF-KR−GFP and FACT−mCherry

at 200 nM for FACT−mCherry and specific concentrations for NDF variants in droplet formation buffer. Scale bar, 10 µm. **g**, Nucleosome transcription assay with full-length NDF-WT (25 nM and 50 nM), NDF-K161A (25 nM and 50 nM) or NDF-R162A (25 nM and 50 nM) mutants, together with 200 nM FACT. Transcription reactions were performed with 0.25 mM NTPs for 10 min. **h**, Statistical analysis of the run-off product from **g** (mean ± s.d., $n = 3$ biological replicates). Normalized to run-off product from FACT alone (*$P < 0.05$ with a two-tailed $t$-test). **i**, Nucleosome transcription assay with 0.2 µM NDF and NDF_F1 truncates and FACT with 0.3 µM TFIIS. Transcription reactions were performed with 0.25 mM NTPs for 10 min at 30 °C. Representative results from three independent experiments are shown in **a−f** and **i**.

recovery, even though the non-chromatin-bound Spt16 signal in the bleached region recovered to nearly the same level as non-bleached control regions (Fig. 5f,g and Extended Data Fig. 6i). This indicates high mobility of the large quantity of non-chromatin-bound Spt16 in the nucleoplasm, which can compensate for the loss of puncta intensity after photobleaching. Collectively, these cellular data correlate with our in vitro observations and support NDF−FACT condensate formation in cells.

## NDF−FACT condensates are important for their chromatin occupancy

Having established that NDF and FACT form condensates in cells, we next investigated their functional significance. Our single-molecule studies demonstrated that phase separation recruits large quantities of both proteins to elongating Pol II, synergistically enhancing transcription elongation (Fig. 3). To test whether this condensate-mediated recruitment mechanism operates in cells, we first performed chromatin

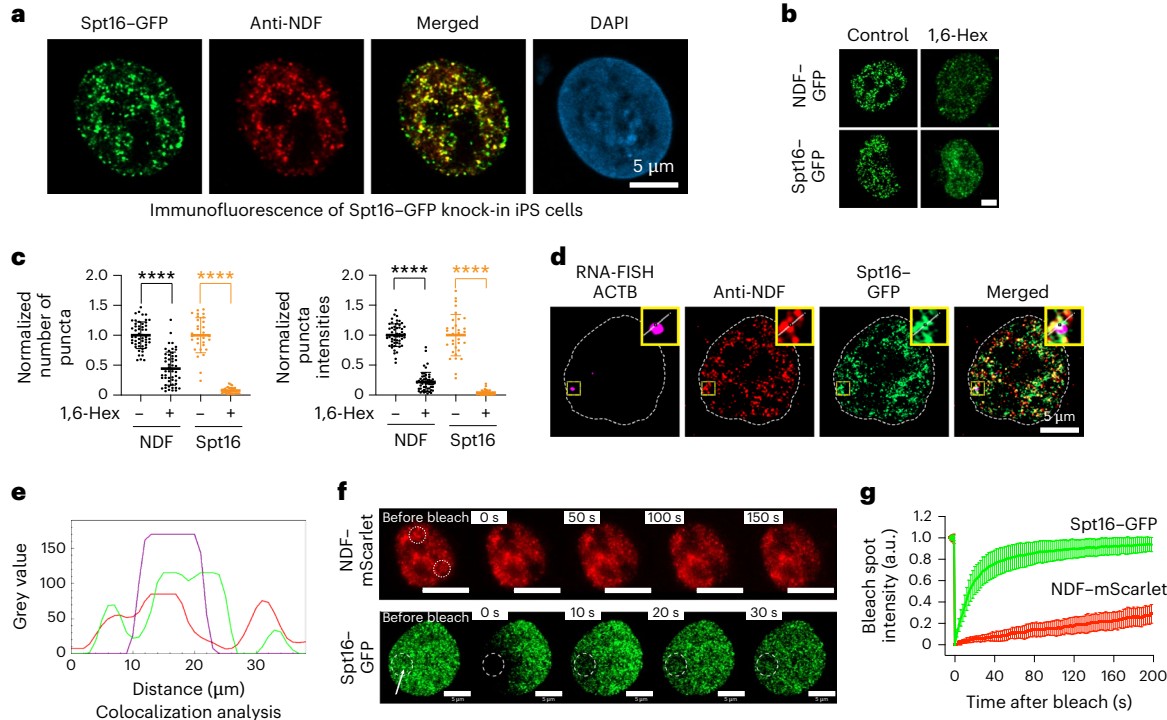

**Fig. 5 | NDF and FACT form phase-separated condensates that colocalize at active transcription sites in human cells. a**, Immunofluorescence analysis of Spt16–GFP knock-in iPS cells after 1 min CSK buffer treatment. Cells were immunostained for NDF (Alexa Fluor 555) and counterstained with DAPI. **b**, Images showing endogenous Spt16–GFP and NDF–GFP condensates in response to 3% 1,6-Hex treatment. Cells were pre-extracted with CSK buffer before fixation to remove non-chromatin-bound proteins. Scale bar, 5 μm. **c**, Scatter plots show normalized numbers (left) and intensities (right, normalized to total cellular intensity) of NDF–GFP and Spt16–GFP puncta after treatment with either vehicle or 3% 1,6-Hex. Statistical significance was determined using two-sided non-parametric t-tests, n (technical replicates) = 52, 53, 32 and 34 for NDF vehicle, NDF 1,6-Hex, Spt16 vehicle and Spt16 1,6-Hex, respectively (mean ± s.d., ****P < 0.0001). **d**, Colocalization of NDF, Spt16–GFP

and ACTB nascent RNA visualized by RNA-FISH in double knock-in iPS cells. NDF was stained with NDF antibody Alexa 555. The nuclear periphery, indicated by a dotted line, was defined by Hoechst staining. **e**, Colocalization analysis of RNA-FISH, NDF and Spt16–GFP signals. Magenta line, RNA-FISH signal; green, Spt16–GFP; red, NDF–Alexa 555. **f**, FRAP of endogenously expressed Spt16–GFP and NDF–mScarlet in human iPS cells, captured at specified intervals post-photobleaching. Scale bars, 10 μm for NDF–mScarlet. Photobleached areas are indicated by dotted circles. The white arrow points to a Spt16–GFP punctum. **g**, Statistical analysis of FRAP experiment. Spt16–GFP and NDF–mScarlet signal intensities (left axis) relative to the photobleached signal are plotted over time (bottom axis). Data represent mean relative intensity ± s.d. for n = 32 (Spt16–GFP, technical replicates) and n = 22 (NDF–mScarlet, technical replicates). Representative results from three independent experiments are shown in **a** and **d**.

immunoprecipitation sequencing (ChIP–seq) analysis of NDF and Spt16 chromatin occupancy in iPS cells. We found that NDF and Spt16 occupy largely the same active genes, with NDF predominantly enriched in transcribed regions and Spt16 present in both promoter-proximal and gene body regions (Fig. 6a), consistent with the known role of FACT in promoting nucleosome dynamics at gene bodies and regulatory regions[41]. K-means clustering of genome-wide occupancy patterns revealed that clusters 1–3 contain active genes while cluster 4 consists predominantly of inactive genes (Fig. 6b). Analysis of genomic features across these clusters, including GC content, nucleosome-depleted region lengths and pausing indices, revealed some trends between clusters, though the large number of genes within each cluster made definitive mechanistic interpretations challenging (Extended Data Fig. 7a–c). Importantly, genome-wide analysis demonstrated strong NDF–Spt16 co-occupancy, with few instances of high NDF/low Spt16 or vice versa (Extended Data Fig. 7d).

To test whether condensates are required for their chromatin enrichment, we treated cells with 1,6-Hex. If condensate formation drives chromatin recruitment, then 1,6-Hex treatment should reduce NDF and Spt16 occupancy. While we interpret 1,6-Hex results cautiously, given potential off-target effects[39,40], NDF chromatin enrichment was dramatically reduced within 10 min (Fig. 6c,d). This was unexpected, as the localization of NDF was previously attributed to the PWWP domain binding to H3K36me3 (refs. 7,8).

However, control experiments showed that LEDGF, another PWWP-containing protein that binds H3K36me3 (ref. 42), was unaffected by 1,6-Hex (Extended Data Fig. 7e–g), indicating that 1,6-Hex specifically disrupts condensate-dependent recruitment rather than PWWP–histone interactions. The chromatin occupancy of Spt16 was also reduced, though less dramatically than that of NDF. Intriguingly, Pol II occupancy increased, which may be partly due to reduced transcriptional efficiency, in line with our in vitro observation that 1,6-Hex can inhibit Pol II (Extended Data Fig. 7h).

While 1,6-Hex treatment is widely used in the phase-separation field, it can have nonspecific effects and may target other cellular processes[39,40]. To investigate NDF–FACT condensate function with greater specificity, we utilized the auxin (Aux)-inducible degron system. Our knock-in stem cells express proteins tagged with both fluorescent markers and Aux-inducible degron cassettes (Extended Data Fig. 5a), allowing rapid and specific protein depletion upon Aux addition. In NDF degron cells, Aux treatment achieved near-complete depletion within 2 h, while Spt16 and GAPDH levels remained stable (Fig. 6e). Cut&Run experiments confirmed the NDF removal from chromatin (Extended Data Fig. 7i). Although prolonged NDF depletion caused cell death (Extended Data Fig. 7j–l), short-term depletion (≤2 h) allowed functional studies without cell viability issues (Extended Data Fig. 7m).

Using this approach, we found NDF depletion markedly reduced both the number and intensity of Spt16–mScarlet puncta (~50%

reduction; Fig. 6f,g), suggesting that NDF plays a key role in FACT puncta formation. The ~50% reduction indicates that NDF may not be the sole contributor to Spt16 puncta formation in cells. We next assessed whether NDF depletion also affects chromatin occupancy by performing Spt16 ChIP–seq, which showed substantial reduction in Spt16 occupancy at coding regions (Fig. 6h). This effect was evident even after 1 h of partial NDF depletion (Extended Data Fig. 7n,o). Metagene analysis demonstrated globally reduced Spt16 occupancy following NDF depletion (Fig. 6i). As expected, reduced NDF and FACT chromatin occupancy impaired transcription through chromatin, leading to increased Pol II accumulation at promoter-proximal pausing sites (Fig. 6j,k).

Previous studies show that FACT chromatin occupancy requires active transcription[43,44]. Given this dependence, the reduced FACT chromatin occupancy and increased Pol II pausing following NDF depletion could reflect two scenarios: either FACT requires NDF-mediated condensates to recruit, or FACT occupancy simply decreases as a consequence of impaired transcription. To distinguish between these possibilities, we performed MNase-seq to measure nucleosome maintenance. FACT is known to maintain nucleosome stability during transcription by binding transiently exposed H2A/H2B surfaces[22,23,25,44]. If FACT functions independently of NDF, nucleosomes should be maintained at actively transcribed sites despite overall reduced transcription. However, NDF depletion led to reduced nucleosome occupancy particularly at transcriptionally active genes (Fig. 6l,m). Since NDF lacks nucleosome maintenance activity (Extended Data Fig. 3i), this indicates FACT requires NDF for effective nucleosome engagement, suggesting that condensate-mediated recruitment is essential for FACT function.

To examine the reciprocal relationship and test whether FACT influences NDF localization, we first attempted Spt16 depletion using Aux-inducible degron. However, because Spt16 is highly abundant in iPS cells, Aux treatment for 4 h achieved only ~30% protein reduction, while prolonged depletion (24 h) causes extensive cell death (Extended Data Fig. 8a,b). Under this partial Spt16 depletion, NDF ChIP–seq showed only a modest effect at transcribed regions (Extended Data Fig. 8c,d).

To more effectively disrupt FACT function, we treated cells with curaxin (CBL0137), which displaces FACT from actively transcribed chromatin[20,45]. Importantly, curaxin did not alter total Spt16 protein levels (Extended Data Fig. 8e), consistent with functional inhibition rather than protein degradation. In DMSO-treated control cells, Spt16 localized to discrete, bright nuclear puncta as expected. After curaxin treatment, these Spt16 puncta became diffuse and lost their distinct focal pattern (Extended Data Fig. 8f,g). Notably, NDF exhibited a coordinated response: well-defined puncta became less distinct and more diffuse (Extended Data Fig. 8h), with markedly reduced puncta

intensity despite unchanged puncta number (Extended Data Fig. 8i). While curaxin has broader chromatin effects as a DNA intercalator[46], the coordinated disruption of both FACT and NDF puncta support their interdependent relationship.

## NDF−FACT condensates drive transcription and maintain chromatin in cells

While NDF depletion demonstrated the importance of NDF in FACT recruitment, it could not specifically distinguish whether this recruitment depends on NDF−FACT phase separation or simply protein−protein interactions. To directly test the functional significance of phase separation, we utilized the NDF_K161A mutation, which retains protein−protein interaction with FACT and catalytic activity but reduces phase separation ability and eliminates synergistic transcription enhancement (Supplementary Table 2).

Using CRISPR−Cas9, we generated NDF_K161A-expressing human iPS cells from single-cell clones and verified precise knock-in by sequencing. Western blot confirmed normal NDF and Spt16 protein levels (Fig. 7a), and cells maintained pluripotency as evidenced by Oct4 staining (Extended Data Fig. 9a). To assess the functional consequences of impaired condensate formation, we examined cellular phenotypes. Although NDF_K161A stem cells remained viable and pluripotent, they grew noticeably slower than isogenic wild-type (WT) or heterozygous cells (Fig. 7b), suggesting functional defects when condensate formation is impaired. This growth defect prompted us to investigate the underlying transcriptional basis.

We performed global transcriptional profiling to quantify the impact of the K161A mutation on mRNA production. mRNA sequencing of steady-state transcripts (RNA-seq, normalized to ERCC spike-in controls) revealed a broad reduction in transcript levels across the genome in NDF_K161A cells (Fig. 7c). To assess the role of condensates in differentiation, we differentiated NDF-WT and NDF-K161A cells into cardiac lineages (Extended Data Fig. 9b). RNA-seq analysis showed modest differences at the cardiac progenitor cell stage but pronounced changes at the later fibroblast stage, with thousands of genes altered in NDF-K161A cells (Extended Data Fig. 9c–f). While some lineage-specific genes remained unaffected, these results suggest that NDF−FACT condensates contribute to gene expression regulation not only at the stem cell stage but also during differentiation.

To pinpoint how transcription is disrupted, we analysed nascent RNA synthesis using precision nuclear run-on sequencing (PRO-seq)[47]. PRO-seq data indicated that RNA Pol II accumulates at promoter-proximal regions in NDF_K161A cells, consistent with increased promoter-proximal pausing (Fig. 7d,e). To directly assess elongation dynamics, we compared nascent RNA synthesis rates between WT and K161A cells using transient transcriptome sequencing

---

**Fig. 6 | NDF depletion disrupts FACT condensates, reduces chromatin occupancy and impairs transcription. a**, Genome browser visualization showing co-occupancy of NDF, Spt16 and Rpb2 (Pol II) at selected genomic sites in human iPS cells. **b**, A heat map depicting *K*-means clustering analysis of genome-wide ChIP–seq data for NDF, Spt16 and Pol II. Cluster 1 contains mostly small genes. Clusters 2 and 3 contain active genes with high NDF and Spt16 occupancy. Cluster 4 includes a mix of active genes with low NDF/Spt16 occupancy (1/3) and inactive genes without NDF/Spt16 occupancy (2/3). **c**, Genome browser views of NDF, Spt16 and Pol II ChIP–seq data at the SLC2A1 locus, comparing untreated cells with those treated with 1.5% 1,6-Hex for 10 min. Data were normalized to total reads. **d**, Metagene analysis showing NDF, Spt16 and Pol II ChIP–seq signal distribution in untreated and 1,6-Hex treated human iPS cells across gene-coding regions, extending 5 kb upstream of the transcription start site (TSS) and 5 kb downstream of the transcription termination site (TTS). **e**, Western blot analysis of NDF and Spt16 protein levels in NDF−degron iPS cells treated with 0.5 mM Aux for the indicated timepoints. Whole-cell extracts were used for the experiment. The '4 h recover' lane indicates NDF protein levels 24 h after Aux removal, following a 4-h Aux treatment. Representative results from three independent

experiments. **f**, Representative microscopic images of endogenous Spt16−mScarlet puncta with and without 2 h Aux treatment. Scale bars, 5 μm. **g**, A scatter plot of normalized numbers (left) and normalized intensities (right, relative to total cell intensity) of Spt16−mScarlet puncta upon Aux treatment. Statistical analysis was performed using non-parametric two-sided *t*-tests; *n* = 30 untreated and 61 treated (technical replicates), (mean ± s.d., ****$P$ < 0.0001). **h,i**, Genome browser view (**h**) and Metagene analysis (**i**) of Spt16 ChIP–seq from untreated and 2 h Aux-treated NDF−degron iPS cells. Data were normalized to total reads. CPM, counts per million. **j,k**, Genome browser view (**j**) and Metagene analysis (**k**) of Rpb2/Pol II ChIP–seq from untreated and Aux-treated (2 h) NDF−degron iPS cells showing rapid NDF depletion increases Pol II pausing at promoter-proximal sites. Data were normalized to total reads. **l**, Metagene analysis showing the mean smoothed nucleosome signal from MNase-seq after NDF depletion (Aux 2 h) and control (DMSO) experiments, normalized to total reads of transcription inactive genes and centred at the TSS of all genes. **m**, Metagene analysis of transcriptionally inactive genes around the TSS, normalized to total reads from transcriptionally inactive genes.

(TT-seq)[48], which showed decreased nascent RNA synthesis rates in NDF_K161A cells (Extended Data Fig. 9g). Unexpectedly, shorter genes were more severely affected than longer genes, suggesting that NDF–FACT condensates may play a more important role in transcription of smaller genes and indicating complex relationships between condensate function and gene architecture (Extended Data Fig. 9h). Additionally, we performed TT-seq coupled with 5,6-dichlorobenzimidazole 1-β-ᴅ-ribofuranoside (DRB) release, which measures transcription resumption upon release from DRB-mediated promoter-proximal pausing. We found that both the nascent RNA synthesis rate upon DRB release and the transcription elongation rate were reduced in NDF_K161A cells (Fig. 7f–h). These data provide functional evidence

that NDF–FACT phase separation plays important roles for efficient transcription elongation in stem cells.

Having established transcriptional defects, we next tested whether these stem from impaired FACT recruitment to chromatin. Immunofluorescence revealed ~25% reduction in Spt16 puncta number and dimmer remaining foci in NDF_K161A cells (Fig. 7i,j), though the effect was less severe than complete in vitro loss, possibly because FACT is exceptionally abundant in stem cell nuclei. Spt16 ChIP–seq using *Drosophila* spike-in normalization confirmed substantially decreased occupancy at transcriptionally active regions (Fig. 7k,l), demonstrating that phase separation capability, rather than just NDF presence, is required for efficient FACT chromatin recruitment.

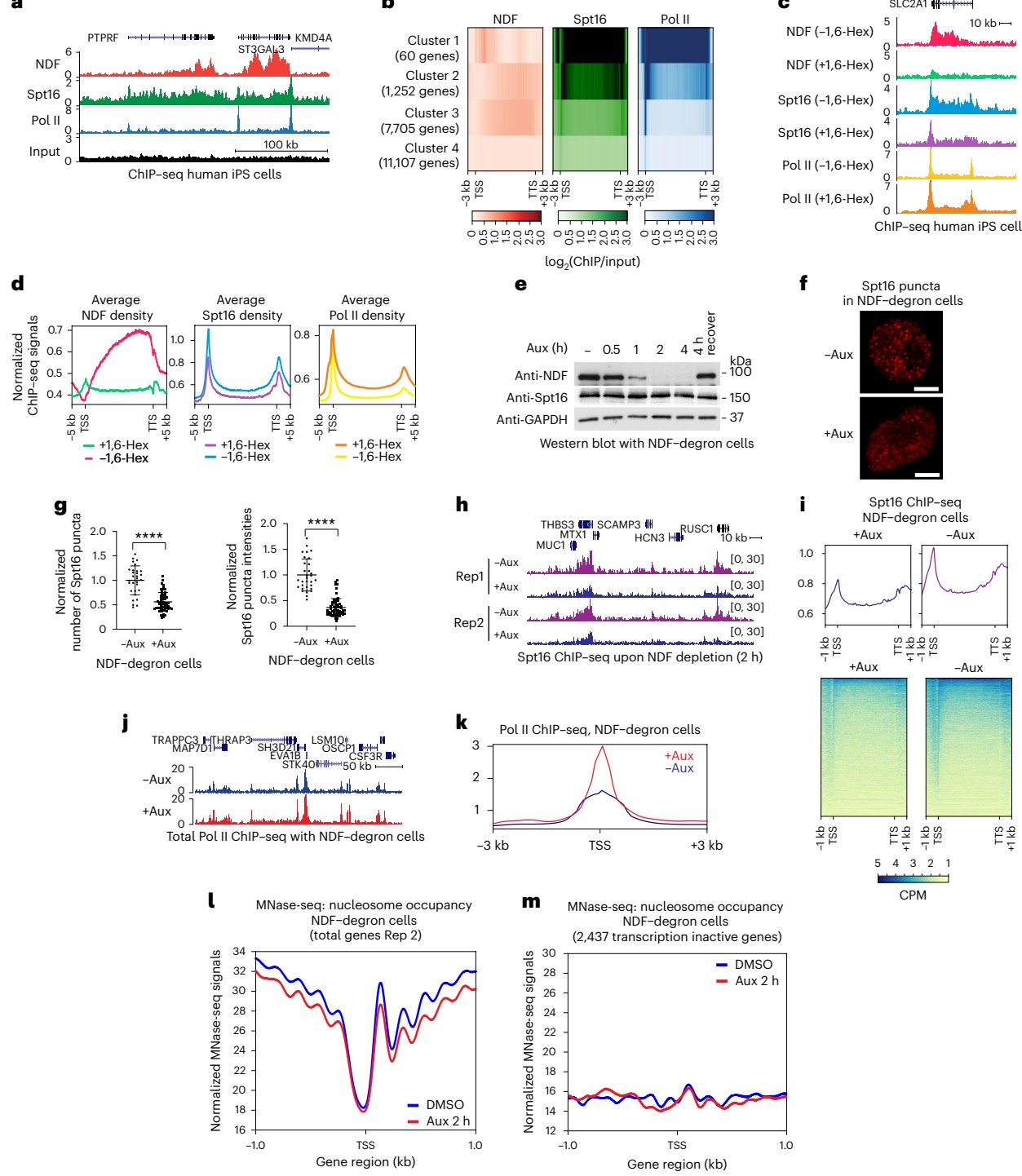

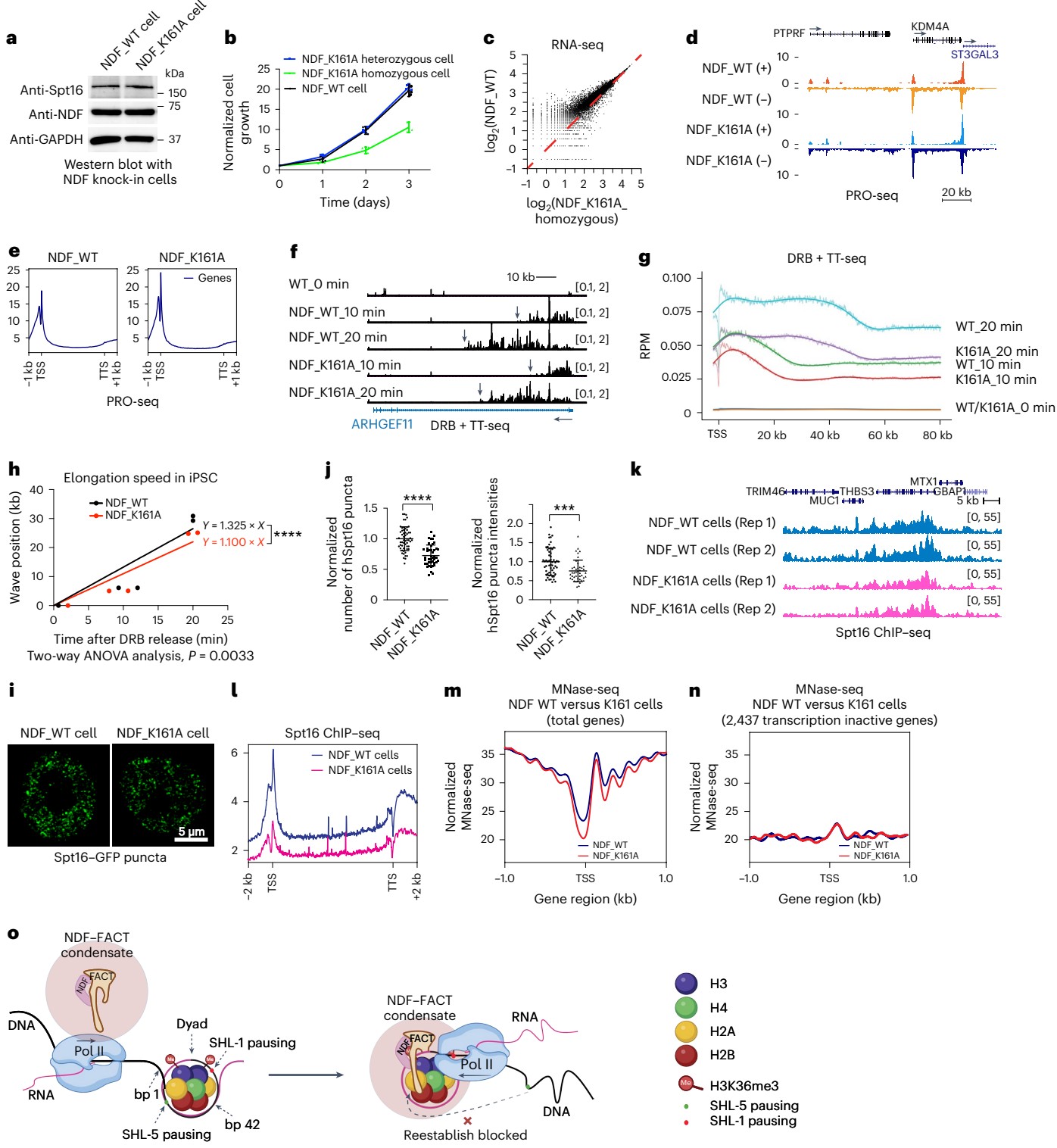

Finally, we examined how the loss of NDF–FACT condensates affects chromatin organization at active genes. Using MNase-seq, we assessed nucleosome occupancy in NDF_K161A versus WT cells. To ensure reliable comparisons, we performed two independent normalizations: one to total reads of inactive genes (Fig. 7m,n) and another using a yeast chromatin spike-in to normalize for MNase digestion efficiency (Extended Data Fig. 9i,j). Both analyses revealed reduced nucleosome occupancy at transcriptionally active loci in NDF_K161A cells. Despite only a ~25% decrease in Spt16 puncta, the failure of FACT to effectively engage with chromatin in these cells leads to marked

nucleosome loss during active transcription. These findings demonstrate that proper nucleosome maintenance during transcription depends on phase separation-mediated FACT activity, revealing a novel mechanism by which condensates enable FACT to fulfil its critical role in preserving chromatin integrity during transcription in human stem cells.

## Discussion

A growing body of evidence indicates that gene transcription is organized within spatially concentrated biomolecular condensates formed by

**Fig. 7 | NDF−FACT phase separation is essential for FACT recruitment, transcription elongation and nucleosome stability in human stem cells.** **a**, Western blot showing NDF and Spt16 protein levels in NDF−K161A knock-in homozygous iPS cells. **b**, Cell growth curve of NDF-WT, NDF-K161A homozygous and heterozygous iPS cells, seeded and maintained under the same conditions, counted every 24 h over 3 independent experiments (mean ± s.d.). **c**, A scatter plot comparing RNA-seq transcriptome data of NDF-WT and NDF-K161A homozygous iPS cells. Average $\log_2$ reads per kilobase million values from NDF-WT plotted against values from NDF-K161A homozygous. Data were normalized to spike-in ERCC control RNAs. **d,e**, Genome browser view (**d**) and Metagene analysis (**e**) (both strands) of PRO-seq data from NDF-WT and NDF-K161A homozygous iPS cells, showing increased Pol II pausing at promoters. Data were normalized to total reads. **f,g**, Genome browser view (**f**) and Metagene analysis (**g**) (non-overlapping genes (50–300 kb, $n = 2,558$), both strands) of DRB + TT-seq data from NDF-WT and NDF-K161A homozygous iPS cells, showing decreased transcription elongation. Data normalized to *Schizosaccharomyces pombe* nascent RNA spike-in. **h**, Quantification of Pol II elongation rates (Y) after DRB release using linear regression (mean ± s.d., $n = 2,558$ long genes analysed, 2 biological replicates, two-way ANOVA test, $P = 0.0033$). **i**, Representative microscopic images of endogenous Spt16 puncta in NDF_WT and NDF_K161A iPS cells. **j**, A scatter plot of normalized numbers (left) and normalized intensities (right) of Spt16 puncta in NDF-WT and NDF-K161A homozygous cells. Non-parametric two-sided *t*-tests; $n = 56$ (NDF-WT, technical replicates) and 43 (NDF-K161A homozygous, technical replicates) (mean ± s.d., ***$P < 0.001$ and ****$P < 0.0001$). **k,l**, Genome browser view (**k**) and Metagene analysis (**l**) of Spt16 ChIP−seq data from NDF-WT and NDF-K161A homozygous iPS cells. Data were normalized to spike-in *Drosophila* chromatin. **m**, Metagene analysis showing the mean nucleosome signal from MNase-seq with NDF-WT and NDF-K161A cells, normalized to total reads of inactive genes and centred at TSS. **n**, Metagene analysis of transcriptionally inactive genes around the TSS, normalized to total reads from inactive genes. **o**, An action model of NDF−FACT condensates mediated Pol II transcription through nucleosomes. Panel **o** created with BioRender.com.

phase separation[49–51]. Key components of the transcription machinery, including Pol II, Mediator and various transcription factors can coalesce at active enhancers and promoters to create high-density 'transcription hubs' that drive efficient initiation. During elongation, Pol II transitions between different condensate environments through CTD phosphorylation, dissociating from Mediator condensates and associating with elongation-specific condensates[37,38]. NDF−FACT condensates exhibit distinct properties compared with other transcriptional condensates. While most condensates self-assemble through their own IDRs, NDF−FACT condensates require heterotypic interactions between two distinct proteins, with neither forming condensates alone. These condensates show gel-like properties with slower dynamics compared with highly dynamic RNA Pol II condensates, and uniquely travel processively with transcribing Pol II to facilitate nucleosome barrier traversal.

In this study, we have uncovered how a phase separation-mediated mechanism enables NDF and FACT to cooperate during Pol II transcription elongation through chromatin. We propose a model where, after the transcription initiation and pause release of Pol II, NDF−FACT condensates are recruited to support the progression of Pol II through chromatin, probably via interactions between NDF and Pol II (ref. 11). When Pol II encounters a nucleosome, condensates can actively incorporate the nucleosome, substantially increase the local residence time of both proteins and enhancing their catalytic efficiency. Inside the condensate, NDF destabilizes the nucleosome, allowing Pol II to efficiently bypass the pausing near SHL −7 and SHL −4/5. As Pol II moves into the nucleosome, the interaction domains of FACT become exposed, enabling FACT to engage with the partially disassembled nucleosome[22,23,25] and prevent DNA re-attachment[52] while preserving histones[33]. This coordinated process lowers the energy barrier for Pol II traversal, promoting efficient elongation while maintaining chromatin integrity (Fig. 7o and Extended Data Fig. 10).

This condensate-mediated mechanism probably operates within a broader network of elongation factors. Our proteomic data identified PAF1 complex and Spt5 as NDF-associated partners, suggesting that elongation condensates serve as molecular hubs coordinating multiple factors. Additionally, chromatin remodellers such as Chd1 and histone chaperones such as Spt6, which coordinate with FACT during transcription[44,53,54], may transiently associate with these condensates. Given that NDF binds H3K36me3 through its PWWP domain[7,8], phase separation may also integrate histone modifications with transcription elongation.

Why would condensates be necessary rather than simpler protein complexes? The speed and efficiency required for transcription elongation under physiological conditions probably necessitates this organization[18]. RNA Pol II transcribes at approximately 3.5 kilobases per minute in cells[55], demanding rapid nucleosome disassembly and reassembly. While FACT alone can promote nucleosome bypass in vitro[13–15], it requires extended time, higher concentrations and cannot effectively overcome certain pausing sites under our conditions. NDF-mediated phase separation provides high local FACT concentrations and a dynamic environment facilitating efficient nucleosome restructuring. In this context, even transient disruptions to NDF−FACT condensates can result in severe transcription defects and chromatin instability in cells.

Several observations suggest additional complexity in FACT regulation. The partial colocalization of NDF and FACT puncta indicates that some FACT functions occur independently of NDF, possibly reflecting FACT's roles in other chromatin-related processes[17–19]. Similarly, not all actively transcribing genes exhibit detectable FACT or NDF occupancy, supporting the existence of alternative mechanisms for facilitating Pol II transcription through chromatin[42]. Understanding these NDF−FACT condensate mechanisms may ultimately inform the development of more specific cancer therapeutic strategies targeting FACT, given its importance in chromatin biology and established therapeutic potential.

## Online content

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

[1]Department of Molecular Biology and Biochemistry, Rutgers University, Piscataway, NJ, USA. [2]Department of Molecular and Cell Biology, University of California, Berkeley, CA, USA. [3]California Institute for Quantitative Biosciences, University of California, Berkeley, CA, USA. [4]Howard Hughes Medical Institute, University of California, Berkeley, CA, USA. [5]Jason Choy Laboratory of Single Molecule Biophysics, University of California, Berkeley, CA, USA. [6]Biophysics Graduate Group, University of California, Berkeley, CA, USA. [7]Department of Cell Biology and Neurosciences, Rutgers University, Piscataway, NJ, USA. [8]Department of Chemistry, University of California, Berkeley, CA, USA. [9]The Salk Institute for Biological Studies, La Jolla, CA, USA. [10]Department of Chemistry and Chemical Biology, Rutgers University, Piscataway, NJ, USA. [11]Department of Biomedical Sciences, School of Medicine and Health Sciences, University of North Dakota, Grand Forks, ND, USA. [12]Cancer Institute of New Jersey, Rutgers University, New Brunswick, NJ, USA. [13]Department of Integrative Structural and Computational Biology, The Scripps Research Institute, La Jolla, CA, USA. [14]Graduate School of Biological Sciences, Section of Molecular Biology, University of California San Diego, La Jolla, CA, USA. [15]Kavli Energy Nanoscience Institute, University of California, Berkeley, CA, USA. [16]Department of Physics, University of California, Berkeley, CA, USA. [17]Molecular Biophysics and Integrative Bioimaging Division, Lawrence Berkeley National Laboratory, Berkeley, CA, USA. [18]These authors contributed equally: Ziwei Li, Francesca Burgos-Bravo. ✉e-mail: carlosb@berkeley.edu; jia.fei@rutgers.edu

## Methods

To ensure the reproducibility of the results, each experimental condition was independently performed at least twice, with most experiments conducted three times.

### Cell culture

The human iPS cell line GM25256 (Coriell Institute for Medical Research) was cultured in Essential 8 medium (Thermo Fisher) per the manufacturer's instructions. Rock inhibitor Y23672 (Selleck Chemicals) was added to a final concentration of 10 μM during cell passaging. Hela, HEK293T and SW480 cells were maintained in DMEM (Corning) with 10% FBS (Corning) and 100 U ml$^{-1}$ penicillin–streptomycin (Thermo Fisher). All mammalian cells were incubated at 37 °C with 5% $CO_2$. iPS cells were differentiated into neural progenitor cells using PSC Neural Induction Medium (Gibco) following the manufacturer's protocol. iPS cells were differentiated into cardiac progenitor cells and epicardial-derived fibroblasts as described[56].

### Antibodies

Rabbit polyclonal antisera against hNDF were generated as previously described[10]. The following commercial primary antibodies were used in this study: anti-hSpt16 (Cell Signaling, 12191S), anti-GAPDH (Cell Signaling, 5174S), anti-LEDGF (Proteintech, 25504-1-AP), anti-H3 (Cell Signaling, 4499), anti-H3K36me3 (Abcam, ab9050), anti-Rpb2/Pol II (Genetex, GTX102535), anti-Oct4 (Cell Signaling, 2750S), anti-NDF/GLYR1 (proteintech, 14833-A-AP, for immunofluorescence only), anti-Nestin (Cell Signaling, 33475S), ChIP–seq spike-in antibody (Active Motif, 61686), donkey anti-rabbit Alexa Fluor 555 cross-absorbed secondary antibody (Thermo Fisher, A31572), anti-rabbit Alexa Fluor 647 conjugate secondary antibody (Cell Signaling, 4414S), anti-Vimentin (Cell Signaling, 5741), anti-RNA pol II CTD phospho Ser2 antibody (Abcam, ab237280) and anti-RNA poly II CTD phospho Ser5 antibody (Active Motif, 61085).

### Plasmids and cloning

Recombinant hNDF expressing vectors pET21b-His6-hNDF and pET21b-hNDF-FLAG for bacterial expression were cloned as described previously[10,11]. The pET21b-His6-hNDF-mEGFP vector was generated by inserting an EGFP coding DNA fragment into the pET21b-His6-hNDF vector using the Gibson Assembly cloning kit (NEB). Expression vectors for NDF truncates and mutants were generated with a Q5 site-directed mutagenesis kit (NEB), using the following primers: for NDF-K161A mutants, primers NDF-K-For (5′-AGAGCCCAAGAGCAAAGTCCCCGG-3′) and K-Rev (5′-TGCCAGAGGGGATTTGGAGCCTCT-3′) were used; for NDF-R162A mutants, primers R-For (5′-GCAGCCCAAGAGCAAAGTCCCCGG-3′) and R-Rev (5′-TTTCAGAGGGGATTTGGAGCCTCT-3′) were used; for NDF-KR mutants, primers KR-For (5′-ATCCCCTCTGGCTGCAGCCCAAGAGC-3′) and KR-Rev (5′-TTGGAGCCTCTCTCTGAAG-3′) were used. The sf9 insect cells expression vectors for recombinant hFACT proteins were constructed by cloning hSSRP1 and SUPT16H/hSpt16 cDNAs into pFastBac vectors. N-terminal His6-, His6-GFP- or His6-mCherry-tags were inserted in-frame into pFastBac-FLAG-hSpt16 by using the NEBuilder HiFi DNA Assembly Master Mix (NEB) to give pFastBac-His6-mEGFP-hSpt16 and pFastBac-His6-mCherry-hSpt16. For yeast FACT complex expression vector, ySpt16 and yPob3 cDNAs were PCR amplified from purified *S. cerevisiae* genomic DNA and inserted into a dual-expression pET vector to obtain pET21b-His6-ySpt16-Flag-yPob3. For yeast NHP6 expression vector, NHP6 cDNA was amplified from purified *S. cerevisiae* genomic DNA and cloned into a pET21b vector with an N-terminal His6 tag and C-terminal mEGFP tag to produce pET21b-His6-yNHP6-mEGFP. For yeast DSIF complex expression vector, ySpt5 and ySpt4 cDNAs were PCR amplified from purified *S. cerevisiae* genomic DNA and inserted into a dual-expression pET vector to obtain pET21b-His6-ySpt5-Flag-ySpt4.

For yeast TFIIS expression vector, full-length yTFIIS cDNA was produced as a gBlock (IDT) and cloned into a pET21b vector with an N-terminal His6 tag to produce pET21b-His6-yTFIIS. All expression constructs were sequenced for verification. CRISPR guide RNA (gRNA) sequences for targeting the hNDF (5′-TATGTAGGCTCGGTACACGG-3′) or hSpt16 (5′-TCGGGCCGTGGCTCTAACCG-3′) loci were inserted into the PX459-Cas9 vector (Addgene, 62988). Homologous recombination donor DNA templates were generated using two homology arms (~0.8 kb each), PCR amplified with Q5 DNA polymerase (NEB) from human iPS cell genomic DNA. The mini-IAA_degron-mEGFP fragment was PCR amplified from the plasmid pSH-EFIRES-B-Seipin-miniIAA7-mEGFP (Addgene, 129719). The mScarlet fragment was PCR amplified from pLBS-mScarlet (Addgene, 129337), and the fragments IRES and AFB2 were PCR amplified from plasmid pSH-EFIRES-P-AtAFB2-mCherry-weak NLS (Addgene, 129716), the V5 tag and nuclear localization signal sequence (5′-GTAAGCCTATCCCTAACCCTCTCCTCGGTCTCGATTCTACGGCCGCAGCCAAACGGGTCAAACTTGATTAG-3′) was synthesized by IDT. All components were assembled into the CP1024 vector using NEBuilder HiFi DNA Assembly Master Mix (NEB), as per the manufacturer's suggestions. gRNA sequences in the donor DNA templates were altered to create silent mutations using a Q5 mutagenesis kit (NEB). To generate NDF_K161A mutant human iPS cells, a 20-nt gRNA sequence (5′-CTCTTGGGCTCTTTTCAGAG-3′) was cloned into the PX458-Cas9-GFP vector. A single-strand DNA oligo (/AlT-R-HDR1/G*G*GGCCGACCCCGCTTCCGGGGACTTTGCTCTTGGGCTCTTGCCAGAGGGGATTTGGAGCCTCTCTCTGAAGAGCCTGAAGAC*A*C/AlT-R-HDR2/) synthesized by IDT was used as a donor DNA template.

### Generating knock-in cells

To generate the knock-in cell lines, 0.7 μg PX459-Cas9-gRNA, 0.2 μg pCE-mP53DD (Addgene, 41586) and 1.2 μg donor vectors were transfected into 100,000 human iPS cells using 2 μl Lipofectamine Stem Transfection reagent (Thermo Fisher) following the manufacturer's instructions. The non-homologous end-joining (NHEJ) inhibitor SCR7 (Sigma) was added at a final concentration of 1 μM to enhance homology-directed repair (HDR) efficiency. A 3 days post-transfection, cells expressing GFP or mScarlet were sorted by FACS into Matrigel-coated 96-well plates. Single-cell clones were maintained and expanded. Genotyping was performed using NDF primers, forward (5′-GCCAGAGACTGAGCCATCTC-3′) and reverse (5′-GTGGGCTGGTCCAGAATGAAC-3′) to identify hNDF locus knock-in clones, and hSpt16 primers, forward (5′-GCTTACCTATTGCTAACTAATAATGCC-3′) and reverse (5′-CCCCTAAACCCATAAACACAAATG-3′) for the hSpt16 locus. For the NDF and SUPT16H double knock-in cell lines, two separate experiments were carried out sequentially. Genome-edited iPS cells were confirmed for pluripotency with Oct4 staining and differentiation into neural progenitor cells. For the NDF-K161A knock-in, 3 μl of 10 μM ssDNA donor template was cotransfected with 0.7 μg PX458-Cas9-mcherry-gRNA vector and 0.15 μg pCE-mP53DD into 100,000 human iPS cells using 2 μl Lipofectamine Stem Transfection reagent (Invitrogen). At 2 days post-transfection, GFP-positive single cells were FACS sorted into 96-well plates. Genotyping used primers KR_F (5′-GACGTCATCCCACAATTCTTCT-3′) and R-R1 (5′-CGGGGACTTTGCTCTTGAGCT-3′). The NDF-K161A mutants produced a PCR product that, when digested with XcmI restriction enzyme (NEB), resulted in 181 bp fragments, unlike the PCR product from NDF_WT cells. All clones, either homozygous or heterozygous, were validated by western blot analysis and DNA sequencing (Azenta).

### Lentiviral infection

HeLa cells expressing hNDF−Flag−GFP were produced by cotransfecting HEK293T cells with pHR-IRES-puro-NDF-FLAG-GFP vector, psPAX2 and pMD2.G to generate lentivirus. At 48 h post-transfection, the media from HEK293T cells, containing the lentiviruses, was collected and

used to infect HeLa cells. Stable cells were established by selecting with 1 µg ml$^{-1}$ puromycin. For tandem affinity purification, HeLa cells stably expressing NDF–Flag–GFP were collected and lysed in NETN buffer (20 mM Tris–HCl, pH 8.0, 100 mM NaCl, 1 mM EDTA, 1 mM MgCl$_2$, 0.5% Nonidet P-40 and 1 µg ml$^{-1}$ each pepstatin A and aprotinin) for 30 min at 4 °C. The crude lysates were sonicated three times for 10 s at 4 °C to release chromatin-bound proteins, and cleared by centrifugation (15,000$g$, 30 min, 4 °C). The cleared lysate was incubated with Flag M2 agarose beads (Sigma) for 4 h at 4 °C with gentle agitation. Beads were washed four times with NETN buffer and bound proteins eluted by incubating with 150 ng µl$^{-1}$ Flag peptide (Sigma) in NETN buffer for 2 h at 4 °C. The eluate was then incubated with GFP-Trap beads (Chromotek) for 2 h at 4 °C with gentle agitation. After three washes with NETN buffer, bound proteins were eluted in Laemmli sample buffer, resolved by SDS–PAGE and analysed by mass spectrometry at the Taplin Mass Spectrometry Facility (Harvard Medical School).

## Western blot

Cells were lysed using 1× SDS sample buffer (50 mM Tris–HCl pH 6.8, 1% SDS, 8% glycerol, 0.02% bromophenol blue and 2% 2-mercaptoethanol). The lysates were briefly sonicated and denatured at 95 °C for 5 min. Proteins were then separated by SDS–PAGE. Following the hybridization with the primary antibodies (described above), the membrane was incubated with HRP-conjugated Protein A (Cell Signaling, 12291). The protein of interest was visualized using SuperSignal West Pico PLUS Chemiluminescent Substrate (Thermo Fisher) and imaged with a ChemiDoc Imaging System (Bio-Rad).

## Protein purification

Human NDF recombinant proteins were purified as previously described[10,11]. Recombinant *Drosophila melanogaster* core histones were synthesized in *Escherichia coli* and purified following established methods[57]. The *S. cerevisiae* RNA Pol II was purified as described[58]. Recombinant full-length yeast TFIIS was prepared as described[59]. Yeast Spt4 and Spt5 were co-expressed in Rosetta 2 (DE3) pLysS (Millipore Sigma) using a dual-expression vector. Protein expression was induced with 0.4 mM IPTG for 16 h at 19 °C. All subsequent steps were conducted at 4 °C. The bacterial pellets were resuspended in lysis buffer (50 mM Tris–HCl pH 7.5, 700 mM NaCl, 10 mM 2-mercaptoethanol, 10 mM imidazole, 5% glycerol, 0.2 mM PMSF and 1× protease inhibitor cocktail (Roche)) and sonicated on ice. Insoluble material was removed by centrifugation at 26,916$g$ for 30 min, and the supernatant was incubated with Ni-NTA agarose (Qiagen) for 4 h on a nutator. The bound proteins were washed twice with buffer A (50 mM Tris–HCl pH 7.5, 700 mM NaCl, 10 mM 2-mercaptoethanol, 25 mM imidazole and 5% glycerol) followed by two washes with buffer B (20 mM Tris–HCl pH 7.5, 150 mM KCl, 10 mM 2-mercaptoethanol, 25 mM imidazole, 5% glycerol and 1 µM ZnCl$_2$). The protein was eluted with buffer E1 (20 mM Tris–HCl pH 7.5, 150 mM KCl, 1 mM DTT, 250 mM imidazole, 5% glycerol and 1 µM ZnCl$_2$). The fractions containing the proteins were combined and passed through a Bio-Rad Econo-Pac chromatography column containing 500 µl Flag M2 agarose beads (Sigma) five times. The proteins were then washed twice with buffer C (20 mM Tris–HCl pH 7.5, 400 mM KCl, 1 mM DTT, 5% glycerol and 1 µM ZnCl$_2$), and twice with buffer E1 without imidazole. Finally, the protein was eluted eight times with 0.5 ml buffer E2 (20 mM Tris–HCl pH 7.5, 150 mM KCl, 1 mM DTT, 5% glycerol, 1 µM ZnCl$_2$ and 400 ng µl$^{-1}$ 3× Flag peptide (Sigma)). The fractions containing yDSIF protein were combined and stored at −80 °C. Recombinant human his-FACT subunits were expressed separately in Sf9 cells (Thermo Fisher) using the Bac-to-Bac Baculovirus Expression System, and purified similar to his-hNDF proteins. Both hNDF and hFACT proteins were stored in a buffer containing 10 mM Tris–HCl at pH 7.5, 0.2% (v/v) nonidet P-40, 0.2 M NaCl, 10% (v/v) glycerol and 5 mM 2-mercaptoethanol. For the Flag-tagged FACT complex, Flag-hSpt16 and untagged-hSSRP1 were co-expressed in Sf9 cells and

purified following previously described methods[14]. Cell pellets were washed with ice-cold PBS and lysed in FLAG-600 buffer (600 mM NaCl, 10 mM Tris–HCl at pH 7.5, 0.1% NP-40 (v/v), 15% (v/v) glycerol and 1 mM PMSF). The cell suspension was briefly sonicated and clarified by centrifugation at 26,916$g$ at 4 °C. The cleared cell extract was incubated with anti-Flag M2 agarose (Sigma) for 4 h at 4 °C, and the resin was subsequently washed with FLAG-600 buffer. Proteins bound to the resin were eluted using 400 ng µl$^{-1}$ 3× Flag peptide (Sigma) containing NDF buffer (10 mM Tris–HCl at pH 7.5, 0.2% (v/v) NP-40, 0.2 M NaCl, 10% (v/v) glycerol and 5 mM 2-mercaptoethanol). All proteins were quantified by SDS–PAGE, and a standard curve was established using purified BSA (Thermo Fisher). Recombinant his-yNHP6-GFP and his-ySpt16-Flag-yPob3 proteins were purified similar to his-hNDF proteins. his-ySpt16-Flag-yPob3 proteins were further purified with Flag M2 agarose beads, proteins finally eluted in the buffer containing 10 mM Tris–HCl pH 7.5, 200 mM NaCl, 5% glycerol, 1 mM 2-mercaptoethanol and 400 ng µl$^{-1}$ 3× Flag peptide (Sigma)).

## Nucleosome preparation

The Widom and PC 601 DNA used for the in vitro transcription elongation assay (as listed in Supplementary Table 1) were amplified using PCR and purified using the QiaQuick gel extraction kit (Qiagen). The purified dsDNA was subsequently digested with the TspRI restriction enzyme (NEB). The resulting DNA fragment was gel purified and reconstituted into nucleosomes using the sequential salt dialysis method, as described previously[57]. For the Cy2-labelled Widom 601 nucleosome, Cy2-labelled 601 DNA was generated by amplifying the 601 Widom nucleosome positioning sequence with a Cy2-labelled primer at the 5′ end.

## Transcription elongation assays

The transcription elongation assays were performed with slight modifications to previously described methods[22]. In our in vitro transcription elongation assays, the transcripts generated were sensitive to RNase A but not RNase H, indicating that the nascent RNA did not form a DNA–RNA hybrid during Pol II elongation, which can sometimes occur with purified Pol II. Initially, a 100 nM Widom 601 nucleosomal template was incubated with equimolar amounts of a 6-FAM 5′-labelled 11-nt RNA (5′-/56-FAM/ rUrUrA rUrCrA rCrUrG rUrC-3′) at 30 °C for 10 min to facilitate RNA annealing to the nucleosomal template. RNA Pol II (120 nM) was then added and incubated for an additional 10 min at 30 °C. Each incubation step was balanced with compensation buffer and water to maintain the final buffer composition: 130 mM NaCl, 20 mM Na–HEPES (pH 7.4), 3 mM MgCl$_2$, 4% (v/v) glycerol and 1 mM DTT. Transcription elongation was initiated by adding ATP, CTP, GTP and UTP (1 mM each), along with TFIIS (100 nM), and continued for 10 min at 30 °C unless otherwise specified. The reaction was terminated by adding an equal volume of gel loading buffer (1 ml of gel loading buffer contains 0.9 ml of deionized formamide, 0.1 ml of 0.5 M EDTA and 2 µl of 4% (w/v) bromophenol blue). Samples were denatured at 95 °C for 10 min with the cap open and immediately chilled on ice. Subsequently, the samples were separated by denaturing acrylamide gel electrophoresis. Gel visualization was performed using the 6-FAM label with an Amersham Typhoon at an excitation wavelength of 490 nm and an emission wavelength of 510 nm. All transcription elongation assays were conducted independently and repeated at least three times. The full length of transcripts was quantified using ImageJ software, with products normalized against the total intensity of the respective reaction lane to correct potential errors during gel loading. Enzyme kinetic parameters were determined using the Michaelis–Menten model. While our nucleosome transcription assay does not strictly follow simple enzyme kinetics, this approach allowed us to calculate comparable efficiency ratios between conditions. The goodness of fit for each kinetic model was evaluated using the coefficient of determination ($R^2$). Nonlinear regression analysis was performed using GraphPad

Prism 9. The Michaelis−Menten equation ($v = V_{max}[S]/(K_m + [S])$) was fit to the initial velocity data using the least squares method. $V_{max}$ and $K_m$ were estimated for each condition, along with their standard errors and 95% confidence intervals. The specificity constant (analogous to $k_{cat}/K_m$) was calculated using Monte Carlo simulation methods using a uniform distribution sampling method to account for uncertainties in $V_{max}$ and $K_m$ estimates. Fold changes between conditions were determined by comparing these ratios. For the transcription products sedimentation assay, each reaction was performed in a final volume of 20 µl. After 10 min chasing with 1 mM rNTP, the transcription products were immediately put on ice for 5 min, followed by centrifugation at 17,200g for 5 min at 4 °C. The pellet was resuspended with 20 µl gel loading buffer and supplemented with an additional 20 µl transcription reaction buffer before the denaturing urea gel analysis. For the RNase A protection assays, RNase A (NEB) was added to the transcription products at final concentrations of 0, 0.02 or 0.2 ng µl$^{-1}$, and samples were incubated at 30 °C for 3 min. Subsequently, samples were processed using the same procedure as described for the transcription products sedimentation assay.

## Immunoprecipitation

For co-immunoprecipitation, approximately 10 million cells were lysed in 1 ml of RIPA buffer (25 mM Tris−HCl pH 7.6, 150 mM NaCl, 1% Nonidet P-40, 1% sodium deoxycholate, 0.1% SDS and Protease Inhibitor Cocktail (Roche)). The lysates were sonicated twice for 10 s and then centrifuged for 10 min (26,916g, 4 °C) to remove cell debris. Then, 500 µl of the supernatant was incubated with 20 µl of Protein A agarose (Thermo Fisher) precoated with 4 µl of hNDF antiserum. The mixture was incubated at 4 °C for 3 h on a nutator, then the beads were captured by centrifugation for 1 min at 1,000g and washed five times with 1 ml of cold RIPA buffer. The proteins bound to the beads were eluted in SDS−PAGE buffer and analysed by western blot. For the in vitro pull-down assay, unless specified, 40 pmol of His6-tagged NDF-WT, truncates or mutants were incubated with 20 pmol of Flag-tagged hSpt16 or FACT proteins and 40 µl beads (Ni-NTA beads for his-tagged bait protein, or Flag M2 beads for Flag-tagged bait protein) in 300 µl of pull-down buffer (10 mM Tris−HCl (pH 7.5), 200 mM NaCl, 0.2% Nonidet P-40, 10% glycerol and Protease Inhibitor Cocktail (Roche)) at 4 °C for 3 h. The beads were then washed three times with the same pull-down buffer, following the procedure used for the co-immunoprecipitation. Proteins were eluted with SDS−PAGE buffer and visualized by SDS−PAGE and silver staining.

## In vitro droplet and sedimentation assay

Recombinant GFP or mCherry fusion proteins were first diluted and equilibrated in a specific salt concentration by adding NaCl or droplet formation buffer (10 mM Tris−HCl pH 7.5, 5% glycerol and 0.1% NP-40), followed by a 10-min 17,200g centrifugation at 4 °C to remove any insoluble proteins formed during the freeze−thaw process. The proteins were then mixed at appropriate concentrations and promptly loaded into a custom-made chamber, which consisted of a glass slide sealed with a coverslip using two parallel strips of double-sided tape. For droplet formation experiments in alternative buffers, the indicated protein amounts were mixed to achieve the final buffer concentration. Alternative buffers included transcription buffer (20 mM Na−HEPES, pH 7.4, 130 mM NaCl, 3 mM MgCl$_2$, 4% (v/v) glycerol, 1 mM DTT and 1 mM rNTPs) and pull-down/immunoprecipitation buffer (10 mM Tris−HCl, pH 7.5, 0.2 M NaCl, 0.2% NP-40, 10% glycerol and 1 mM β-mercaptoethanol). The samples were then loaded into imaging chambers and visualized. Imaging was performed using a Zeiss AXIO epifluorescence microscope or Zeiss LSM800 confocal microscope with a 63× oil objective. Images typically show droplets settled on the glass coverslip under 100 mM NaCl conditions. For the sedimentation assay, each reaction had a typical final volume of 100 µl. After a 10-min equilibration at room temperature (22 °C), protein samples were centrifuged at 17,200g for

10 min at 22 °C using a tabletop temperature-controlled microcentrifuge. Post-centrifugation, the supernatant and pellet were promptly separated into two tubes. The pellet was thoroughly resuspended in the same buffer to match the volume of the supernatant. Proteins from both fractions were denatured and analysed on 8% SDS−PAGE stained with Coomassie blue. Band intensities were quantified using ImageJ software. To assess condensate formation at different temperatures, proteins were mixed at a final concentration of 1 µM and incubated for 10 min at 4 °C, 25 °C and 37 °C. Samples were then loaded onto CELLview cell culture dishes and imaged using an internal reflection fluorescence microscope (DMi8 TIRF, Leica) at corresponding chamber temperatures of 25 °C (for both 4 °C and 25 °C pre-incubated samples) and 37 °C (for the 37 °C sample).

## FRAP in live cells

FRAP experiments were conducted using a total internal reflection fluorescence microscope (DMi8 TIRF, Leica) equipped with an Infinity Scanner system (Leica) at room temperature. Images were captured using a 100× oil objective at a rate of 1 Hz. A circular region with a radius of 1.5 µm was photobleached at the centre of the puncta using a brief pulse (-1 s) of a focused 488 nm laser. Subsequently, fluorescence recovery was monitored every 2 s for the specified duration, and the data were analysed using Fiji 2.1.0 software. To minimize the impact of photobleaching and boundary effects, fluorescence of the bleached region ($I_{ROI}$) was normalized by the fluorescence of the entire puncta ($I_{punc}$) using the following equation, following background subtraction:

$$I(t) = \frac{I_{ROI}(t) - I_{background}(t)}{I_{punc}(t) - I_{background}(t)}$$

The timepoint immediately following the bleaching event was designated as time zero. The fluorescence intensity ($I(t)$) was normalized such that the average intensity at $I(t < 0)$ equals 1.

## Cell treatments

For Aux-inducible degron experiments, cells with degron tags were treated with 0.5 mM 3-indoleacetic acid dissolved in DMSO (IAA) (Sigma) or 0.1% (v/v) DMSO in the culture media for the specified duration before collecting for analysis. For ChIP−seq experiments involving 1,6-Hex treatment, cells were treated with 1.5% 1,6-hexanediol for 10 min before crosslinking. For imaging studies, cells were treated with a 3% 1,6-Hex solution for 10 min before being fixed.

## Cell cycle analysis

NDF−degron iPS cells were treated with DMSO or Aux for 2 h and 48 h. After treatment, cells were dissociated with 1× TrypLE Select Enzyme (Thermo Fisher, 12563029). Two million cells were washed with PBS and and resuspended in 300 µl of cold (4 °C) PBS. Cells were fixed by the dropwise addition of 0.8 ml of cold (4 °C) ethanol followed by incubation at 4 °C for a minimum of 24 h. After fixation, the cells were pelleted and resuspended in 1 ml of PBS containing 50 µg ml$^{-1}$ of propidium iodide (Sigma, P4170) and 10 µg ml$^{-1}$ of RNase A (NEB, T3018). The cells were incubated for 30 min at 22 °C and then subjected to flow cytometry analysis.

## Cell apoptosis analysis

NDF−degron iPS cells were treated with DMSO or Aux for 48 h. After treatment, the cells were dissociated using 1× TrypLE Select Enzyme (Thermo Fisher, 12563029), washed with cold 1× PBS and stained with Annexin V Conjugate (Thermo Fisher, A23202) according to the manufacturer's instructions. The cells were then subjected to flow cytometry analysis.

## RNA-FISH

Cells were seeded on Matrigel-coated coverslips and cultured for 24 h before the experiment. Cells were then fixed with 4% paraformaldehyde

(PFA) (Fisher) in PBS for 10 min at room temperature. This was followed by two PBS washes and a permeabilization step using 0.3% Triton X-100 (Sigma-Aldrich) in PBS for 5 min, then washed three times with PBS. Next, cells were stained with NDF antibody and Alexa 555 secondary antibody. After protein staining, cells were fixed again with 4% PFA and continued with RNA-FISH staining. Cells were first incubated in wash buffer A, containing 20% Stellaris RNA-FISH wash buffer A (Biosearch Technologies) and 10% deionized formamide (EMD Millipore) in RNase-free water (Life Technologies), for 5 min. Then, 100 µl of a 12.5 µM human ACTB probe (Biosearch Technologies, ISMF-2003-5) in hybridization buffer, consisting of 90% Stellaris RNA-FISH hybridization buffer (Biosearch Technologies) and 10% deionized formamide, was added to the cells and incubated overnight at 37 °C. Post-hybridization, cells were washed for 30 min with wash buffer A at 37 °C, followed by a nuclear stain of 5 ng ml$^{-1}$ 4,6-diamidino-2-phenylindole (DAPI) (Bio-Rad) in wash buffer A for 30 min at 37 °C, and a final 5-min wash in wash buffer B (Biosearch Technologies) at room temperature. Coverslips were then mounted on glass slides using Vectashield mounting medium (Vector Laboratories), sealed with nail polish and imaged using a Zeiss LSM800 confocal microscope with a 63× oil objective and Airyscan module. Image processing and export were conducted using ZEN imaging software.

## Immunofluorescence

Cells were seeded on Matrigel-coated CELLview cell culture dishes (Greiner) for 24 h before the experiment. For native condition immunofluorescence, cells were washed with 1× PBS and fixed with 4% PFA for 10 min at 21 °C. Permeabilization was performed with 0.3% Triton X-100 for 5 min at room temperature. Cells were blocked with 2% goat serum for Oct4 immunofluorescence or 2% donkey serum for NDF/GLYR1 immunofluorescence in 1× PBS (blocking buffer) for 30 min at room temperature. The cells were then incubated with a 1:500 dilution of Oct4 antibody (Cell Signaling, 2750S) or 1:100 dilution of NDF/GLYR1 antibody (Proteintech, 14833-1-AP) in the blocking buffer at room temperature for 1 h. Following this, the cells were washed three times with 1× PBS + 0.1% Tween-20 for 5 min each and incubated with a 1:500 dilution of goat anti-rabbit IgG (H + L), F(ab')¬2 Fragment (Alexa Fluor 647 conjugate) (Cell Signaling, 4414S) for Oct4 or Donkey anti-Rabbit IgG (H + L) Highly Cross-Adsorbed Secondary Antibody, Alexa Fluor 555 (Thermo Fisher, A31572) in the blocking buffer for 1 h at room temperature. The microscope was specifically configured and tested to ensure no signal bleed-through between fluorescence channels. After three additional washes with 1× PBS + 0.1% Tween-20, nuclei were counterstained with 1 µg ml$^{-1}$ DAPI in PBS for 5 min. After a brief wash with 1× PBS, cells were imaged using a Zeiss LSM800 confocal microscope with a 63× oil objective and Airyscan module. Image processing and export were performed using ZEN imaging software. To image chromatin-bound proteins, cultured cells were washed with 1× PBS and treated with a CSK buffer[35] (10 mM Pipes, pH 7.0, 100 mM NaCl, 300 mM sucrose and 3 mM MgCl$_2$) containing 0.2% Triton X-100 at 21 °C for 1 min to remove non-chromatin-bound proteins. Following the treatment, cells were quickly washed with 1× PBS and fixed with 4% PFA for 10 min at 21 °C. After staining the nuclei with DAPI and a brief wash with PBS, cells were imaged. For computational image enhancement, epifluorescence images were enhanced using DeconvolutionLab2 deconvolution software with a synthetic point spread function generated from our 40× air objective specifications. Following deconvolution, a 50-pixel rolling ball background subtraction was applied to correct uneven illumination. Enhanced images were processed and analysed using Fiji software to improve visualization of nucleosome recruitment dynamics.

## STORM imaging

NDF−FACT condensates containing mCherry-labelled proteins were assembled using equimolar concentrations (0.4 µM each)

in condensate formation buffer and combined with 0.1 µM Pol II elongation complexes incorporating 6-FAM-labelled RNA primers, followed by NTP addition to initiate transcription. The presence of condensates was confirmed by brightfield microscopy, and mCherry-positive condensates were identified by epifluorescence microscopy to establish regions of interest. Super-resolution imaging was performed using a custom STORM system based on an Olympus IX83 frame equipped with a spinning disk confocal unit (DSU) and 40× UPlanSApo objective (NA 1.3). The spinning disk pinhole was employed to block out-of-focus fluorescence. Single-molecule fluorescence from 6-FAM-labelled RNA was acquired at 50 Hz using the Andor Zyla 4.2 sCMOS camera as 512 × 512 pixel images at 50 Hz with a 10−20 ms exposure time for each frame. Approximately 200 sequential frames per field were acquired before 6-FAM photobleaching occurred. Single-molecule localization and super-resolution reconstruction from the raw image sequences were performed using the ThunderSTORM plugin in Fiji, and $XY$ drift correction was applied using cross-correlation image analysis. Condensate boundaries were identified, and centroids from the RNA signal were determined. Spatial distances were quantified by measuring centre-to-centre distances between condensates and RNA. Measured distances were compared to assess localization patterns.

## Puncta calling

The number of the puncta in cells was counted using the '3D Objects Counter' plugin in FIJI. All images were acquired under identical confocal settings (for example, laser power and exposure time) to maintain consistency, and nuclei of similar size were selected and cropped from raw images. To address cell-to-cell variation, we normalized the total fluorescence of each nucleus to a common reference, an approach supported by western blot data demonstrating stable protein levels across samples. After normalization, each image was converted to 16 bit and subjected to a uniform histogram-based threshold, highlighting the brightest signals (defined as puncta) without artificially creating signals (confirmed by DAPI-only controls). This analysis revealed a treatment-specific leftward shift in the upper tail of the intensity distribution, indicating reduced bright puncta intensity rather than a global decrease in fluorescence. In addition, puncta counts were normalized to nuclear area (for example, puncta per µm$^2$) to correct for differences in cell size or focal plane. By combining consistent acquisition parameters, total fluorescence normalization, threshold verification and area-based adjustments, we ensured that our quantification pipeline captures both bright and moderate signals while minimizing technical artefacts and providing a reliable measure of condensate formation. The 'Min number of voxels' parameter was set to at least 10. For the percentage of puncta intensity calculation, the total intensity of puncta in each cell was calculated using '3D Objects Counter'. The total intensity of the fluorescence in each cell was measured by Fiji[60]. The percentage of the intensity of the puncta was calculated as the total intensity of the puncta divided by the total intensity of the fluorescence in the corresponding cell. Colocalization in Puncata was quantified using the Coloc 2 plugin in Fiji/ImageJ to determine the Pearson correlation coefficient.

## Mechanical unwrapping assay of a single nucleosome

The mechanical unwrapping assay of a single nucleosome was conducted following the method described before[34], utilizing a custom-built dual-trap optical tweezers instrument[61]. Briefly, individual nucleosomes were attached to two 1 µm polystyrene beads via DNA handles of 570 bp and 700 bp. Nucleosomes were ligated to the DNA handles using *E. coli* DNA ligase (NEB) at 16 °C for 2 h in the presence of 0.02% NP-40. A subsequent ligation was then carried out to attach the handle−nucleosome−handle complex to oligo beads. The latter beads were prepared by conjugating carboxyl-functionalized polystyrene beads (Bangs Labs) to double-stranded oligonucleotides (IDT) with

a 5′ NH-ester end and a 4 nt sticky end for handle ligation. The 700 bp DNA handle was biotinylated at one end to allow for attachment to a streptavidin-coated bead, enabling precise control and manipulation within the optical tweezers. Before the assay, the tweezing chamber and microfluidic channels were passivated twice, first with 1 mg ml$^{-1}$ BSA and then with 0.25% Pluronic F-127, both diluted in 20 mM Tris (pH 7.5) and 50 mM KCl. The unwrapping assay was conducted at room temperature in the pulling buffer (20 mM Tris pH 7.5, 50 mM KCl, 100 mM NaCl, 5 mM MgCl$_2$, 10 mM NaN$_3$, 0.1 mg ml$^{-1}$ BSA and 1.0 mM DTT). The NDF and FACT proteins were introduced into the main channel at a final concentration of 50 nM. In addition, nucleosome samples were pre-incubated with 50 nM FACT or 50 nM NDF protein at room temperature for 10 min, followed by a tweezing period of 1 h. To prevent nucleosome disassembly in solution, 0.02% NP-40 was added in the diluted nucleosome sample. Mechanical unwrapping and rewrapping of nucleosomes were carried out at a constant pulling rate of 20 nm s$^{-1}$. The force–extension trajectories were continuously monitored during the experiment, allowing the detections of the low-force and high-force transitions during nucleosome unwrapping and the assignment of the subnucleosomal particles, according to their different force–extension features[34]. For repetitive pulling and relaxing cycles, subsequent pulling was initiated only after the nucleosome, hexasome or tetrasome had fully rewrapped, as indicated by the trapping force returning to approximately 0 pN. The trap stiffness during the experiment ranged between 0.28 and 0.4 pN nm$^{-1}$. Raw data were collected at a sampling rate of 2.5 kHz, and subsequent data processing and analysis were conducted using a custom-written MATLAB (v25.1) code following the methodology outlined by Diaz-Celis et al.[34].

### Optical tweezers assay of Pol II transcription through the nucleosome

The assembly of Pol II stalled complex was performed as described before[33]. Briefly, the bubble initiation method followed by uridine triphosphate (UTP) starvation was applied to prepare the stalled elongation complex using a biotinylated yeast Pol II. The stalled complex was further ligated to 2 kb upstream DNA preligated to 1 μm polystyrene oligo beads using *E. coli* DNA ligase (NEB). Concurrently with the upstream ligation, the stalled complex was ligated downstream to the molecular ruler made up of eight repeats of a sequence containing a well-defined pause site for Pol II (845 bp) followed by the human nucleosome loaded on the 601 nucleosome positioning sequence (NPS). The resulting Pol II–nucleosome beads were diluted with 1 ml of transcription buffer (TB50: 20 mM Tris pH 8.0, 50 mM KCl, 100 mM NaCl, 5 mM MgCl$_2$, 1 mM DTT, 10 mM NaN$_3$ and 0.1 mg ml$^{-1}$ BSA) to a final bead density of 0.00003%. On the other hand, the 1.5 kb biotin-DNA handles were ligated to 1 μm polystyrene oligo beads and pre-incubated with 0.5 μM streptavidin for 10 min at room temperature and diluted with 1 ml of TB50 buffer to a final bead density of 0.00008%. Prior adding the beads, the tweezing chamber was passivated with 0.5% Pluronic F-127 (Invitrogen) and 1 mg ml$^{-1}$ BSA (NEB) and washed with TB50 buffer. The optical tweezers experiments were performed in a custom-made dual-trap optical tweezers instrument[33]. The 1.5 kb biotin–DNA handle beads and Pol II–nucleosome beads were captured in each optical trap. A single tether was formed by bringing both beads into close proximity, and force feedback was applied to maintain a constant force of 10 pN. Under these conditions, transcription was restarted by flowing 0.5 mM of each NTP (Fisher Scientific) into the chamber, either in the absence or the presence of 10 nM FACT with or without 50 nM NDF. Data acquisition was recorded at 800 Hz and terminated once the polymerase reached the end of the template or arrested for more than 600 s without dynamics. The alignment of the transcriptional traces, data analysis on residence time of Pol II along the nucleosome and preparation of the transcriptional maps were conducted following the methodology described before using a custom-written MATLAB code[33].

### Optical tweezers with confocal scanning assay of Pol II transcription through the nucleosome

Simultaneous force and fluorescence measurements were performed on a C-trap (Lumicks). The five-channel laminar flow cell was passivated with 0.5% Pluronic F-127 and 1 mg ml$^{-1}$ BSA, and subsequently washed with TB50 buffer. The Pol II stalled elongation complex was prepared as described above and ligated upstream to 3.5 kb (Fig. 3e) or 5.5 kb (Fig. 3f) DNA handles that had been preligated to 1 μm oligo beads, and downstream to the molecular ruler (845 bp; Fig. 3e) or a longer DNA sequence (3.5 kb; Fig. 3f), followed by the human nucleosome. The resulting Pol II–nucleosome beads were diluted with 1 ml of TB50 and flowed into channel 4 (Extended Data Fig. 3b). On the other side, the 3.5 kb biotin-2xCy5-DNA handles were prepared via PCR using lambda DNA as the template and biotin-2xCy5-labelled and BsaI-containing oligonucleotides (IDT) as primers. The PCR products were subjected to BsaI digestion, gel purification and ligation to 1 μm polystyrene oligo beads. The 3.5 kb biotin-2xCy5-handle beads were pre-incubated with 0.5 μM streptavidin, diluted with 1 ml of TB50 buffer and added into channel 1. The NTP was diluted in TB50 buffer to a final concentration of 0.5 mM in the presence of transcriptional regulators alone (10 nM FACT–mCherry or 10 nM NDF–GFP) or in combination, and then flowed into channel 4. Channels 2 and 3 were filled with TB50 buffer (Extended Data Fig. 3b). After catching the beads without flow (that is, biotin–handle bead in channel 1 on trap 2 and Pol II–nucleosome bead in channel 4 on trap 1), the trapped beads were moved to channel 2 for force calibration. Experiments were done at trap stiffness of 0.15 pN nm$^{-1}$. A single tether was formed and verified by measuring a force-extension curve, with the force then clamped to 10 pN. Subsequently, confocal scanning was started, and the tether and beads were moved to the NTP/protein channel (channel 5) to resume transcription. Continuous two-dimensional confocal scans were conducted to obtain a complete image of the tether and beads. The images were acquired using 488, 532 and 639 nm lasers (GFP, mCherry and Cy5 excitation, respectively) at a laser power of 5%. Fluorescence emission was recorded using blue (512/25 nm), green (582/75 nm) and red (680/42 nm) filters. All experiments were performed at room temperature. Raw data exported from Lumicks Bluelake as .h5 files were converted to .mat files to be analysed with custom-written MATLAB code, as described before[33]. The latter was used to align the transcription traces, plot the transcribed distance in bp and define the position of Pol II along the template. The kymographs were generated from confocal 2D scans by using Fiji. Additional adjustments of the confocal 2D scans were performed in Fiji. The percentage of bleed-through of GFP from blue to green was determined by measuring the photon counts intensity on both blue ($I_{b\text{-GFP}}$) and green ($I_{g\text{-GFP}}$) channels obtained when NDF–GFP was added in the absence of FACT–mCherry. For this, a custom-written Jupyter Notebook in Python 3.9 was used. Subsequently, the correction factor ($a$) was calculated by dividing $I_{g\text{-GFP}}/I_{b\text{-GFP}}$, and used to correct mCherry signal obtained when both factors were added to the transcription assay, as follow: $I_{mCherry\ corrected} = I_g - (I_b \times a)$.

### ChIP–seq

ChIP–seq cell crosslinking was performed as described previously[10]. Formaldehyde crosslinked cells were then treated for 5 min on ice with cell lysis buffer (5 mM HEPES, K+ pH 8.0, 85 mM KCl, 0.5% (v/v) Nonidet P-40 and Protease Inhibitor Cocktail (Roche)). After treatment, cell pellets were resuspended in a Nuclear Lysis Buffer (50 mM Tris–HCl pH 8.0, 10 mM EDTA, 0.2% (w/v) SDS and Protease Inhibitor Cocktail (Roche)) and sonicated with a Covaris S220 sonicator. Chromatin concentrations were estimated using a Nanodrop spectrophotometer (Thermo Fisher) and diluted to a final concentration of 0.5 mg ml$^{-1}$ with ChIP dilution buffer (16.7 mM Tris–HCl pH 8.0, 167 mM NaCl, 1.2 mM EDTA, 0.01% (w/v) SDS, 1.1% (v/v) Triton X-100 and Protease Inhibitor Cocktail (Roche)). Then, 80% glycerol was added to the chromatin sample to make it 0.5 mg ml$^{-1}$ with 10% glycerol. For immunoprecipitation,

0.15 mg chromatin was added to 20 µl of Dynabeads Protein A (Thermo Fisher) pre-bound with specific antibodies (1 µl hNDF antiserum for NDF ChIP, 8 µl hSpt16 antibody for hSpt16 ChIP, 4 µl RPB II antibody for total Pol II ChIP and 2 µg LEGDF antibody for LEDGF ChIP). For ChIP–seq spike-in chromatin normalization, 1 µl spike-in antibody and 2.5 µl spike-in chromatin (Active Motif) were added to the sample. After incubation at 4 °C for 4 h on a nutator, beads were captured with a magnetic stand and washed as described[10]. Chromatin bound on the beads was eluted twice with 50 µl of TE (10 mM Tris–HCl pH 8 and 1 mM EDTA) containing 1% (w/v) SDS for 20 min at 20 °C. ChIP–seq libraries were prepared with the NEBNext Ultra II DNA Library Prep kit for Illumina (NEB) according to the kit instructions. Final libraries were quantified by a Qubit 4 fluorometer and sequenced with an Illumina Novaseq X sequencer (Admera). The sequencing data were aligned to the hg19 reference genome using Bowtie2 (v2.5.3)[62]. Spike-in reads were mapped to the dm3 genome build. Normalized bigwig files and metagene analysis were prepared by Deeptools (v3.5.4)[63]. Spike-in normalizations were performed as suggested by the manufacturer. GC content was analysed for each gene from the TSS to TTS across different gene clusters. Nucleosome-depleted regions (NDRs) were identified from MNase-seq data using DANPOS2 (v2.2.2)[64]. Gaps between well-positioned nucleosomes are defined as NDRs. NDR length was calculated as the distance between the upstream and downstream boundaries of the depleted region. Only NDRs with a length less than 500 bp and located in the promoter region (TSS ± 1 kb) were retained for analysis. NDR length distributions were compared across gene clusters and visualized with ggplot. Pausing indices were calculated as the ratio of Pol II ChIP–seq read density within the promoter region (±300 nt around the TSS) to the read density in the gene body (from 500 nt downstream of the TSS to the TES) across gene clusters longer than 1,000 bp.

## Cut&Run

Cut&Run libraries were prepared using the CUT&RUN assay kit (Cell Signaling Technology). A total of 100,000 live cells were used per sample, and spike-in DNA was incorporated according to the manufacturer's protocol. For NDF Cut&Run, 0.5 µl hNDF antiserum was used for each reaction. The libraries were sequenced and analysed following the procedures outlined for ChIP–seq. Spike-in reads were mapped to the sacCer3 genome build.

## PRO-seq

PRO-seq with two biotin-NTPs was performed as described[65]. Cells were permeabilized following the protocol, and aliquoted with 1 million cells in 52 µl freeze buffer. To make the fragments not too small, the base hydrolysis time reduce from 10 min to 8 min. The libraries were sequenced and analysed following the procedures outlined for ChIP–seq.

## RNA-seq

Total RNA was purified from cultured cells by TRIzol Reagent (Thermo Fisher) following the manufacturers' instructions. RNA was treated with DNaseI (NEB) for 10 min at 22 °C to remove residual genomic DNA. Then, 2 µl of 100-times diluted ERCC RNA spike-in Mix 1 (Thermo Fisher) was added to 1 µg of purified total RNA. mRNA was isolated with the NEBNext Poly(A) mRNA Magnetic Isolation Module (NEB). mRNA libraries were prepared using the NEBNext Ultra II RNA library prep kit for Illumina (NEB). The libraries were sequenced with an Illumina Novaseq X sequencer (Aderma). The raw sequencing data were trimmed and aligned to the hg19 reference genome and the ERCC spike-in RNA annotation using Hisat2 (v2.2.1)[66]. Total reads from each gene were calculated using featureCounts (v2.1.1)[67]. The standard curve for normalization was calculated following a detailed published protocol[68]. Statistics analysis was performed using JMP Pro 18 (JMP Statistical Discovery). Differential gene expression analysis was performed

using DESeq2 (v2.11.40.8), and the results were visualized using the EnhancedVolcano (v1.20.0)[73] package[69].

## TT-seq

TT-seq was performed as previously described[12,70] with slight modifications. For each biological replicate, one 15-cm dish of human iPS cells at approximately 80% confluency was used. For TT-seq, cells were labelled with 1 mM 4-thiouridine (4sU; Sigma) for 10 min. For DRB/TT-seq, cells were pretreated with 0.1 mM DRB (TCI) for 3 h. For the DRB release 0-min condition, 4sU was added to the culture medium 10 min before the end of the 3-h DRB treatment, without removing DRB; for the DRB release 10-min condition, cells were rapidly washed twice with warm PBS to remove DRB, then incubated in 4sU-containing medium for 10 min; for the DRB release 20-min condition, after DRB removal (via two washes with warm PBS), cells were first incubated in DRB-free medium for 10 min, followed by incubation in 4sU-containing medium for an additional 10 min. After labelling, cells were quickly washed with ice-cold 1× PBS, and total RNA was extracted using standard TRIzol RNA extraction. A total of 100 µg of RNA was mixed with 0.5 µg of *S. pombe* 4TU-labelled RNA as a spike-in control. This mixture was then subjected to RNA fragmentation, biotinylation and streptavidin pull-down as previously described[70]. RNA libraries were prepared using the Illumina Stranded Total RNA Prep with Ligation and Ribo-Zero Plus kit, and sequenced on an Illumina NovaSeq X system (Aderma). Raw sequencing reads were trimmed and aligned to both the human reference genome (hg38) and *S. pombe* using STAR (v2.7.11b). PCR duplicates were removed using samtools (v1.15.1). Mapped BAM files were used to analyse transcriptional wave peaks and RNA Pol II elongation speed, following previously described protocols[70] with minor modifications. Analyses were restricted to genes longer than 50 kb with non-overlapping transcription units ($n = 2,558$ transcripts in hg38). Gene regions were extended from −2 kb upstream to +80 kb downstream of the TSS; any extensions beyond chromosome boundaries were excluded.

## MNase-seq

We performed MNase-seq with 1 million iPS cells per biological replicate. The digestion time was optimized beforehand to yield ~80% mononucleosomes and ~20% dinucleosomes. The 140–180 bp fragments corresponding to mononucleosomes were selected with 6% native gel for library construction using NEBNext Ultra II DNA Library prep kit for Illumina (NEB) according to the kit instructions. Final libraries were quantified by a Qubit 4 fluorometer and sequenced with an Illumina Novaseq X sequencer (Admera). The sequencing data were trimmed as described above and aligned to the hg19 reference genome using Bowtie2 (ref. 62). For Figs. 6 and 7, we normalized to the total reads of 2,437 transcriptionally inactive genes in cluster 4 (Fig. 6b), assuming that nucleosome occupancy near TSS in these genes should not be affected by NDF depletion. The transcriptionally inactive genes were defined by no total Pol II ChIP–seq reads on those genes, as counted by HOMER (v4.10)[71]. For Extended Data Fig. 9, we performed an independent normalization using *S. pombe* spike-in chromatin that was co-digested with the human chromatin samples, where reads were normalized to the total counts of *S. pombe* nucleosomes to control for digestion efficiency. For MNase-seq with yeast chromatin spike-in, *S. pombe* nuclei were prepared as previously described[72]. To optimize the spike-in ratio, varying amounts of *S. pombe* nuclei were digested under previously optimized conditions. The resulting mononucleosome fractions were purified to determine the amount of *S. pombe* nuclei required to constitute approximately 5% of the iPS cell mononucleosome fraction. Based on this determination, the appropriate amount of *S. pombe* nuclei was mixed with 1 million iPS cells and subjected to MNase digestion under the same optimized conditions. The spike-in reads were mapped to the *S. pombe* (GCF_000002945.1) genome build. PCR duplicates were removed. Nucleosome positions were determined with the DANPOS (v2.2.2)[64] with the options --smooth_width 75. The wig file generated

by DANPOS was further converted to a bigwig file with the UCSC Tool wigToBigWig (v482). The nucleosome coverage and metagene profiles were further generated by Deeptools (v3.5.4)[63].

### Statistics and reproducibility

No statistical methods were used to predetermine sample sizes, but our sample sizes are similar to those reported in previous publications[14,44]. Data distribution was assumed to be normal but this was not formally tested. Data collection and analysis were not performed blind to the conditions of the experiments. No randomization was used in the organization of experimental conditions or stimulus presentation. All data points were included in the analyses.

### Reporting summary

Further information on research design is available in the Nature Portfolio Reporting Summary linked to this article.

### Data availability

Genome sequencing results are publicly available on GEO under accession numbers GSE273678, GSE273679, GSE273680, GSE273681 and GSE300859. Whole-genome sequencing data are available as BioProject PRJNA1154251. Raw data from the single-molecule experiments have been deposited on Mendeley Data at https://data.mendeley.com/datasets/h5b45cy54s/1. Source data are provided with this paper.

### Code availability

This paper does not report any original code.

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

### Acknowledgements

We thank J. Kadonaga and G. Kassavetis for critical reading of the paper. We thank A. Roberts for cell sorting services. We are grateful to C. Kaplan (University of Pittsburgh) for providing the Pol II expression yeast strain. We thank M. Zaratiegui (Rutgers University) for providing the *S. pombe*. This work was supported by NIH grants R01GM032543 to C.B., R01GM145748 to J.F., R35GM147027 to Z. Shi and R01GM148729 to M.T. D.L. acknowledges support from grant MCB-2048095 from the NSF, grants U01AI136680, R01GM151305, U54AI170855 and P30CA014195 from the NIH, as well as the Margaret T. Morris Foundation and the Hearst Foundations for funding support. J.F. also acknowledges start-up funds from the Rutgers School of Arts and Sciences. C.B. is a Howard Hughes Medical Institute investigator.

### Author contributions

K.X. conducted foundational studies identifying NDF–FACT interactions and synergistic transcriptional effects that initiated this project. Z.L. performed the majority of experiments presented in the paper, including condensate characterization, transcriptional mechanism studies and cellular validation. F.B.-B., A.B.T. and C.L. conducted all single-molecule studies under the supervision of C.B. Structural studies were performed by Z.L., K.X., Z. Shan and D.L., with D.L. providing supervision. FRAP experiments were carried out by Z.L., H.W. and Z. Shi. J.L. performed the experiments shown in Extended Data Fig. 4d. M.T. conducted the *K*-means clustering analysis of genomic data. K.Y.K. performed the STORM analysis and interpretation. The paper was written by J.F. with input from all authors. J.F. and C.B. supervised the overall project and interpreted the results together with Z.L. and F.B.-B. Conceptualization: Z.L., F.B.-B., C.B. and J.F. Funding was acquired by D.L., Z. Shi, M.T., K.Y.K., C.B. and J.F. All authors discussed the results and commented on the paper.

### Competing interests

The authors declare no competing interests.

### Additional information

**Extended data** is available for this paper at https://doi.org/10.1038/s41556-025-01778-8.

**Correspondence and requests for materials** should be addressed to Carlos Bustamante or Jia Fei.

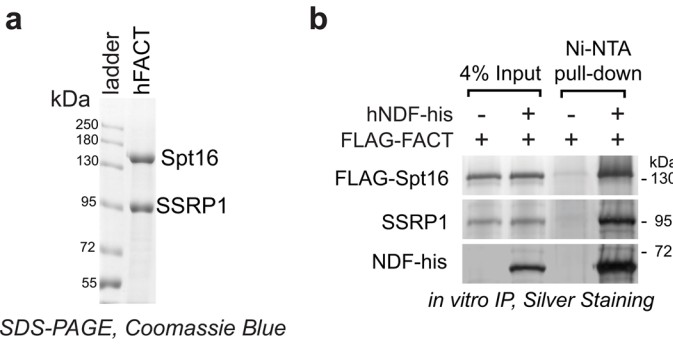

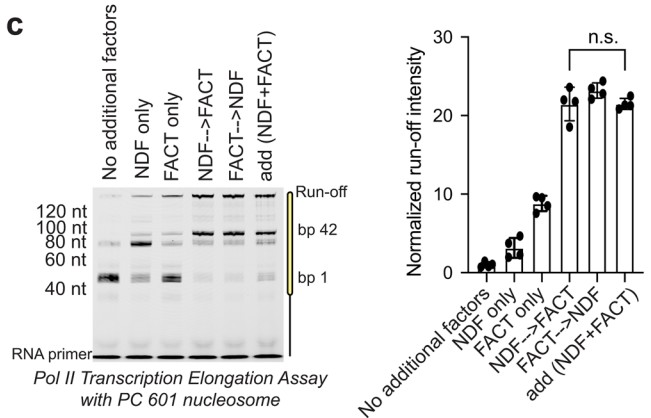

**Extended Data Fig. 1 | Control experiments demonstrating NDF-FACT interactions in vitro and their synergistic stimulation of transcription.**
**a**, SDS-PAGE analysis of FACT purified from Sf9 cells, with proteins separated on an 8% SDS-PAGE gel and visualized using Coomassie Blue staining. Three biological replicates were performed. **b**, Analysis of the interaction between recombinant NDF-his (0.3 µM) and FACT-FLAG (0.15 µM) using a Ni-NTA pull-down assay, with products separated on a 12% SDS-PAGE and detected by silver staining. Three biological replicates were performed. **c**, Nucleosome transcription assay demonstrates the synergistic effect between 0.2 µM NDF and 0.2 µM FACT when added either sequentially with 10-minute intervals or premixed before introduction to the Pol II elongation complex before adding 1 mM NTPs. The graph on the right shows statistical analysis of the run-off products based on three independent experiments (mean ± SD, two-tailed t-test).

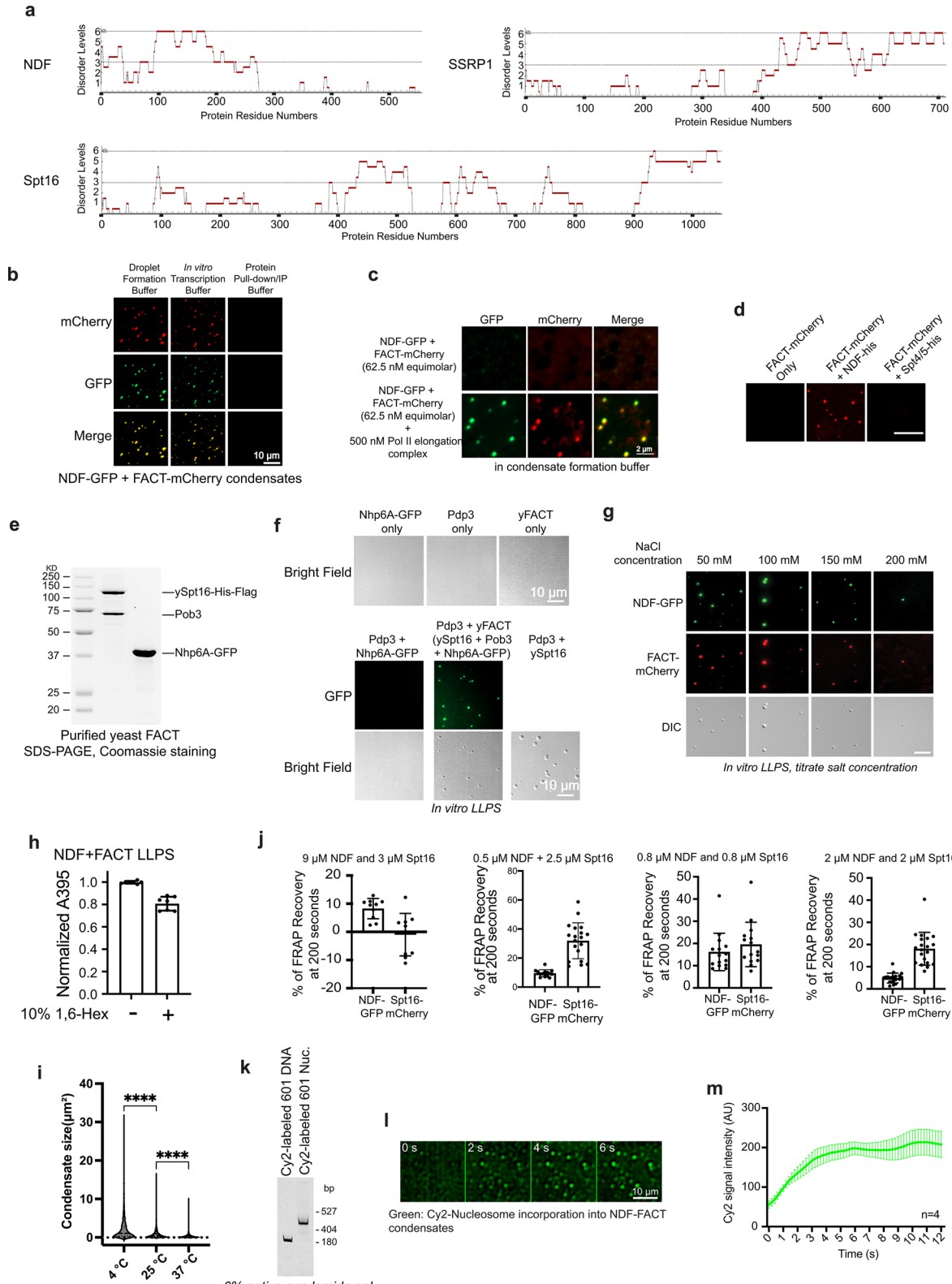

**Extended Data Fig. 2 | See next page for caption.**

**Extended Data Fig. 2 | Supporting data for in vitro characterization of NDF-FACT condensates. a**, Disorder profile of human NDF, Spt16, and SSRP1 predicted by DisMeta. **b**, Epifluorescence images of NDF-GFP (1 µM) and FACT-mCherry (1 µM) condensate under different buffer conditions. **c**, Addition of Pol II–DNA elongation complex (500 nM) promotes NDF-FACT condensate at sub-threshold protein levels (62.5 nM each) in droplet formation buffer. **d**, Purified Spt4/5 (1 µM) does not form condensates with FACT-mCherry (1 µM) in vitro. Scale bar=10 µm. **e**, 8% SDS-PAGE of yeast FACT purified from bacteria. **f**, Droplet formation assay with purified proteins (1 µM) in droplet buffer. Upper: Nhp6A, Pdp3 (yeast NDF), and yFACT (ySpt16 + Pob3) alone. Lower: Pdp3 + yFACT. **g**, Ionic strength sensitivity of NDF-FACT condensates (1 µM each) in buffers with varying NaCl. Scale bar = 10 µm. **h**, Normalized A395 after 1,6-Hex treatment (6 µM NDF + 3 µM FACT, ±10% 1,6-Hex, 5 min). Mean of three independent

experiments (7 technical replicates, mean ± SD). **i**, Condensates (1 µM each) imaged after incubation at the indicated temperatures. 4 ˚C (n = 561), 25 ˚C (n = 1,481) and 37 ˚C (n = 3,736, technical replicates). Nonparametric two-sided t-tests, ****$p < 0.0001$ **j**, FRAP photobleaching recovery rates at 200 seconds post-photobleaching for droplets in droplet formation buffer (100 mM NaCl). Mean values ± SD are shown, with n = 15, 20, 10 and 14 technical replicates for the respective panels. **k**, The native gel shows the Cy2 nucleosome used in Fig. 2h. **l-m**, Nucleosome recruitment dynamics in NDF-FACT condensates. (l)Time course of Cy2 nucleosome (green) incorporation into NDF-FACT condensates visualized by deconvolution-enhanced microscopy. Scale bar = 10 µm; (**m**) Quantification of nucleosome fluorescence intensity in condensates over time (n = 4 condensates, technical replicates, mean ± SEM). Three biological replicates were performed for panels **b-g**.

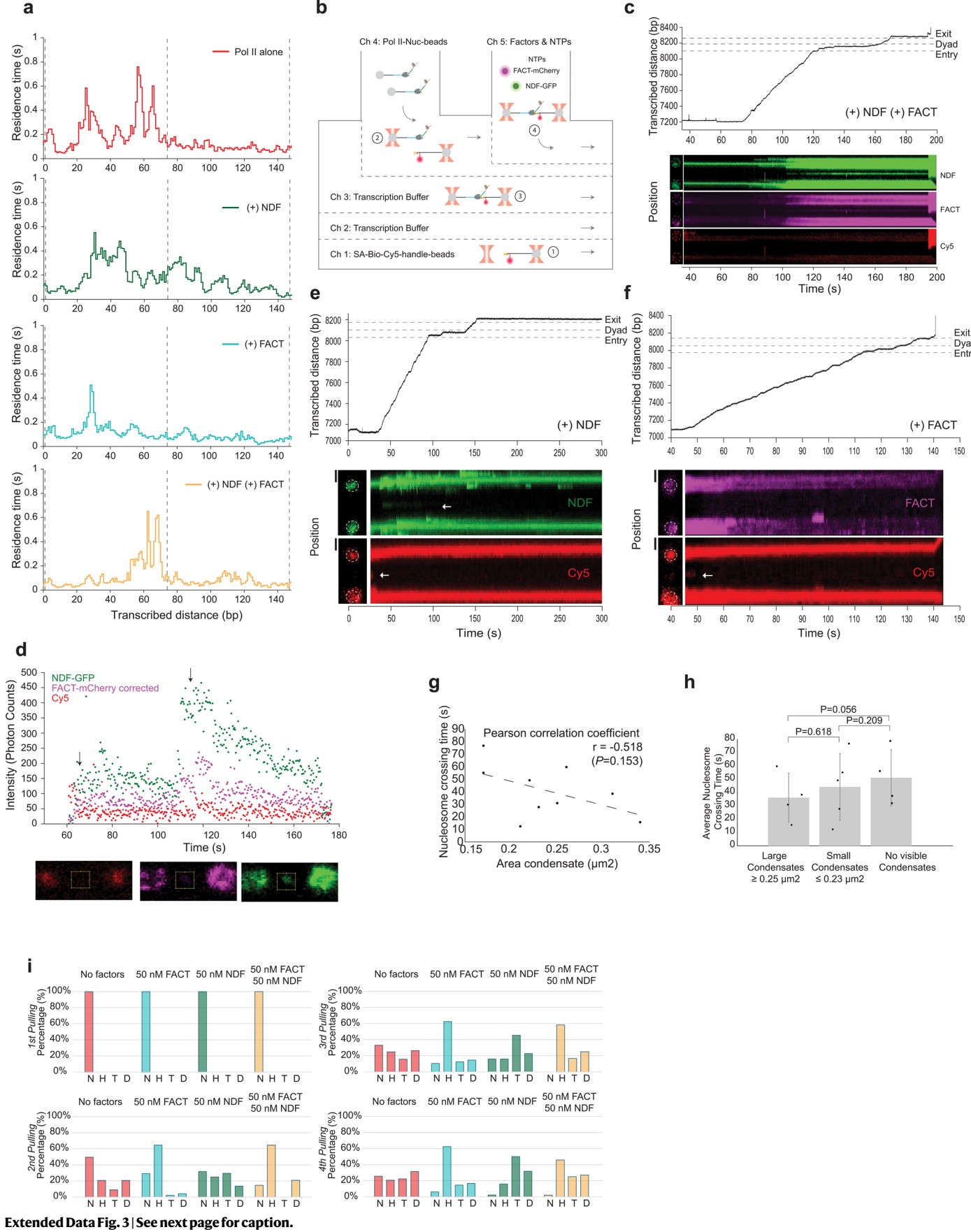

**Extended Data Fig. 3 | See next page for caption.**

**Extended Data Fig. 3 | Additional information on the single-molecule Pol II–nucleosome transcription assays. a**, Median residence time (seconds per bp) histograms of Pol II transcription through a nucleosome in the absence (red, n = 30 biological replicates), or presence of NDF (green, n = 24 biological replicates), FACT (cyan, n = 27 biological replicates), or both FACT and NDF proteins (yellow, n = 25 biological replicates). **b**, Schematic of the tweezing chamber used for the C-trap (Lumicks) experiments. Representative trace of a single Pol II transcribing through the molecular ruler and nucleosome, in the presence of both (**c**) or either (**e**) NDF or (**f**) FACT alone, accompanied by the corresponding fluorescence kymographs. Position of Cy5 molecule is shown (white arrow). The entry, dyad and exit of the nucleosome are indicated by black dashed lines. **d**, Fluorescence intensity profile of the confocal image depicted in Fig. 3e. The total number of photon counts for each laser was determined by delineating the region of interest, that is, where the condensates interact with the Pol II–nucleosome transcription system (yellow box). **g-h**, Relationship between NDF-FACT condensate size and Pol II nucleosome transcriptional efficiency. (**g**) Scatter plot showing nucleosome crossing time as a function of condensate area. (**h**) Comparison of average nucleosome crossing times across three groups: large condensates (N = 4 biological replicates), small condensates (N = 5 biological replicates), and molecules in which no condensates were detected (N = 4 biological replicates). Error bars represent standard deviation; *p*-values are from unpaired two-tailed t-tests. **i**, Nucleosome disassembly pathway obtained by performing successive pulling and relaxation cycles of a single nucleosome tethered between DNA handles attached to two optically trapped beads. The cycles were conducted in the absence (red), or presence of NDF (green), FACT (cyan), or both FACT and NDF proteins (yellow). The percentages of nucleosomes (N), hexasomes (H), tetrasomes (T), and bare DNA (D) observed after each cycle are presented. N = 50 biological replicates per condition.

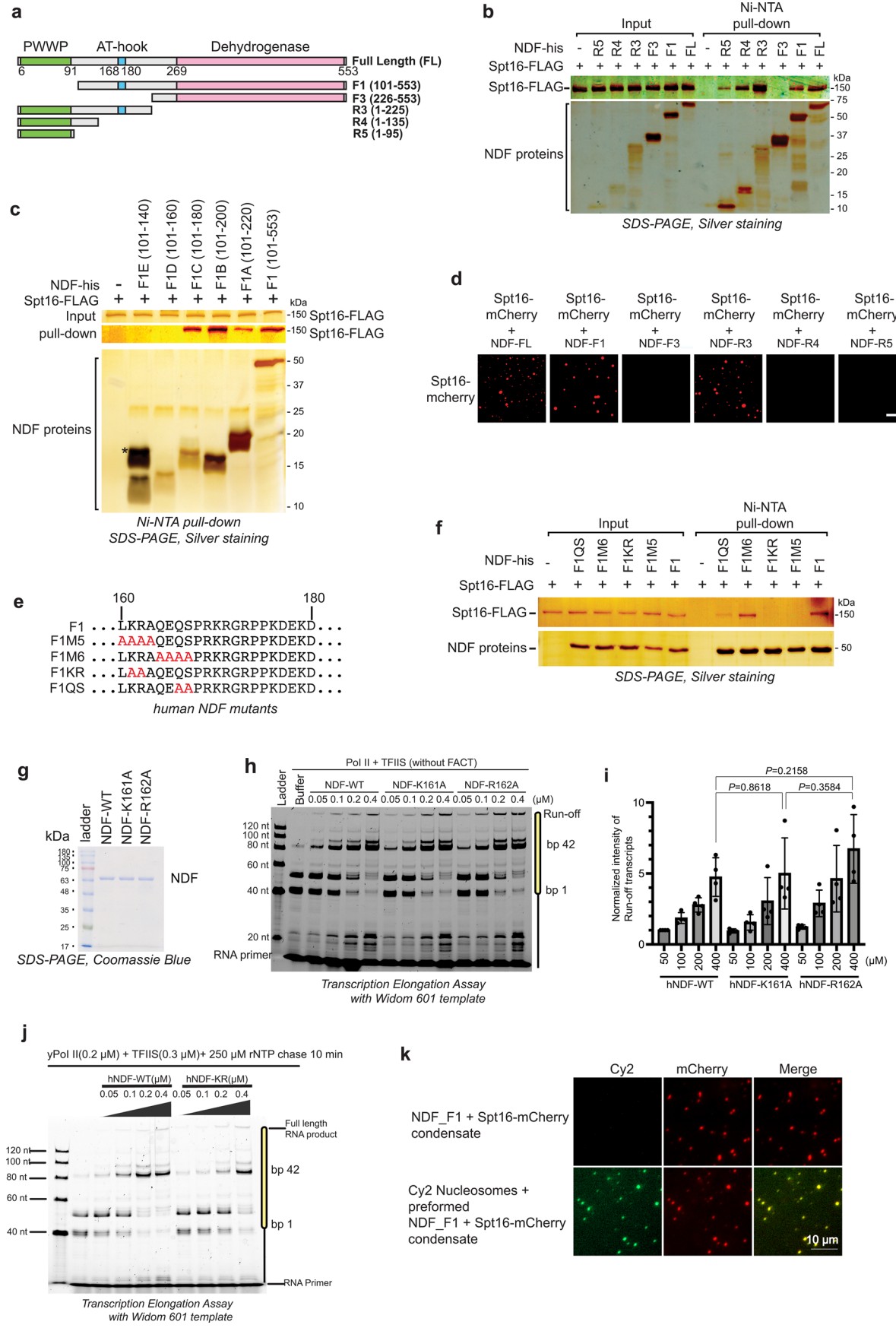

**Extended Data Fig. 4 | See next page for caption.**

**Extended Data Fig. 4 | Molecular mapping of NDF-FACT interaction domains and their impact on condensate formation and transcription. a**, Schematic of NDF and its truncated variants. **b**, Interactions between His6-tagged truncated NDF variants (**d**) and Spt16-FLAG using a Ni-NTA pull-down assay (4-20% SDS-PAGE, Silver staining). **c**, Interaction study between His6-tagged truncated variants of F1 and Spt16-FLAG using a Ni-NTA pull-down assay. An asterisk indicates a non-specific band. The F1C construct includes a linker region between NDF and His-tag, resulting in a larger apparent size on the gel. **d**, Droplet formation visualized with various NDF proteins and Spt16-mCherry at 2 µM in droplet formation buffer (100 mM NaCl, no PEG). Scale bar=10 µm. Note: While the N-terminal PWWP domain interacts with FACT *in vitro*, proteins containing only the PWWP domain (R4 and R5) do not form condensates. **e-f**, Amino acid sequences of various F1 mutants (**e**) and their interaction with Spt16, showing disruption of interactions by mutations at specific residues, as analyzed by a

Ni-NTA pull-down assay (**f**). **g**, SDS-PAGE analysis of purified full-length NDF and its mutants. **h**, Nucleosome transcription assay evaluating the impact of full-length NDF and its mutants, with transcription terminated 10 min at 30 °C after addition of NTP (0.25 mM). **i**, Bar graph comparing the production of full-length transcription products following the addition of full-length NDF or its mutants to the Pol II-nucleosome complexes over 10 min. Results are based on three independent experiments (mean ± SD, two-tailed t-test). **j**, Assessment of full-length NDF and its KR mutant (K161 and R162 mutated to alanine, AA) on Pol II transcription through nucleosomes, particularly noting decreased activity at the SHL -5 pausing site (near 40nt/bp1). Transcription was terminated 10 min after addition of NTP (0.25 mM). **k**, Microscopic images show the incorporation of Cy2-labeled nucleosomes into pre-formed (NDF_F1 + Spt16-mCherry) condensates at room temperature. Protein concentrations were 1 µM. Scale bar = 10 µm.

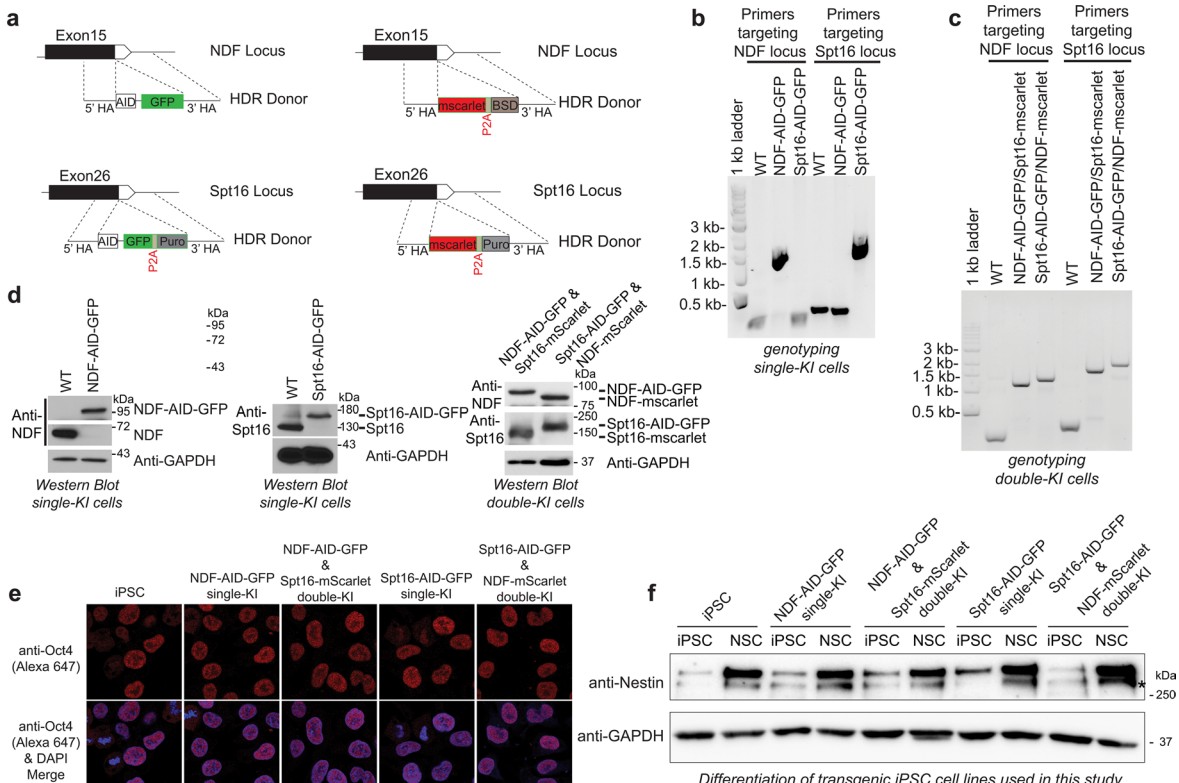

**Extended Data Fig. 5 | Generation and characterization of NDF and Spt16 single- and double-Knockin human iPSC cells. a**, Design of donor vectors for CRISPR-Cas9-mediated homologous recombination to generate single- and double-knockin (KI) cells targeting NDF and Spt16 genes. **b-c**, Agarose gel images displaying genotyping results for homozygous single (**b**) and double (**c**) KI iPSC clones. Genotyping primers: NDF-genotyping-F2 & R2 for NDF, Spt16-genotyping-F2 & R2 for Spt16. **d**, Western blot analysis comparing endogenous expression of NDF and Spt16 in wild-type (WT) and KI iPSC cells. Left Panel: WT

and NDF single KI; Middle panel: WT and Spt16 single KI; Right panel: NDF and Spt16 double KI. NDF proteins were stained with anti-NDF antibody, **e**, Immunofluorescence staining for Oct4 showing its expression pre- and post-genome editing in iPSC cells. Scale bar=5 μm. **f**, Western blot demonstrating Nestin expression in neuron stem cells derived from WT and genome-edited iPSC cells. This confirms that the edited cells retain pluripotency and physiological relevance to human stem cells. The star indicates non-specific band.

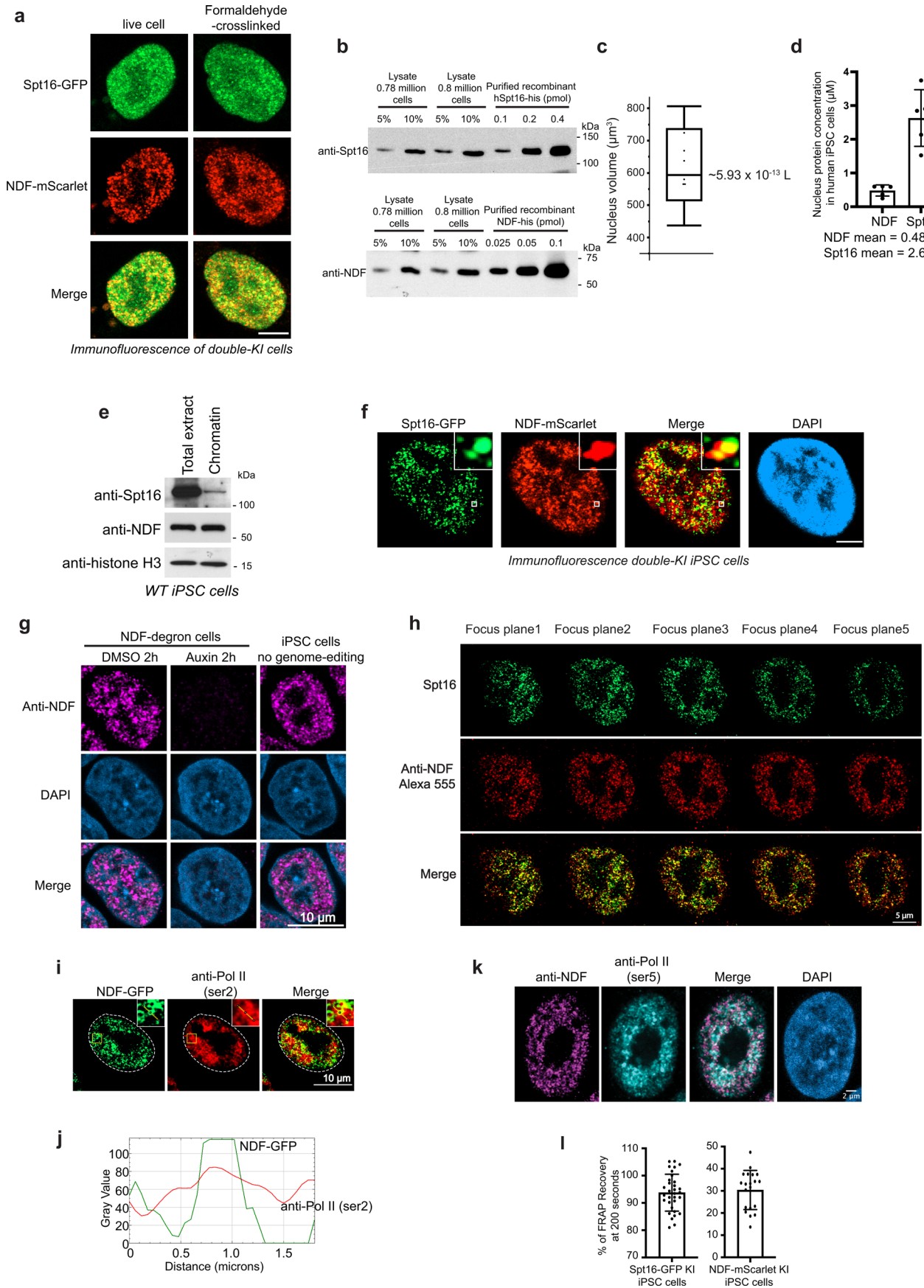

**Extended Data Fig. 6 | See next page for caption.**

**Extended Data Fig. 6 | Spatial distribution of NDF-FACT condensates in human iPSCs. a**, Representative micrographs show that Spt16 and NDF puncta patterns in live cells and cells crosslinked with formaldehyde. Scale bar = 5 μm. **b**, Western blot analysis quantifies Spt16 and NDF proteins in human iPSC cells across five independent experiments. Standard curves were constructed using known concentrations of purified recombinant proteins. **c**, Box-and-whisker plot represents the average nucleus volumes of 10 random cells (technical replicates). The ellipsoid volume is calculated as $(4/3)\pi abc$ where a,b,c are the lengths of the three semi-axes of the ellipsoid. Middle line: median; box: 25th to 75th percentiles; whiskers: minimum to maximum. **d**, Estimated concentrations of NDF and Spt16 in human iPSC cells. (mean ± SD, n = 5 biological replicates). **e**, Western blot shows total and chromatin-bound Spt16, NDF and histone H3 proteins in human iPSCs, post CSK buffer treatment for 1 minute to isolate chromatin-bound proteins. **f**, Representative images of colocalized Spt16-GFP and NDF-mScarlet in double-knock-in iPSCs. Cells were pre-extracted with CSK buffer (1 min, RT) to remove non-chromatin-bound proteins. Scale bar = 5 μm. **g**, Immunofluorescence of endogenous NDF in NDF degron cells ± auxin. Staining with NDF antibody (Alexa 647) confirms antibody specificity. **h**, NDF-FACT colocalization across nuclear distribution patterns. Representative images of four different focus planes show FACT (green) and NDF (red) distributions. Despite nuclear pattern variation, NDF and FACT remain strongly colocalized (merged, yellow). **i-k**, Immunofluorescence analysis of iPSC cells showing localization of NDF-FACT condensates and Ser2- (**i-j**) and Ser5- (**k**) phosphorylated RNA Pol II. **l**, FRAP recovery percentages of Spt16-GFP and NDF-mScarlet in human iPSC at 200 seconds post-photobleaching. Mean ± SD, n = 32 and n = 19 for Spt16-GFP and NDF-mScarlet, respectively.

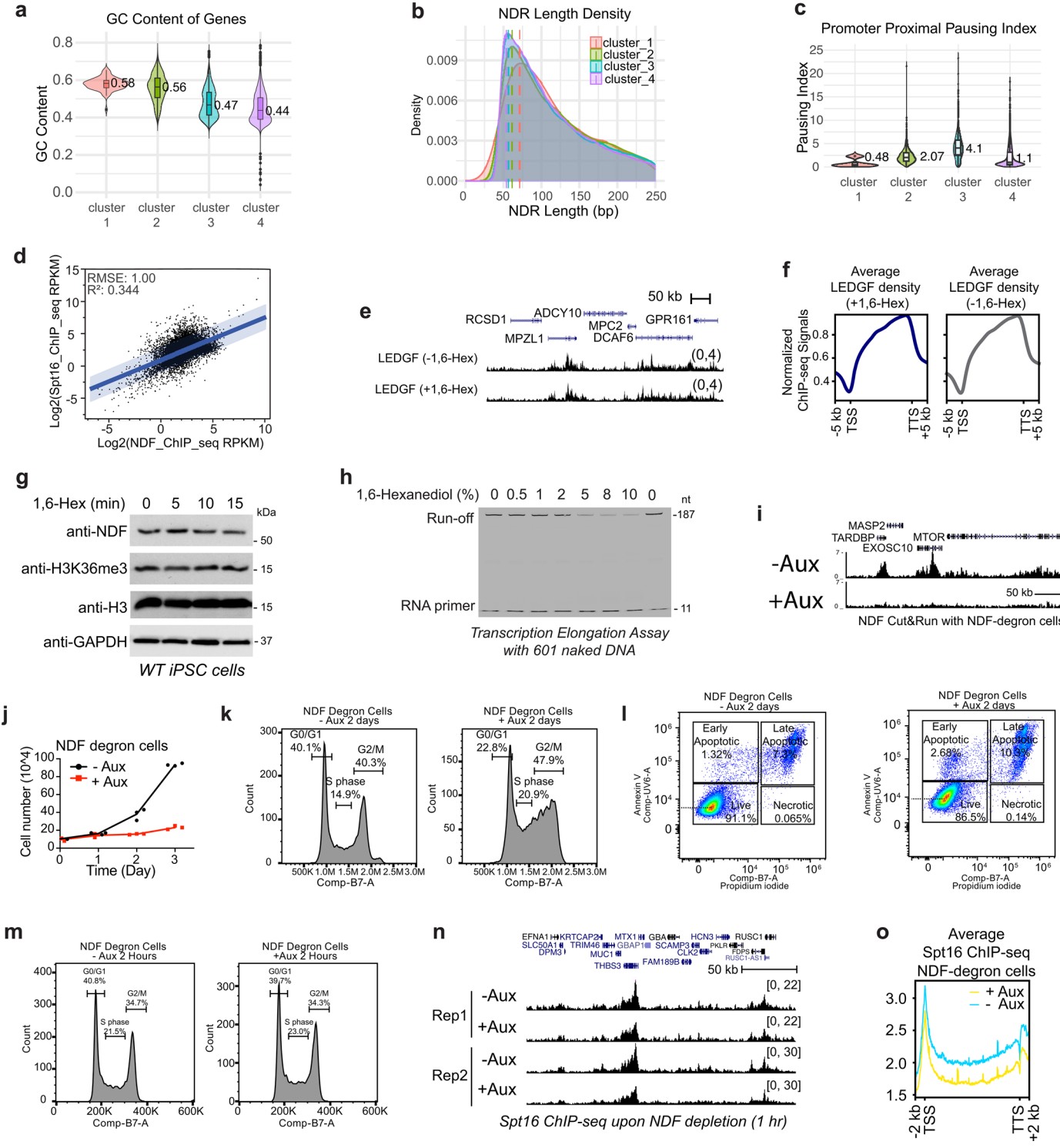

**Extended Data Fig. 7 | Validation and characterization of NDF depletion effects on chromatin association and cellular function. a-c**, Genomic features of NDF-Spt16 co-occupancy gene clusters. Box plots of (a) GC content, (b) NDR lengths, and (c) pausing indices across four clusters from k-means analysis of NDF and Spt16 ChIP-seq data. Cluster mean values are labeled. n = genes per cluster (Cluster 1: 60; Cluster 2: 1,252; Cluster 3: 7,705; Cluster 4: 11,107). Box plots show median (center line), quartiles (box), and whiskers (1.5× IQR). **d**, NDF and Spt16 ChIP-seq reads (RPKM, TSS-TTS) in iPSC show a positive correlation (best-fit linear regression). **e-f**, LEDGF ChIP-seq genome browser view (**e**) and metagene analysis (**f**) in iPSC treated with 1.5% 1,6-Hex for 10 min vs DMSO. **g**, Western blot of NDF, H3K36me3, and H3 after 3% 1,6-Hex treatment for indicated times in iPSC. **h**, Transcription elongation assay on a 601 DNA template with varying 1,6-Hex concentrations shows Pol II inhibition in vitro. **i**, Genome browser view of NDF Cut&Run in untreated and 2 h auxin-treated NDF-degron iPSCs. **j**, Cell growth curve of NDF-degron iPSC ± auxin. n = 3. **k**, Cell cycle analysis of NDF-degron iPSC cells after 2 days of treatment with DMSO or auxin. Results suggest that NDF depletion in human stem cells leads to substantial cell cycle disruptions. **l**, Annexin V staining after 2 days ± auxin reveals increased apoptosis with NDF depletion. **m**, Initial impact of NDF depletion on cell cycle analyzed over the first two h of DMSO or auxin treatment in NDF-degron iPSCs, showing no immediate abnormalities. **n-o**, ChIP-seq data presentation for Spt16 in NDF-degron iPSCs, untreated and 1-hour post-auxin treatment. Genome browser views (**n**) and metagene analysis (**o**) normalized to total reads across two independent experiments highlight changes in Spt16 binding dynamics.

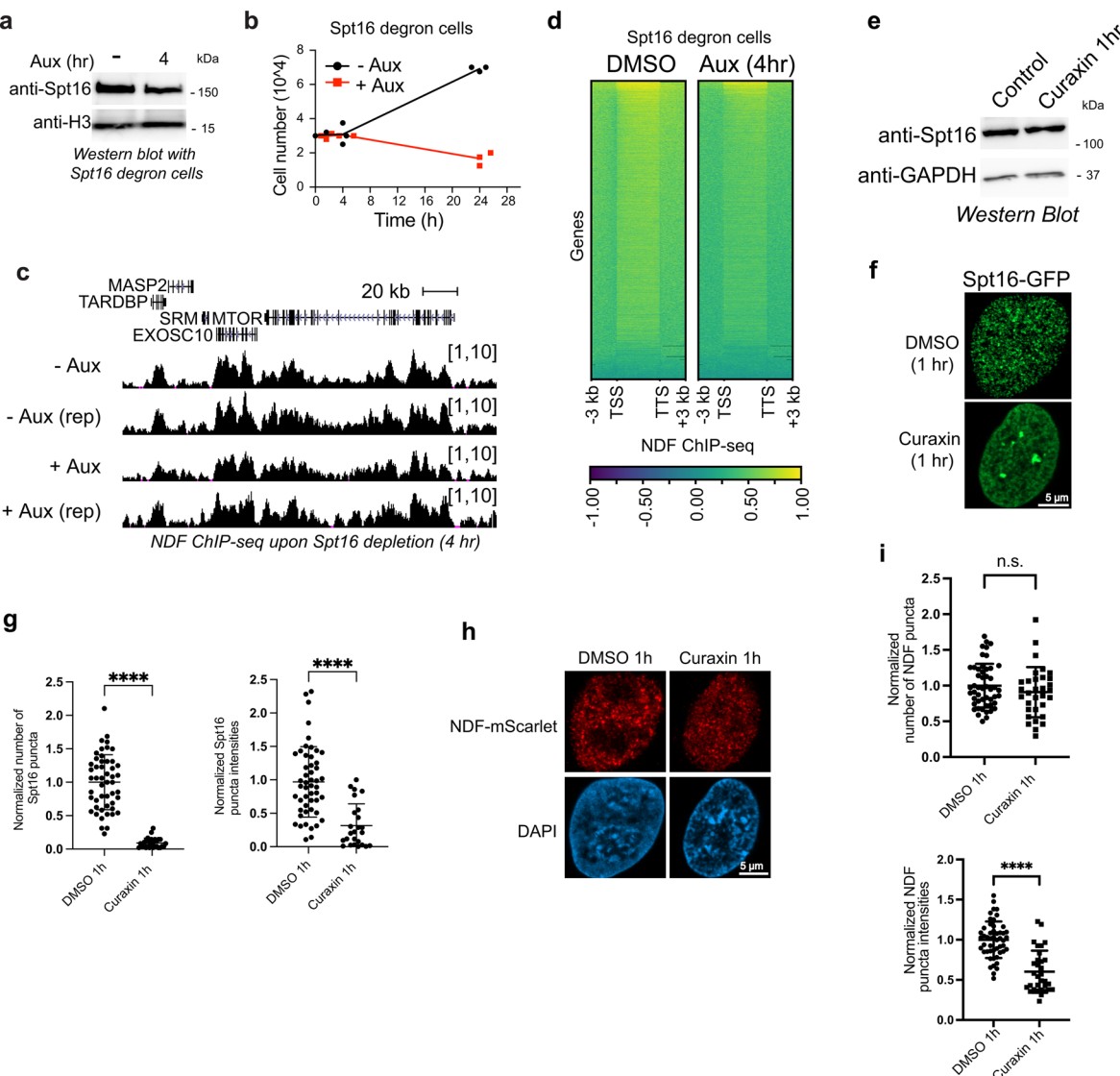

**Extended Data Fig. 8 | FACT inhibition disrupts NDF nuclear localization.**
**a**, Western blot analysis of Spt16 levels in Spt16-degron iPSCs treated with DMSO or auxin for 4 h, revealing a slight reduction in Spt16 protein levels. **b**, Growth curve of Spt16-degron iPSCs treated with or without auxin over 24 h. The data indicate a critical dependency on Spt16, with over 90% cell death observed within 24 h. **c-d**, ChIP-seq data for NDF in Spt16-degron iPSCs, untreated and 4-hour post-auxin treatment. Genome browser views (**c**) and metagene heatmap (**d**) normalized to total reads across two independent experiments, suggesting mild changes in NDF chromatin enrichment upon partial Spt16 depletion.
**e**, Western blot analysis of Spt16 levels upon curaxin treatment. **f**, Treatment of Spt16-GFP knock-in iPSCs with 2 µM curaxin for 1 hour rapidly alters the pattern of Spt16 puncta. **g**, Scatter plot of normalized numbers (left panel) and normalized intensities (right panel) of Spt16 puncta upon treatment of curaxin.

Puncta intensity was normalized to total cellular fluorescence. Data show ~90% decrease in puncta number (p < 0.0001) and ~70% reduction in puncta intensity (****p < 0.0001) (DMSO 1 h: n = 50 cells; curaxin 1 h: n = 32 cells for number analysis, and n = 25 cells for intensity analysis, mean ± SD, technical replicates, repeated twice) **h**, Treatment of NDF-mScarlet knock-in iPSC with 2 µM curaxin for 1 hour rapidly abolished the formation of NDF puncta. **i.** Scatter plots display the normalized counts and normalized intensities (relative to total cellular intensity) of NDF-mScarlet puncta after treatment with either DMSO or 2 µM curaxin. The data indicate that while the number of puncta remained largely unaffected, the intensity of the puncta was significantly reduced. ****P < 0.0001. n.s. not significant. n = cells (DMSO: 49; curaxin: 31), technical replicates, repeated twice, mean ± SD, panels g and i were analyzed by nonparametric two-sided t-test.

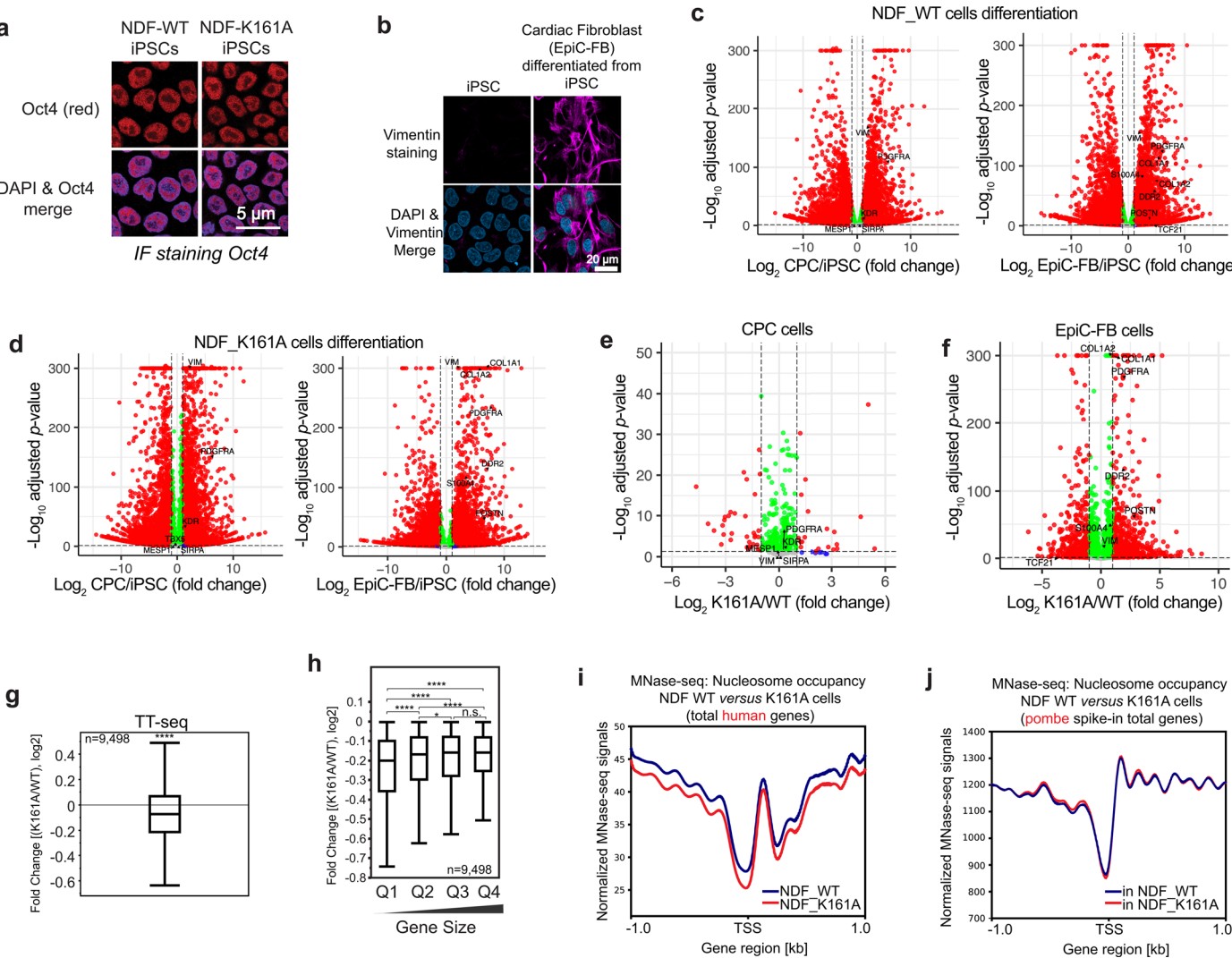

**Extended Data Fig. 9 | Characterization of NDF_K161A stem cells and alternative normalization of nucleosome occupancy data. a**, Oct4 immunofluorescence in NDF_WT and NDF_K161A iPSCs. Scale bar = 5 μm. **b**, Vimentin immunofluorescence in undifferentiated iPSCs and differentiated EPIC-FB cells. Scale bar = 20 μm. **c**, Volcano plots showing differential gene expression in NDF-WT cells during differentiation. Left: iPSC vs cardiac progenitor cell (CPC) stage. Right: iPSC vs EPIC-derived fibroblast-like cell (EPIC-FB) stage. **d**, Volcano plots showing differential gene expression in NDF-K161A cells during differentiation. Left: iPSC vs CPC stage. Right: iPSC vs EPIC-FB stage. **e**, Volcano plot comparing gene expression between NDF-K161A and NDF-WT cells at the cardiac progenitor cell (CPC) stage. **f**, Volcano plot comparing gene expression between NDF-K161A and NDF-WT cells at the EPIC-FB stage. **g**, Boxplot of nascent RNA synthesis rates (TT-seq) in WT and NDF_K161A cells normalized to *S. pombe* spike-in nascent RNA. NDF_K161A shows reduced synthesis (n = 14,956 genes, P < 0.001, two-sided one-sample t-test, Box plots

show median (centre), Q1 and Q3 (box), and min/max (whiskers).) **h**, Nascent RNA synthesis impairment (NDF_K161A/WT fold change) stratified by gene length quartiles (Q1-Q4, shortest to longest). Only genes with at least 1 read were analyzed. Shorter genes showed significantly greater impairment compared to longer genes (*P = 0.0266, ****P < 0.0001, two-sided t-test). (n = 9,498 genes with decreased nascent RNA levels in K161A cells relative to WT cells. Box plots show median (centre), Q1 and Q3 (box), and min/max (whiskers).) **i**, Metagene analysis of nucleosome positioning from MNase-seq in NDF_WT and NDF_K161A cells with S. pombe spike-in controls. Reduced nucleosome occupancy is observed in NDF_K161A. **j**, Metagene analysis of *S. pombe* spike-in nucleosome signals from the same MNase-seq experiment described in panel i. Analysis includes all S. pombe genes, demonstrating consistent digestion efficiency between NDF_WT and NDF_K161A samples. Genes with adjusted p-value (padj) < 0.05 and |log₂ fold change | ≥ 1 from DESeq2 analysis are highlighted in red (**c**–**f**).

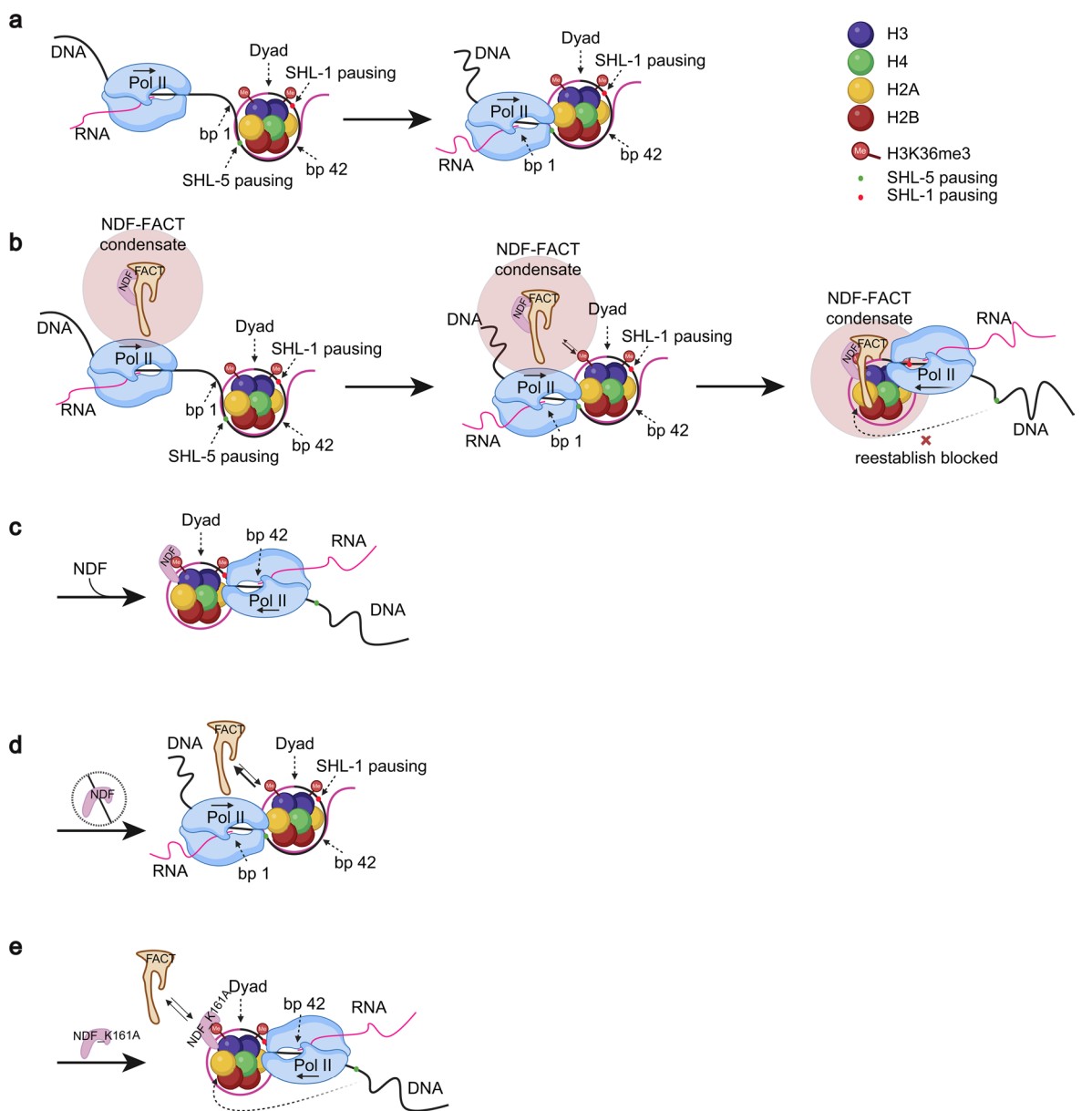

**Extended Data Fig. 10 | A model of NDF-FACT condensates mediated Pol II transcription through nucleosomes. a.** Nucleosomes form barriers to Pol II transcription elongation, causing the first pause near superhelical location SHL-5, where the leading edge of Pol II cannot bypass. At this point, the catalytic center of Pol II reaches approximately base pair 1 (bp1). The second pause occurs near SHL-1, where the catalytic center of Pol II reaches approximately base pair 42 (bp42). **b.** During Pol II elongation, NDF-FACT condensates move together with Pol II. When Pol II brings the condensates into proximity of the nucleosome, the condensates can interact with the nucleosome. NDF-FACT condensates enhance transcription efficiency by engaging the FACT complex. The SPT16 subunit of FACT binds nucleosomal DNA, tethers the proximal H2A-H2B dimer, and acts as a placeholder for DNA, preventing re-coiling of DNA behind the enzyme. **c.** NDF specifically alleviates the pause near SHL-5, allowing Pol II to continue transcription downstream until it reaches the SHL-1 pause. Without additional factors, Pol II cannot transcribe further, and the DNA behind the enzyme may re-coil around the octamer. **d.** Upon rapid depletion of NDF, Pol II requires other protein factors, to bypass the SHL-5 pause site. Without the formation of condensates, FACT recruitment is substantially impaired, resulting in low transcription efficiency and cell death in human stem cells. **e.** In cells expressing NDF_K161A, the formation of NDF-FACT condensates is reduced. This affects the formation of condensates and recruitment of FACT, impairing transcription elongation efficiency. Figure created with BioRender.com.

# Reporting Summary

## Statistics

For all statistical analyses, confirm that the following items are present in the figure legend, table legend, main text, or Methods section.

| n/a | Confirmed | |
|---|---|---|
| ☐ | ☒ | The exact sample size (*n*) for each experimental group/condition, given as a discrete number and unit of measurement |
| ☐ | ☒ | A statement on whether measurements were taken from distinct samples or whether the same sample was measured repeatedly |
| ☐ | ☒ | The statistical test(s) used AND whether they are one- or two-sided<br>*Only common tests should be described solely by name; describe more complex techniques in the Methods section.* |
| ☒ | ☐ | A description of all covariates tested |
| ☒ | ☐ | A description of any assumptions or corrections, such as tests of normality and adjustment for multiple comparisons |
| ☐ | ☒ | A full description of the statistical parameters including central tendency (e.g. means) or other basic estimates (e.g. regression coefficient) AND variation (e.g. standard deviation) or associated estimates of uncertainty (e.g. confidence intervals) |
| ☐ | ☒ | For null hypothesis testing, the test statistic (e.g. *F*, *t*, *r*) with confidence intervals, effect sizes, degrees of freedom and *P* value noted<br>*Give P values as exact values whenever suitable.* |
| ☒ | ☐ | For Bayesian analysis, information on the choice of priors and Markov chain Monte Carlo settings |
| ☒ | ☐ | For hierarchical and complex designs, identification of the appropriate level for tests and full reporting of outcomes |
| ☒ | ☐ | Estimates of effect sizes (e.g. Cohen's *d*, Pearson's *r*), indicating how they were calculated |

*Our web collection on statistics for biologists contains articles on many of the points above.*

## Software and code

Policy information about availability of computer code

| | |
|---|---|
| Data collection | Paired-end sequencing was conducted using Illumina NovaSeq X Plus.<br>ZEN (version 2.6) imaging software was used for collecting confocal images.<br>Leica Application Suite X (LAS X) was used for collecting FRAP images. |
| Data analysis | For ChIP-seq, Cut&Run, Pro-seq and MNase-seq, the sequencing data were trimmed with Fastp (version 0.23.4) and Trim Galore! (version 0.6.7) Programs and aligned to the hg19 reference genome using Bowtie2 (version 2.5.3). Spike-in reads were mapped to the dm3 genome build. PCR duplicates were removed with the samtools (version 1.15.1). For ChIP-seq, Cut&Run, Pro-seq, the normalized bigwig files and metagene analysis were prepared by Homer (version 4.10) or Deeptools (version 3.5.4).  For MNase-seq, the bam files generated by Bowtie2 were further analysis by Danpos (version 2.2.2), the wig files generated by Danpos were converted to bigwig files with UCSC Tool wigToBigWig (v482). The nucleosome coverage an metagene profiles were further generated by Deeptools and normalized base on the reads of 2,437 transcriptionally inactive genes in cluster 4 (Figure 4c).  For RNA-seq, the raw sequencing data were trimmed and aligned to the hg19 reference genome and the ERCC Spike-in RNA annotation using Hisat2 (version 2.2.1). Total reads from each gene were calculated using featureCounts (version 2.0.3). Differential gene expression analysis was performed using DESeq2 (v2.11.40.8), and the results were visualized using the EnhancedVolcano (v1.20.0, https://github.com/kevinblighe/EnhancedVolcano) package. For TT-seq, raw sequencing reads were trimmed and aligned to both the human reference genome (hg38) and S. pombe genome using STAR (version 2.7.11b). PCR duplicates were removed using samtools. Mapped BAM files were used to analyze transcriptional wave peaks and RNA polymerase II elongation speed, following previously described protocols with minor modifications. Analyses were restricted to genes longer than 50kb with non-overlapping transcription units (n = 2,558 transcripts in hg38). Gene regions were extended from −2kb upstream to +80kb downstream of the transcription start site (TSS); any extensions beyond chromosome boundaries were excluded.The images were processed and analysis with Fiji (version 2.14.0). For single-molecule data, the raw data was exported from the optical tweezers instruments and converted to .mat files to be analyzed with custom-written Matlab code (https://github.com/abmtong/BLabOTMatlab/archive/refs/heads/master.zip). This code was used |

to align the transcription traces, plot the transcribed distance in bp, define the position of Pol II along the template, generate the transcriptional maps of the nucleosome, and correlate the optical tweezers channel with the single molecule fluorescence channel.

For manuscripts utilizing custom algorithms or software that are central to the research but not yet described in published literature, software must be made available to editors and reviewers. We strongly encourage code deposition in a community repository (e.g. GitHub). See the Nature Portfolio guidelines for submitting code & software for further information.

## Data

Policy information about availability of data

All manuscripts must include a data availability statement. This statement should provide the following information, where applicable:
- Accession codes, unique identifiers, or web links for publicly available datasets
- A description of any restrictions on data availability
- For clinical datasets or third party data, please ensure that the statement adheres to our policy

ChIP-Seq, CUT&RUN, Pro-seq and RNA-Seq data have been deposited at the NCBI gene expression omnibus (GEO; www.ncbi.nlm.nih.gov/geo/) with the GEO accession number GSE273678, GSE273679, GSE273680 and GSE273681 respectively.

## Research involving human participants, their data, or biological material

Policy information about studies with human participants or human data. See also policy information about sex, gender (identity/presentation), and sexual orientation and race, ethnicity and racism.

| Reporting on sex and gender | N/A |
|---|---|
| Reporting on race, ethnicity, or other socially relevant groupings | N/A |
| Population characteristics | N/A |
| Recruitment | N/A |
| Ethics oversight | N/A |

Note that full information on the approval of the study protocol must also be provided in the manuscript.

# Field-specific reporting

Please select the one below that is the best fit for your research. If you are not sure, read the appropriate sections before making your selection.

☒ Life sciences          ☐ Behavioural & social sciences          ☐ Ecological, evolutionary & environmental sciences

For a reference copy of the document with all sections, see nature.com/documents/nr-reporting-summary-flat.pdf

# Life sciences study design

All studies must disclose on these points even when the disclosure is negative.

| Sample size | The deep sequencing data were at least independently repeated twice. And all statistically analyzed experiments were independently repeated at least three times and determined according to standard molecular biology procedures. |
|---|---|
| Data exclusions | None |
| Replication | Biological replicates were used for the deep sequencing data. The ChIP-seq data in Extended Data Fig. 9d (without replicates) was validated by CUT&RUN data, and vice versa. At least three independent replicates were performed for statistically analyzed experiments. |
| Randomization | N/A |
| Blinding | N/A |

# Reporting for specific materials, systems and methods

We require information from authors about some types of materials, experimental systems and methods used in many studies. Here, indicate whether each material, system or method listed is relevant to your study. If you are not sure if a list item applies to your research, read the appropriate section before selecting a response.

## Materials & experimental systems

| n/a | Involved in the study |
|---|---|
| ☐ | ☒ Antibodies |
| ☐ | ☒ Eukaryotic cell lines |
| ☒ | ☐ Palaeontology and archaeology |
| ☒ | ☐ Animals and other organisms |
| ☒ | ☐ Clinical data |
| ☒ | ☐ Dual use research of concern |
| ☒ | ☐ Plants |

## Methods

| n/a | Involved in the study |
|---|---|
| ☐ | ☒ ChIP-seq |
| ☐ | ☒ Flow cytometry |
| ☒ | ☐ MRI-based neuroimaging |

# Antibodies

| | |
|---|---|
| Antibodies used | Rabbit polyclonal antisera against hNDF; anti-hSpt16 (Cell Signaling, 12191S); anti-GAPDH (Cell Signaling, 5174S); anti-LEDGF (Proteintech, 25504-1-AP); anti-H3 (Cell Signaling, 4499); anti-H3K36me3 (Abcam, ab9050); anti-Rpb2/Pol II (Genetex, GTX102535); anti-Oct4 (Cell Signaling, 2750S); anti-GLYR1 (proteintech, 14833-A-AP); anti-Nestin (Cell Signaling, 33475S); ChIP-seq spike-in antibody (Active Motif, 61686); Donkey anti-Rabbit IgG (H+L) Highly Cross-Adsorbed Secondary Antibody, Alexa Fluor™ 555 (ThermoFisher, A31572); anti-rabbit IgG (H+L), F(ab')¬2 Fragment (Alexa Fluor 647 conjugate) (Cell Signaling, 4414S); anti-Vimentin (Cell Signaling, 5741); anti-RNA pol II CTD phospho Ser2 antibody (Abcam, ab237280); anti-RNA pol II CTD phospho Ser5 antibody (Active Motif, 61085). |
| Validation | Rabbit polyclonal antisera against hNDF: Tested in Western Blot and chromatin IP. Fei, J. et al. NDF, a nucleosome-destabilizing factor that facilitates transcription through nucleosomes. Genes Dev 32, 682-694 (2018). https://doi.org/10.1101/gad.313973.118<br>anti-hSpt16: Tested in Western Blot and chromatin IP. RRID: AB_2732025<br>anti-GAPDH: Tested in Western Blot. RRID: AB_10622025<br>anti-LEDGF:  Tested in Western blot (Proteintech)<br>anti-H3: Tested in Western Blot. RRID: AB_10544537<br>anti-H3K36me3: Tested in Western Blot. RRID: AB_306966<br>anti-Rpb2/Pol II: Tested in Western Blot and chromatin IP. RRID: AB_1951313<br>anti-Oct4: Tested in immunofluorescence. RRID: AB_823583<br>anti-GLYR1: Tested in immunofluorescence. RRID: AB_10859775<br>anti-Nestin: Tested in Western Blot. RRID: AB_2799037<br>ChIP-seq spike-in antibody: Tested in chromatin IP. RRID: AB_2737370<br>Donkey anti-Rabbit IgG (H+L) Highly Cross-Adsorbed Secondary Antibody, Alexa Fluor™ 555: Tested in immunofluorescence. RRID: AB_162543<br>anti-rabbit IgG (H+L), F(ab')-2 Fragment (Alexa Fluor 647 conjugate): Tested in immunofluorescence. RRID: AB_10693544<br>anti-Vimentin:  Tested in Western blot and Immunofluorescence. RRID: AB_10695459<br>anti-RNA pol II CTD phospho Ser2 antibody: Tested in Immunofluorescence (Abcam);<br>anti-RNA pol II CTD phospho Ser5 antibody: Tested in Immunofluorescence. RRID: AB_2687451. |

# Eukaryotic cell lines

Policy information about cell lines and Sex and Gender in Research

| | |
|---|---|
| Cell line source(s) | iPSC cell line GM25256, purchased from the Coriell Institute for Medical Research<br>Hela, HEK293T, and SW480 cells |
| Authentication | None of the cell lines were authenticated. |
| Mycoplasma contamination | Cell lines were not tested for mycoplasma contamination. |
| Commonly misidentified lines (See ICLAC register) | No commonly misidentified cell lines were used. |

# Plants

| | |
|---|---|
| Seed stocks | N/A |
| Novel plant genotypes | N/A |
| Authentication | N/A |

# ChIP-seq

## Data deposition

☒ Confirm that both raw and final processed data have been deposited in a public database such as GEO.

☒ Confirm that you have deposited or provided access to graph files (e.g. BED files) for the called peaks.

| | |
|---|---|
| Data access links<br>*May remain private before publication.* | https://www.ncbi.nlm.nih.gov/geo/query/acc.cgi?acc=GSE273678<br>Genome sequencing results are available on GEO under accession number GSE273678, GSE273679, GSE273680, and GSE273681 are publicly available. |
| Files in database submission | FASTQ and bigWig files for all ChIP-seq performed in this study |
| Genome browser session<br>(e.g. UCSC) | https://genome.ucsc.edu/s/ziweilirutgers/NDF%2DFACT%20Collection%2Dseperate%206 |

## Methodology

| | |
|---|---|
| Replicates | 2-5 |
| Sequencing depth | ≥ 20 million read pairs per sample<br>Number of Reads for each samples are listed below.<br>iPSC, NDF, ChIP_Rep1 19497569<br>iPSC, NDF, ChIP_Rep2 18296153<br>iPSC, Spt16, ChIP_Rep1 26023384<br>iPSC, Spt16, ChIP_Rep2 25132222<br>iPSC, Pol II, ChIP_Rep1 20305379<br>iPSC, Pol II, ChIP_Rep2 27163792<br>iPSC, Input 43154770<br>iPSC, NDF, ChIP_Con_Rep2 44721659<br>iPSC, NDF, ChIP_Hex_Rep1 49798214<br>iPSC, NDF, ChIP_Hex_Rep2 44050321<br>iPSC, Spt16, ChIP_Con_Rep1 39163867<br>iPSC, Spt16, ChIP_Con_Rep2 39613162<br>iPSC, Spt16, ChIP_Hex_Rep1 37919738<br>iPSC, Spt16, ChIP_Hex_Rep2 28861296<br>iPSC, Pol II, ChIP_Con_Rep1 37034310<br>iPSC, Pol II, ChIP_Con_Rep2 38533346<br>iPSC, Pol II, ChIP_Hex_Rep1 44004748<br>iPSC, Pol II, ChIP_Hex_Rep2 43605006<br>iPSC, LEDGF, ChIP_Con_Rep1 27134165<br>iPSC, LEDGF, ChIP_Con_Rep2 31075853<br>iPSC, LEDGF, ChIP_Hex_Rep1 32555943<br>iPSC, LEDGF, ChIP_Hex_Rep2 27590244<br>NDF-degron, Spt16, ChIP_DMSO1h_Rep1 27858945<br>NDF-degron, Spt16, ChIP_Aux1h_Rep1 27019702<br>NDF-degron, Spt16, ChIP_DMSO1h_Rep2 9253232<br>NDF-degron, Spt16, ChIP_Aux1h_Rep2 27697793<br>NDF-degron, Spt16, ChIP_DMSO1h_Rep3 31638338<br>NDF-degron, Spt16, ChIP_Aux1h_Rep3 41968610<br>NDF-degron, Spt16, ChIP_DMSO2h_Rep1 35993847<br>NDF-degron, Spt16, ChIP_Aux2h_Rep1 36676851<br>NDF-degron, Spt16, ChIP_DMSO2h_Rep2 50388156<br>NDF-degron, Spt16, ChIP_Aux2h_Rep2 42865323<br>NDF-degron, Spt16, ChIP_DMSO2h_Rep3 25126834<br>NDF-degron, Spt16, ChIP_Aux2h_Rep3 8336327<br>NDF-degron, Pol II, ChIP_DMSO2h_Rep1 88514789<br>NDF-degron, Pol II, ChIP_DMSO2h_Rep2 101037781<br>NDF-degron, Pol II, ChIP_Aux2h_Rep1 91731299<br>NDF-degron, Pol II, ChIP_Aux2h_Rep2 81334799<br>Spt16-degron, NDF, ChIP_DMSO4h_Rep1 28078730<br>Spt16-degron, NDF, ChIP_DMSO4h_Rep2 27575689<br>Spt16-degron, NDF, ChIP_Aux4h_Rep2 25262944<br>Spt16-degron, NDF, ChIP_Aux4h_Rep1 23137748<br>iPSC, NDF-WT, NDF, ChIP 29390351<br>iPSC, NDF-K, NDF, ChIP 27862531<br>iPSC, NDF-WT, Spt16, ChIP_Rep1 47814475<br>iPSC, NDF-WT, Spt16, ChIP_Rep2 43207389<br>iPSC, NDF-K, Spt16, ChIP_Rep1 47832626<br>iPSC, NDF-K, Spt16, ChIP_Rep2 43066968<br>NDF-degron, MNase_DMSO2h_Rep1 69562538<br>NDF-degron, MNase_Aux2h_Rep1 67487269<br>NDF-degron, MNase_DMSO2h_Rep2 85199559 |

NDF-degron, MNase_Aux2h_Rep2 79756405
NDF_WT, MNase 137013965
NDF_K161A, MNase 141917982
CPC, NDF-WT, RNA_seq_Rep1 15057749
CPC, NDF-WT, RNA_seq_Rep2 17048811
CPC, NDF-K, RNA_seq_Rep1 16370770
CPC, NDF-K, RNA_seq_Rep2 15596866
EpiC-FB, NDF-WT, RNA_seq_Rep1 16290706
EpiC-FB, NDF-WT, RNA_seq_Rep2 16517222
EpiC-FB, NDF-K, RNA_seq_Rep1 16007324
EpiC-FB, NDF-K, RNA_seq_Rep2 17535979
iPSC, NDF-WT, TT_seq_Rep1 41310192
iPSC, NDF-K, TT_seq_Rep1 47253876
iPSC, NDF-K, TT_seq_Rep2 47832290
iPSC, NDF-WT, DRB_TT_seq_0min_Rep1 42783378
iPSC, NDF-WT, DRB_TT_seq_10min_Rep1 45491724
iPSC, NDF-WT, DRB_TT_seq_10min_Rep2 36792511
iPSC, NDF-WT, DRB_TT_seq_20min_Rep1 41911359
iPSC, NDF-WT, DRB_TT_seq_20min_Rep2 35123866
iPSC, NDF-K, DRB_TT_seq_0min_Rep1 38242253
iPSC, NDF-K, DRB_TT_seq_0min_Rep2 41009986
iPSC, NDF-K, DRB_TT_seq_10min_Rep1 36425927
iPSC, NDF-K, DRB_TT_seq_10min_Rep2 33586462
iPSC, NDF-K, DRB_TT_seq_20min_Rep1 32218824
iPSC, NDF-K, DRB_TT_seq_20min_Rep2 30264401

**Antibodies**

See antibodies above

**Peak calling parameters**

no peak calling.

**Data quality**

Number of reads are listed above.

**Software**

See software above

# Flow Cytometry

## Plots

Confirm that:

☒ The axis labels state the marker and fluorochrome used (e.g. CD4-FITC).

☒ The axis scales are clearly visible. Include numbers along axes only for bottom left plot of group (a 'group' is an analysis of identical markers).

☒ All plots are contour plots with outliers or pseudocolor plots.

☒ A numerical value for number of cells or percentage (with statistics) is provided.

## Methodology

**Sample preparation**

For cell cycle analysis, the cells were dissociated with 1x TrypLETM-Select Enzyme (ThermoFisher, 12563029), about 2 million cells were washed with PBS and and resuspended in 300 µL of cold (4°C) PBS. Cells were fixed by the dropwise addition of 0.8 mL of cold (4°C) ethanol followed by incubation at 4°C for a minimum of 24 h. After fixation, the cells were pelleted and resuspended in 1 mL of PBS containing 50 µg/mL of propidium iodide (Sigma, P4170) and 10 µg/mL of RNase A (NEB, T3018). The cells were incubated for 30 min at 22°C and then subjected to flow cytometry analysis.
For cell apoptosis analysis, the dissociated cells were stained with Annexin V Conjugate (ThermoFisher, A23202) according to the manufacturer's instructions. The cells were then subjected to flow cytometry analysis.

**Instrument**

Cytek Aurora Spectral Cell analyzer

**Software**

The fcs files were processed with FlowJo

**Cell population abundance**

The filter retained about 40% of the original cells for cell cycle analysis (~10k out of 25k) and about 80% of the original cells for cell apoptosis analysis (~50k out of 60k).

**Gating strategy**

For cell cycle analysis, the debris were removed by filtering FSC-A and SSC-A values higher than 200000 and 600000, respectively. The singlets were further selected by filtering FSC-A and FSC-H with FSC-H value higher than 300000.
For cell apoptosis analysis, the debris were removed by filtering FSC-A and SSC-A values higher than 200000 and 200000, respectively. The singlets were further selected by filtering FSC-A and FSC-H with FSC-H value higher than 100000.

☐ Tick this box to confirm that a figure exemplifying the gating strategy is provided in the Supplementary Information.

