## [Peer Review File · Nature Cell Biology]

Phase-Separated NDF-FACT Condensates Facilitate Transcription Elongation on Chromatin

Corresponding Author: Dr Jia Fei

Version 0:

Decision Letter:

*Please delete the link to your author homepage if you wish to forward this email to co-authors.

Dear Dr Fei,

Your manuscript, "Phase-Separated NDF-FACT Condensates Facilitate Transcription Elongation on Chromatin.", has now been seen by 3 referees, who are experts in chromatin remodeling and transcriptional elongation (referee 1); single nucleosome imaging and phase separation (referee 2); and optical tweezers (referee 3). As you will see from their comments (attached below) they find this work of potential interest, but have raised substantial concerns, which in our view would need to be addressed with considerable revisions before we can consider publication in Nature Cell Biology.

Nature Cell Biology editors discuss the referee reports in detail within the editorial team, including the chief editor, to identify key referee points that should be addressed with priority, and requests that are overruled as being beyond the scope of the current study. To guide the scope of the revisions, I have listed these points below. We are committed to providing a fair and constructive peer-review process, so please feel free to contact me if you would like to discuss any of the referee comments further.

Please address all of the reviewer points, taking particular care to further develop with regard to:

- 1) experiments to assess condensate properties.
 - 2) mechanism, including the role of catalytic activity, effect of size, and how the clusters modulate elongation
-) All other referee concerns pertaining to strengthening existing data, providing controls, methodological details, clarifications and textual changes, should also be addressed.
-) Finally please pay close attention to our guidelines on statistical and methodological reporting (listed below) as failure to do so may delay the reconsideration of the revised manuscript. In particular please provide:
- a Supplementary Figure including unprocessed images of all gels/blots in the form of a multi-page pdf file. Please ensure that blots/gels are labeled and the sections presented in the figures are clearly indicated.
 - a Supplementary Table including all numerical source data in Excel format, with data for different figures provided as different sheets within a single Excel file. The file should include source data giving rise to graphical representations and statistical descriptions in the paper and for all instances where the figures present representative experiments of multiple independent repeats, the source data of all repeats should be provided.

We would be happy to consider a revised manuscript that would satisfactorily address these points, unless a similar paper is published elsewhere, or is accepted for publication in Nature Cell Biology in the meantime.

- ensure that it conforms to our format instructions and publication policies (see below and <https://www.nature.com/nature/for-authors>).
- provide a point-by-point rebuttal to the full referee reports verbatim, as provided at the end of this letter.

- provide the completed Reporting Summary (found here <https://www.nature.com/documents/nr-reporting-summary.pdf>). This is essential for reconsideration of the manuscript will be available to editors and referees in the event of peer review. For more information see <http://www.nature.com/authors/policies/availability.html> or contact me.

Nature Cell Biology is committed to improving transparency in authorship. As part of our efforts in this direction, we are now requesting that all authors identified as 'corresponding author' on published papers create and link their Open Researcher and Contributor Identifier (ORCID) with their account on the Manuscript Tracking System (MTS), prior to acceptance. ORCID helps the scientific community achieve unambiguous attribution of all scholarly contributions. You can create and link your ORCID from the home page of the MTS by clicking on 'Modify my Springer Nature account'. For more information please visit www.springernature.com/orcid.

This journal strongly supports public availability of data. Please place the data used in your paper into a public data repository, or alternatively, present the data as Supplementary Information. If data can only be shared on request, please explain why in your Data Availability Statement, and also in the correspondence with your editor. Please note that for some data types, deposition in a public repository is mandatory - more information on our data deposition policies and available repositories appears below.

Link Redacted

We would like to receive a revised submission within six months.

We hope that you will find our referees' comments, and editorial guidance helpful. Please do not hesitate to contact me if there is anything you would like to discuss.

Best wishes,

Angela Parrish

Angela R Parrish, PhD
Locum Senior Editor
Nature Cell Biology

Reviewers' Comments:

Reviewer #1 (Remarks to the Author):

Phase-Separated NDF-FACT Condensates Facilitate Transcription Elongation on Chromatin
Ziwei Li et al.

Overall, this study provides a comprehensive investigation into the potential existence of NDF-FACT condensates and their biochemical and cellular functional roles. It was a pleasure to review this article, which employs diverse approaches, including in vitro cell-based systems and biochemical/biophysical assays, to substantiate the authors' hypothesis. The experiments were meticulously designed, with careful attention to detail, and were thoroughly analyzed.

As I read through the article, several questions and points for clarification arose, which I outline below:

- Evolutionary Conservation of NDF-FACT Condensates

Given that both FACT and NDF are highly conserved across eukaryotes, have you examined or considered the evolutionary conservation of their ability to form condensates and function in this manner? Investigating whether this phenomenon is

preserved across different species could provide valuable insights into its biological significance.

- **Impact of Condensate Formation on NDF and FACT Activity**

Authors propose that NDF and FACT form condensates via interactions between their IDRs, particularly through the K161 and R162 residues of NDF. A key question is whether the formation of condensates itself enhances the activity of NDF and FACT. If one of these proteins were catalytically inactive, would the other still exhibit enhanced activity? Or is it necessary for both proteins to be functionally active and capable of forming condensates to achieve synergistically enhanced transcription elongation activity? A discussion of these mechanistic aspects would further clarify their functional interplay.

- **Clarification of NDF Abbreviation in the Abstract**

It is recommended to annotate the full name of NDF (nucleosome dissociation factor) in the abstract to ensure clarity for readers unfamiliar with the abbreviation.

- **Rationale for Selecting FACT Over Other Interacting Proteins**

From the tandem-affinity purification results (Fig. 1A), several other proteins, including Rad21, p15, and PAF1, were identified in addition to FACT. Did you investigate whether these proteins also have the capacity to form condensates with NDF? Furthermore, providing a more detailed rationale for selecting FACT as the primary focus of your study would strengthen the justification for this direction and enhance the clarity of your approach.

- **Classification of NDF-FACT Condensates as “Gel-like”**

I am uncertain whether the NDF-FACT condensates can be classified as “gel-like” and instead may resemble a more “solid-like” state, given that they did not exhibit fusion, showed a low FRAP recovery rate, and demonstrated reversibility only at high salt concentrations (Fig. 2D). To further characterize their material properties, it would be valuable to investigate whether these condensates exhibit distinct structural or dynamic changes in response to temperature variations. Such analyses could provide deeper insights into their phase behavior and biophysical state.

- **Functional Implications of NDF-Spt16 Co-occupancy in ChIP-seq**

Based on the ChIP-seq results, authors observed a co-occupancy trend between NDF and Spt16, despite their binding to distinct regions within genes, and subsequently categorized genes into four clusters. However, the biological significance of these clusters is not entirely clear. Could you elaborate on their functional implications? Additionally, do genes with high NDF and Spt16 levels exhibit any specific characteristics, such as higher GC content, promoter-proximal pausing, or shorter nucleosome-depleted regions (NDRs)? A deeper characterization of these clusters may provide further mechanistic insights into how NDF and FACT regulate transcription.

- **Justification for the Use of iPSC Cells**

Regarding cellular models, is there a specific rationale for selecting iPSC cells? I wonder that authors have observed any defects in pluripotency maintenance or differentiation potential when condensate formation is disrupted or upon depletion of NDF and Spt16? Additionally, is there any evidence supporting the significance of the NDF-Spt16 interaction in stem cell function? If condensate formation plays a role in transcriptional regulation during differentiation, it would be interesting to assess whether perturbing NDF-FACT condensates affects lineage-specific gene expression programs or chromatin accessibility.

- **Interpreting PRO-seq Results and Transcription Elongation**

In Figure 7(e), authors propose that NDF-FACT phase separation specifically impairs the elongation phase of transcription through chromatin, based on PRO-seq results. However, the increased PRO-seq signal at the promoter-proximal region alone does not necessarily indicate a direct impairment of elongation, as it could also result from enhanced RNAPII recruitment or defects in pause-release mechanisms. To further substantiate the claim that NDF-FACT condensates affect the elongation phase, it would be valuable to examine whether there is a distinct PRO-seq pattern in long genes compared to shorter ones. If long genes exhibit greater RNAPII accumulation in gene bodies or a shift in RNAPII distribution, this could provide stronger evidence that NDF-FACT condensates influence transcription elongation.

- **Comparison with Other Transcription-Related Condensates**

How does NDF-FACT phase separation compare with other known transcription-related condensates, such as Mediator or Pol II condensates? Are their biophysical properties distinct? A comparative analysis highlighting differences in molecular dynamics, phase behavior, and functional consequences would strengthen the discussion.

- **Figure caption misrepresentation**

- In Fig. 7(E), the analysis presented is a metagene analysis, not a heatmap. Please revise the figure caption accordingly.

- In Fig. 6(I), Fig. 6(K), and Fig. 7(I), the description is confusing, as metagene analysis is performed on all genes, not just selected loci. Clarifying this distinction would enhance accuracy.

- In Fig. 6(M) and Fig. 7(K), the number of transcriptionally inactive genes is missing. Providing this information would improve the interpretability of the data.

- In Fig. S8(I), the p-value representation is missing from the figure caption. Including this detail would help contextualize the statistical significance of the findings.

- In the Methods section, there is no description of the total number of genes used in the metagene analysis or the criteria used to define transcriptionally inactive genes. Clarifying these aspects would strengthen the reproducibility of the analysis. This study provides new insights into the mechanistic action of FACT and its association with NDF, supported by well-designed experiments and thorough analysis. Addressing these points would further enhance the comprehensibility, accuracy, and validity of the study.

Reviewer #2 (Remarks to the Author):

How FACT is involved in transcription remains an intriguing question in cell biology, especially in how Pol II overcomes chromatin barriers by nucleosomes in the cell. To approach this issue, Li et al. elegantly combined biochemistry, single-molecule biophysics assays, cell biology, and genomics. The authors showed that NDF-FACT condensates form on actively transcribing Pol II and move along chromatin during transcription elongation, enhancing transcription-coupled

nucleosome disassembly and Pol II progression. When they disrupted the condensates in human stem cells, they observed reduced FACT chromatin binding and subsequent defects in transcription—similar to the effects caused by FACT depletion. The authors claimed that NDF-mediated phase separation is a key mechanism for FACT's cellular functions. Although the story is exciting and thought-provoking, there are some serious gaps between the experimental data and the conclusions drawn. My specific comments are the following:

Major points:

- 1) Fig. 2E–F. The results seem to suggest that the condensates are more solid-like structures rather than liquid-like. Consistently, no clear fusion events were observed, and the condensates showed resistance to 1,6-hexanediol treatment (Fig. S2F). There is no convincing evidence to support that these condensates are bona fide phase-separated droplets. The authors should reconsider this central aspect of the story.
- 2) Fig. 2H. Exciting data. Time-lapse imaging at higher resolution would provide additional insights into the process.
- 3) Fig. 2I. Again, exciting data. I wondered whether the transcribed RNA is located inside the condensates, attached to their surface, or exported from them. This might provide mechanistic insight into the process.
- 4) Figs. 3D and S3B. Do small and large condensates exhibit similar transcriptional competency?
- 5) Fig. 5A. It would be informative to investigate whether Ser2 or Ser5 phosphorylation of RPB1 colocalizes with NDF and Spt16 foci.
- 6) Figs. 5B–C and 6C–D. Since 1,6-hexanediol affects chromatin organization, these results are not very informative. A discussion is needed here (PMID: 33536240).
- 7) Page 12. "...we hypothesized that condensate formation might be critical for efficiently recruiting both factors to chromatin in cells."
I was a bit confused about the authors' point. Does NDF/FACT leave the condensates during transcription elongation, or does elongation occur on the surface of the condensates?
- 8) Fig. S8E–I. These are interesting data, but drawing conclusions from the Curaxin experiment is difficult. Curaxin is an intercalator, so it can alter chromatin structure—as seen in the DAPI image in Fig. S8H (condensed?)—which may indirectly affect the behavior of various chromatin-associated proteins.

Minor points:

- 1) For general readers, it might be helpful to briefly mention what NDF is in the Introduction.
- 2) The model scheme in Fig. S10 is important and would be better placed as a main figure.

Reviewer #3 (Remarks to the Author):

Li et al presents a tour-de-force investigation of how Pol II transcription is facilitated by FACT-NDF condensates. I recommend publication, but I believe the manuscript would be strengthened: (1) if a cartoon of the proposed FACT-NDF-facilitated transcription were included in the introduction – something like a simplified version of Fig. S10B; and (2) if the manuscript addressed the size of the FACT-NDF condensates involved in in vivo transcription – what limitations on the size of the FACT-NDF condensate droplets observed in vivo are established by imaging of the coincident NDF and FACT puncta? It would be useful to specify this information in the manuscript. In addition, (3) I wonder whether a recent bioRxiv, namely <https://www.biorxiv.org/content/10.1101/2024.08.26.609758v1>, provides a valuable theoretical perspective on the experimental results reported in this manuscript?

Methods should be written concisely, but should contain all elements necessary to allow interpretation and replication of the results. As a guideline, Methods sections typically do not exceed 3,000 words. The Methods should be divided into subsections listing reagents and techniques. When citing previous methods, accurate references should be provided and any alterations should be noted. Information must be provided about: antibody dilutions, company names, catalogue numbers and clone numbers for monoclonal antibodies; sequences of RNAi and cDNA probes/primers or company names and catalogue numbers if reagents are commercial; cell line names, sources and information on cell line identity and authentication. Animal studies and experiments involving human subjects must be reported in detail, identifying the committees approving the protocols. For studies involving human subjects/samples, a statement must be included confirming that informed consent was obtained. Statistical analyses and information on the reproducibility of experimental results should be provided in a section titled "Statistics and Reproducibility".

All Nature Cell Biology manuscripts submitted on or after March 21 2016 must include a Data availability statement as a separate section after Methods but before references, under the heading "Data Availability". For Springer Nature policies on data availability see <http://www.nature.com/authors/policies/availability.html>; for more information on this particular policy see <http://www.nature.com/authors/policies/data/data-availability-statements-data-citations.pdf>. The Data availability statement should include:

- Accession codes for primary datasets (generated during the study under consideration and designated as "primary accessions") and secondary datasets (published datasets reanalysed during the study under consideration, designated as "referenced accessions"). For primary accessions data should be made public to coincide with publication of the manuscript. A list of data types for which submission to community-endorsed public repositories is mandated (including sequence, structure, microarray, deep sequencing data) can be found here <http://www.nature.com/authors/policies/availability.html#data>.
- Unique identifiers (accession codes, DOIs or other unique persistent identifier) and hyperlinks for datasets deposited in an

approved repository, but for which data deposition is not mandated (see here for details <http://www.nature.com/sdata/data-policies/repositories>).

- At a minimum, please include a statement confirming that all relevant data are available from the authors, and/or are included with the manuscript (e.g. as source data or supplementary information), listing which data are included (e.g. by figure panels and data types) and mentioning any restrictions on availability.
- If a dataset has a Digital Object Identifier (DOI) as its unique identifier, we strongly encourage including this in the Reference list and citing the dataset in the Methods.

We recommend that you upload the step-by-step protocols used in this manuscript to [protocols.io](https://www.protocols.io). More details can be found at <https://www.protocols.io/help/publish-articles>.

All imaging data should be accompanied by scale bars, which should be defined in the legend.

Cropped images of gels/blots are acceptable, but need to be accompanied by size markers, and to retain visible background signal within the linear range (i.e. should not be saturated). The boundaries of panels with low background have to be demarked with black lines. Splicing of panels should only be considered if unavoidable, and must be clearly marked on the figure, and noted in the legend with a statement on whether the samples were obtained and processed simultaneously. Quantitative comparisons between samples on different gels/blots are discouraged; if this is unavoidable, it should only be performed for samples derived from the same experiment with gels/blots were processed in parallel, which needs to be stated in the legend.

FIGURE LEGENDS – must not exceed 350 words for each figure to allow fit on a single printed NCB page together with the figure. They must include a brief title for the whole figure, and short descriptions of each panel with definitions of the symbols

used, but without detailing methodology.

The total number of Supplementary Figures (not including the “unprocessed scans” Supplementary Figure) should not exceed the number of main display items (figures and/or tables (see our Guide to Authors and March 2012 editorial <http://www.nature.com/ncb/authors/submit/index.html#supinfo>; <http://www.nature.com/ncb/journal/v14/n3/index.html#ed>). No restrictions apply to Supplementary Tables or Videos, but we advise authors to be selective in including supplemental data.

GUIDELINES FOR EXPERIMENTAL AND STATISTICAL REPORTING

REPORTING REQUIREMENTS – We are trying to improve the quality of methods and statistics reporting in our papers. To that end, we are now asking authors to complete a reporting summary that collects information on experimental design and reagents. The Reporting Summary can be found here <https://www.nature.com/documents/nr-reporting-summary.pdf>. If you would like to reference the guidance text as you complete the template, please access these flattened versions at <http://www.nature.com/authors/policies/availability.html>.

We strongly recommend the presentation of source data for graphical and statistical analyses as a separate Supplementary Table, and request that source data for all independent repeats are provided when representative experiments of multiple independent repeats, or averages of two independent experiments are presented. This supplementary table should be in Excel format, with data for different figures provided as different sheets within a single Excel file. It should be labelled and numbered as one of the supplementary tables, titled “Statistics Source Data”, and mentioned in all relevant figure legends.

Version 1:

Decision Letter:

Our ref: NCB-A57425A

17th July 2025

Dear Dr. Fei,

Thank you for submitting your revised manuscript "Phase-Separated NDF-FACT Condensates Facilitate Transcription Elongation on Chromatin." (NCB-A57425A). It has now been seen by the original referees and their comments are below. The reviewers find that the paper has improved in revision, and therefore we'll be happy in principle to publish it in Nature Cell Biology, pending minor revisions to satisfy the referees' final requests and to comply with our editorial and formatting guidelines.

Thank you again for your interest in Nature Cell Biology Please do not hesitate to contact me if you have any questions.

Sincerely,

Angela R Parrish, PhD
Locum Senior Editor
Nature Cell Biology

Reviewer #1 (Remarks to the Author):

The authors have incorporated substantial new data that effectively addresses the concerns previously raised. These additions considerably strengthen the manuscript. The study presents novel and highly significant conclusions that meaningfully advance our understanding of the field.

Reviewer #2 (Remarks to the Author):

The authors have adequately addressed my previous comments, and the revised manuscript has been improved. I have no further concerns.

Version 2:

Decision Letter:

Dear Dr Fei,

I am pleased to inform you that your manuscript, "Phase-Separated NDF-FACT Condensates Facilitate Transcription Elongation on Chromatin", has now been accepted for publication in Nature Cell Biology.

Please note that *Nature Cell Biology* is a Transformative Journal (TJ). Authors may publish their research with us through the traditional subscription access route or make their paper immediately open access through payment of an article-processing charge (APC). Authors will not be required to make a final decision about access to their article until it has been accepted. [Find out more about Transformative Journals](https://www.springernature.com/gp/open-research/transformative-journals)

Authors may need to take specific actions to achieve compliance with funder and institutional open access mandates. If your research is supported by a funder that requires immediate open access (e.g. according to [Plan S principles](https://www.springernature.com/gp/open-science/plan-s-compliance) or the [NIH public access policy](https://www.springernature.com/gp/open-science/us-federal-agency-compliance)) then you should select the gold OA route, and we will direct you to the compliant route where possible. Because authors warrant under our subscription licensing terms that they haven't committed to licensing any version of their article under a licence inconsistent with the terms of our agreement – including the applicable embargo period – publication under the subscription model isn't suitable for authors whose funders require no embargo.

If you have not already done so, we strongly recommend that you upload the step-by-step protocols used in this manuscript to protocols.io (<https://protocols.io>), an open online resource that allows researchers to share their detailed experimental know-how. All uploaded protocols are made freely available and are assigned DOIs for ease of citation. Protocols and Nature Portfolio journal papers in which they are used can be linked to one another, and this link is clearly and prominently visible in the online versions of both. Authors who performed the specific experiments can act as primary authors for the Protocol as they will be best placed to share the methodology details, but the Corresponding Author of the present research paper should be included as one of the authors. By uploading your Protocols onto protocols.io, you are enabling researchers to more readily reproduce or adapt the methodology you use, as well as increasing the visibility of your protocols and papers. You can also establish a dedicated workspace to collect your lab Protocols. Further information can be found at <https://www.protocols.io/help/publish-articles>.

Nature Cell Biology encourages authors presenting evidence for cell, biological, molecular, and genetic interactions to consider communicating these findings using Biofactoid (<https://biofactoid.org/>). This tool helps users share a searchable representation of interactions (e.g. binding, gene expression, post-translational modification) between genes, gene products, or chemicals. Information added to Biofactoid, with author attribution, is shared on social media and public databases, such as Pathway Commons, where it can be discovered and analyzed in the context of a large and growing corpus of knowledge.

With kind regards,

Angela R Parrish, PhD
Locum Senior Editor
Nature Cell Biology

** Visit the Springer Nature Editorial and Publishing website at http://editorial-jobs.springernature.com?utm_source=ejp_NCB_email&utm_medium=ejp_NCB_email&utm_campaign=ejp_NCB for more information about our career opportunities. If you have any questions please click [here](mailto:editorial.publishing.jobs@springernature.com).

Reviewers' Comments:

Reviewer #1 (Remarks to the Author):

Phase-Separated NDF-FACT Condensates Facilitate Transcription Elongation on Chromatin

Ziwei Li et al.

Overall, this study provides a comprehensive investigation into the potential existence of NDF-FACT condensates and their biochemical and cellular functional roles. It was a pleasure to review this article, which employs diverse approaches, including in vitro cell-based systems and biochemical/biophysical assays, to substantiate the authors' hypothesis. The experiments were meticulously designed, with careful attention to detail, and were thoroughly analyzed. As I read through the article, several questions and points for clarification arose, which I outline below:

• Evolutionary Conservation of NDF-FACT Condensates:

Given that both FACT and NDF are highly conserved across eukaryotes, have you examined or considered the evolutionary conservation of their ability to form condensates and function in this manner? Investigating whether this phenomenon is preserved across different species could provide valuable insights into its biological significance.

The reviewer raised an excellent question regarding the evolutionary conservation of NDF-FACT phase separation. To address this, we tested yeast homologous proteins by purifying recombinant *Saccharomyces cerevisiae* FACT complex (Spt16+Pob3) (NEW Extended Data Fig. 2e) and yeast NDF (Pdp3, *Fei et al., Genes Dev* 2022) from bacteria.

Consistent with human proteins, individual yeast proteins alone did not form condensates, but when mixed together, we observed liquid droplet formation under the microscope (NEW Extended Data Fig. 2f). These results suggest that NDF-FACT condensate formation is evolutionarily conserved from yeast to humans, supporting the biological significance of this transcriptional mechanism.

This revised content has been added to the manuscript in lines 170-175.

8% SDS-PAGE analysis of yeast FACT purified from bacteria. The protein gel was visualized using Coomassie Blue staining.

Droplet formation assay with 1 µM purified proteins in droplet formation buffer. Upper panel: Nhp6A, Pdp3 (yeast NDF), and yFACT (ySpt16 + Pob3) do not form condensates individually. Lower panel: Pdp3 forms condensates with yFACT.

Response to the comments of the reviewers (NCB-A57425A)

• Impact of Condensate Formation on NDF and FACT Activity

Authors propose that NDF and FACT form condensates via interactions between their IDRs, particularly through the K161 and R162 residues of NDF. A key question is whether the formation of condensates itself enhances the activity of NDF and FACT. If one of these proteins were catalytically inactive, would the other still exhibit enhanced activity? Or is it necessary for both proteins to be functionally active and capable of forming condensates to achieve synergistically enhanced transcription elongation activity? A discussion of these mechanistic aspects would further clarify their functional interplay.

We thank the reviewer for asking this excellent question regarding whether catalytic activity is required for condensate formation and synergistic transcriptional effects. To address this, we tested NDF truncation mutants with altered catalytic activity.

We found that the NDF_F1 truncate, which lacks the PWWP domain and has significantly reduced nucleosome destabilization activity, can still form condensates with FACT (old Extended Data Fig. 4d) and incorporate Cy2-labeled nucleosomes into these condensates (**NEW Extended Data Fig. 4k**). However, despite retaining condensate-forming ability, NDF_F1 does not exhibit synergistic transcriptional enhancement with FACT (**NEW Fig. 4i**).

Droplet formation visualized with various NDF proteins and Spt16-mCherry at 2 μ M in droplet formation buffer (100 mM NaCl, no PEG). Scale bar=10 μ m. Note: While the N-terminal PWWP domain interacts with FACT in vitro, proteins containing only the PWWP domain (R4 and R5) do not form condensates.

Microscopic images show the incorporation of Cy2-labeled nucleosomes into pre-formed (NDF_F1 + Spt16-mCherry) condensates at room temperature. Protein concentrations were 1 μ M. Scale bar = 10 μ m.

Nucleosome transcription assay with 0.2 μ M NDF and NDF_F1 truncates and FACT with 0.3 μ M TFIIIS. Transcription reactions were performed with 0.25 mM NTPs for 10 min at room temperature.

These results demonstrate that:

- 1) condensate formation per se does not require full catalytic activity, and
- 2) both condensate formation AND catalytic activity are necessary for synergistic transcription elongation.

Simply increasing local protein concentration through phase separation is insufficient—the intrinsic biochemical activities of both proteins are essential for their functional cooperation. This suggests that condensates provide a platform for concentrating active enzymes, but the catalytic functions themselves drive the synergistic transcriptional effects.

This revised content has been added to the manuscript in lines 277-282.

- Clarification of NDF Abbreviation in the Abstract

It is recommended to annotate the full name of NDF (nucleosome dissociation factor) in the abstract to ensure clarity for readers unfamiliar with the abbreviation.

We have revised the abstract accordingly.

This revised content has been added to the manuscript in line 56.

- Rationale for Selecting FACT Over Other Interacting Proteins

From the tandem-affinity purification results (Fig. 1A), several other proteins, including Rad21, p15, and PAF1, were identified in addition to FACT. Did you investigate whether these proteins also have the capacity to form condensates with NDF? Furthermore, providing a more detailed rationale for selecting FACT as the primary focus of your study would strengthen the justification for this direction and enhance the clarity of your approach.

We thank the reviewer for this important question. We have not yet tested whether other identified proteins (Rad21, p15, PAF1) can form condensates with NDF, but this represents an interesting avenue for future investigation. It should be noted that our tandem-affinity purification was performed under 0.2 M NaCl conditions, which disrupts some condensate formation (see, for example, Extended Data Fig. 2b), so it is possible that the mass spectrometry primarily identified stable protein-protein interactions, instead of condensates.

We specifically focused on FACT because we had purified recombinant human FACT proteins available in our laboratory, so we tested it first. Additionally, both NDF and FACT possess similar biochemical activities in facilitating RNA Pol II transcription through nucleosomal barriers in vitro, making their potential cooperation of particular interest.

This revised content has been added to the manuscript in lines 133-134.

- Classification of NDF-FACT Condensates as “Gel-like”

I am uncertain whether the NDF-FACT condensates can be classified as “gel-like” and instead may resemble a more “solid-like” state, given that they did not exhibit fusion, showed a low FRAP recovery rate, and demonstrated reversibility only at high salt concentrations (Fig. 2D). To further characterize their material properties, it would be valuable to investigate whether these condensates exhibit distinct structural or dynamic changes in response to temperature

Response to the comments of the reviewers (NCB-A57425A)

variations. Such analyses could provide deeper insights into their phase behavior and biophysical state.

We thank the reviewers for this excellent question regarding the biophysical properties of NDF-FACT condensates and their classification.

The distinction between gel-like and solid-like condensates is well-established in the literature based on specific biophysical criteria. According to foundational studies [for example, PMID: 28283059 (stress granules) and PMID: 27471966(Balbiani body)], solid-like condensates are characterized by minimal FRAP recovery of less than 20% fluorescence recovery even after one hour, and extreme salt resistance where structural integrity is maintained under even 2M NaCl treatment for Balbiani bodies.

However, our NDF-FACT condensates exhibit distinctly different properties: 30-40% FRAP recovery after 200 seconds (**Extended Data Fig. 2j**) and sensitivity to 0.4M NaCl with reversible dissolution (**Fig. 2d**). Therefore, we classified the NDF-FACT condensates as gel-like instead of solid-like.

The rationale of this classification has been added to lines 176-189.

Following the reviewers' suggestion, we performed temperature sensitivity experiments showing that preformed condensates respond to elevated temperature, with reduced size at 37°C while maintaining droplet-like morphology (**NEW Extended Data Fig. 2i**). This temperature responsiveness indicates that the condensates are dynamic assemblies rather than stable solid aggregates.

This revised content has been added to the manuscript in lines 179-182.

Condensates were formed with 1 µM proteins and incubated at the indicated temperatures before imaging and size distribution analysis. 4 °C (n=561), 25 °C (n=1,481) and 37 °C (n=3736).

Altogether, these properties clearly position NDF-FACT condensates as gel-like rather than solid-like structures. These intermediate dynamics support biological function while maintaining necessary structural integrity for transcriptional regulation.

- Functional Implications of NDF-Spt16 Co-occupancy in ChIP-seq

Based on the ChIP-seq results, authors observed a co-occupancy trend between NDF and Spt16, despite their binding to distinct regions within genes, and subsequently categorized genes into four clusters. However, the biological significance of these clusters is not entirely clear. Could you elaborate on their functional implications? Additionally, do genes with high NDF and Spt16 levels exhibit any specific characteristics, such as higher GC content, promoter-proximal pausing, or shorter nucleosome-depleted regions (NDRs)? A deeper characterization of these clusters may provide further mechanistic insights into how NDF and FACT regulate transcription.

We thank the reviewer for raising this important question. Our k-means clustering approach was designed to systematically analyze the relationship between NDF and FACT distribution across gene bodies. Since FACT has dual functions at both promoters and gene bodies, direct quantification of ChIP-seq signal correlations across entire genes presented technical challenges, making clustering analysis a more appropriate approach.

We have further analyzed the genomic features of genes in different clusters. As shown in NEW Extended Data Fig. 7a-c, we examined GC content, nucleosome-depleted region (NDR) lengths, and pausing indices across the four gene clusters. While these analyses reveal some trends, the large number of genes within each cluster and the heterogeneity of gene features make it challenging to draw definitive mechanistic conclusions. Additionally, we currently lack validated assays to quantify condensate levels at individual genes on chromatin, preventing us from directly correlating gene features with condensate abundance.

Genomic features of NDF-Spt16 co-occupancy gene clusters. Box plots showing (a) GC content, (b) nucleosome-depleted region (NDR) lengths, and (c) pausing indices across four gene clusters defined by k-means clustering of NDF and Spt16 ChIP-seq data. Mean values for each cluster are labeled next to box plots.

Nevertheless, we have included these data as Extended Data because we believe this initial characterization may provide a valuable foundation for future studies as new techniques are developed to better quantify and analyze transcriptional condensates at the single-gene level.

This revised content has been added to the manuscript in lines 345-348.

- Justification for the Use of iPSC Cells

Regarding cellular models, is there a specific rationale for selecting iPSC cells? I wonder that authors have observed any defects in pluripotency maintenance or differentiation potential when condensate formation is disrupted or upon depletion of NDF and Spt16? Additionally, is there any evidence supporting the significance of the NDF-Spt16 interaction in stem cell function? If condensate formation plays a role in transcriptional regulation during differentiation, it would be

Response to the comments of the reviewers (NCB-A57425A)

interesting to assess whether perturbing NDF-FACT condensates affects lineage-specific gene expression programs or chromatin accessibility.

We thank the reviewer for this important question regarding our choice of cellular model system and the functional significance of NDF-FACT condensates in stem cell biology.

FACT exhibits minimal expression in differentiated somatic cells but shows high expression in stem cells and cancer cells (for example, PMID: 31616795). While cancer cell lines could serve as alternative models, they possess heterogeneous genetic backgrounds that may not reflect physiologically relevant FACT function. Since NDF is also highly expressed in both stem cells and cancer cells (Fei et al., *Genes Dev.* 2018), we selected human iPSCs as our model system to study NDF-FACT condensates in a defined cellular context.

The reviewer's question about condensate function in pluripotency and differentiation is excellent. Since both NDF and FACT are essential for stem cell viability, direct depletion is lethal (Extended Data Fig. 7j-l for NDF, Extended Data Fig. 8b for Spt16). However, the NDF_K161A mutation provides an ideal system to study condensate function, as it impairs condensate formation (~25% reduction) while maintaining cell viability.

We performed comparative differentiation of NDF-WT and NDF-K161A cells into cardiac lineages, generating cardiac progenitor cells (CPC) and EPIC-derived fibroblast-like cells (EPIC-FB). Successful differentiation was confirmed by expression of lineage markers such as Vimentin. Both cell lines responded to differentiation conditions, as evidenced by global gene expression changes. Comparing NDF-WT to NDF-K161A cells, we observed mild differences at the CPC stage, with only modest numbers of genes significantly affected. However, at the EPIC-FB stage, differences were more pronounced, with thousands of genes showing altered expression. Interestingly, some lineage-specific genes such as VIM and S100A4 were not significantly affected, suggesting that FACT condensate formation may not be critical for all lineage-specific programs. However, condensates clearly play important roles in regulating large groups of genes

during differentiation. These data are included in the revised manuscript as NEW Extended Data Fig. 9b-f.

This revised content has been added to the manuscript in lines 422-427.

b, Immunofluorescence staining for Vimentin in undifferentiated iPSCs and differentiated EPIC-FB cells, confirming successful cardiac fibroblast differentiation. Scale bar = 20 μm.
c, Volcano plots showing differential gene expression in NDF-WT cells during differentiation. Left: iPSC vs cardiac progenitor cell (CPC) stage. Right: iPSC vs EPIC-derived fibroblast-like cell (EPIC-FB) stage. Significantly upregulated and downregulated genes (both are shown as red) are highlighted.
d, Volcano plots showing differential gene expression in NDF-K161A cells during differentiation. Left: iPSC vs CPC stage. Right: iPSC vs EPIC-FB stage. Significantly upregulated and downregulated genes (both are shown as red) are highlighted.
e, Volcano plot comparing gene expression between NDF-K161A and NDF-WT cells at the cardiac progenitor cell (CPC) stage. Significantly upregulated and downregulated genes (both are shown as red) are highlighted.
f, Volcano plot comparing gene expression between NDF-K161A and NDF-WT cells at the EPIC-FB stage. Significantly upregulated and downregulated genes (both are shown as red) are highlighted.

• Interpreting PRO-seq Results and Transcription Elongation

In Figure 7(e), authors propose that NDF-FACT phase separation specifically impairs the elongation phase of transcription through chromatin, based on PRO-seq results. However, the increased PRO-seq signal at the promoter-proximal region alone does not necessarily indicate a direct impairment of elongation, as it could also result from enhanced RNAPII recruitment or defects in pause-release mechanisms. To further substantiate the claim that NDF-FACT condensates affect the elongation phase, it would be valuable to examine whether there is a distinct PRO-seq pattern in long genes compared to shorter ones. If long genes exhibit greater RNAPII accumulation in gene bodies or a shift in RNAPII distribution, this could provide stronger evidence that NDF-FACT condensates influence transcription elongation.

We agree with the reviewer that PRO-seq may not truly reflect elongation status, although PRO-seq has been widely used in the field. Therefore, we performed **three additional experiments** to directly assess elongation dynamics:

First, we performed TT-seq to measure nascent RNA synthesis in both WT and NDF_K161A cells. We observed that a large number of genes showed decreased nascent RNA synthesis rates in NDF_K161A cells, as shown in the boxplot below (NEW Extended Data Fig. 9g).

NEW Extended
Data Fig. 9b

Second, we performed TT-seq coupled with DRB treatment, which measures ribonucleotide incorporation into nascent RNA in a time-dependent manner. DRB treatment arrests RNA polymerase II at promoter-proximal pausing sites globally. Since DRB arrest is reversible, upon DRB removal from culture media, Pol II resumes elongation, allowing side-by-side comparison of nascent RNA generation rates between NDF-WT and NDF_K161A cells. As shown in NEW Fig. 7f-h, nascent RNA synthesis levels upon DRB release were lower in NDF_K161A cells for many long genes. Additionally, elongation rates were significantly affected by the K161A mutation.

NEW Fig. 7 f-h

New Fig. 7f-h. Impaired transcription elongation in NDF-K161A cells. TT-seq analysis following DRB release shows (f) reduced nascent RNA synthesis at representative loci, (g) decreased transcript levels across gene bodies of 3709 genes that are 30–300 kb (metagene analysis), and (h) significantly slower RNA polymerase II elongation rates in NDF-K161A cells compared to NDF-WT cells. Statistics: Two-way ANOVA, $p < 0.0001$

Third, as suggested by the reviewer, we analyzed whether nascent RNA synthesis is more impaired in longer versus shorter genes. Unexpectedly, we found that shorter genes were more severely affected when NDF-FACT condensates were disrupted (**NEW Extended Data Fig. 9h**).

NEW Extended Data Fig. 9h

h, Nascent RNA synthesis impairment (NDF_K161A/WT fold change) stratified by gene length quartiles (Q1-Q4, shortest to longest). Only genes with at least 1 read were analyzed. Shorter genes showed significantly greater impairment compared to longer genes (* $P < 0.05$, **** $P < 0.0001$).

This finding suggests that the relationship between gene length and condensate dependency may be more complex than initially anticipated. Several factors could contribute to this observation: (1) shorter genes may have higher condensate density per unit length, (2) longer genes may have redundant elongation mechanisms, or (3) condensate distribution may not correlate linearly with gene length. Since current techniques cannot quantify condensate levels on individual genes, the mechanisms underlying this observation require further investigation.

Overall, while the gene length analysis revealed unexpected complexity in condensate function, our TT-seq and DRB-release experiments provide direct evidence that NDF-FACT condensates promote transcription elongation. The consistent reduction in nascent RNA synthesis and elongation rates in K161A cells demonstrates that phase separation enhances transcriptional efficiency beyond what PRO-seq alone could reveal.

This revised content has been added to the manuscript in lines 431-439.

- Comparison with Other Transcription-Related Condensates

How does NDF-FACT phase separation compare with other known transcription-related condensates, such as Mediator or Pol II condensates? Are their biophysical properties distinct? A comparative analysis highlighting differences in molecular dynamics, phase behavior, and functional consequences would strengthen the discussion.

We thank the reviewer for this excellent suggestion. We have added a brief comparative discussion to position NDF-FACT condensates within the transcriptional phase separation landscape. However, due to the large number of transcriptional condensate types in the literature and limited manuscript space, we believe a comprehensive comparative analysis would be more suitable for a dedicated review article.

This revised content has been added to the manuscript in lines 467-472.

Brief Comparative Analysis:

Biophysical Properties: NDF-FACT condensates exhibit gel-like properties (30-40% FRAP recovery) contrasting with highly dynamic RNA Pol II condensates (90% exchange in ~10 seconds) and Mediator condensates (60% exchange) (PMID: 29930094). They form at low concentrations (0.125 μ M) with moderate 1,6-hexanediol sensitivity.

Assembly Mechanisms: Unlike most transcriptional condensates that self-assemble through their own IDRs (MED1, BRD4, RNA Pol II CTD), NDF-FACT condensates require heterotypic interactions between two distinct proteins. Neither protein alone forms condensates—phase separation depends on specific NDF-Spt16 interactions, representing a unique co-dependent assembly mechanism.

Functional Specialization: NDF-FACT condensates uniquely travel processively with transcribing Pol II along chromatin, specifically facilitate nucleosome barrier traversal, and actively recruit nucleosomal substrates, unlike stationary initiation condensates or local elongation condensates.

- Figure caption misrepresentation

- In Fig. 7(E), the analysis presented is a metagene analysis, not a heatmap. Please revise the figure caption accordingly.

Corrected. Thank you.

Response to the comments of the reviewers (NCB-A57425A)

- In Fig. 6(I), Fig. 6(K), and Fig. 7(I), the description is confusing, as metagene analysis is performed on all genes, not just selected loci. Clarifying this distinction would enhance accuracy.

Corrected. Thank you.

- In Fig. 6(M) and Fig. 7(K), the number of transcriptionally inactive genes is missing. Providing this information would improve the interpretability of the data.

We thank the reviewer for this observation. There are 2,437 genes that were defined as transcriptionally inactive based on the number of total Pol II ChIP-seq reads on these genes. This number was included in the materials and methods section, but for convenience, we have now added this number to the figures.

- In Fig. S8(I), the p-value representation is missing from the figure caption. Including this detail would help contextualize the statistical significance of the findings.

Corrected. Thank you.

- In the Methods section, there is no description of the total number of genes used in the metagene analysis or the criteria used to define transcriptionally inactive genes. Clarifying these aspects would strengthen the reproducibility of the analysis.

We thank the reviewer for pointing out this omission. Transcriptionally inactive genes (n=2,437) were defined by no total Pol II ChIP-seq reads as counted by HOMER. We have now added the clarification to the Methods section.

This study provides new insights into the mechanistic action of FACT and its association with NDF, supported by well-designed experiments and thorough analysis. Addressing these points would further enhance the comprehensibility, accuracy, and validity of the study.

Reviewer #2 (Remarks to the Author):

How FACT is involved in transcription remains an intriguing question in cell biology, especially in how Pol II overcomes chromatin barriers by nucleosomes in the cell. To approach this issue, Li et al. elegantly combined biochemistry, single-molecule biophysics assays, cell biology, and genomics. The authors showed that NDF-FACT condensates form on actively transcribing Pol II and move along chromatin during transcription elongation, enhancing transcription-coupled nucleosome disassembly and Pol II progression. When they disrupted the condensates in human stem cells, they observed reduced FACT chromatin binding and subsequent defects in transcription—similar to the effects caused by FACT depletion. The authors claimed that NDF-mediated phase separation is a key mechanism for FACT's cellular functions. Although the story is exciting and thought-provoking, there are some serious gaps between the experimental data and the conclusions drawn. My specific comments are the following:

Major points:

- 1) Fig. 2E–F. The results seem to suggest that the condensates are more solid-like structures rather than liquid-like. Consistently, no clear fusion events were observed, and the condensates showed resistance to 1,6-hexanediol treatment (Fig. S2F). There is no convincing evidence to

Response to the comments of the reviewers (NCB-A57425A)

support that these condensates are bona fide phase-separated droplets. The authors should reconsider this central aspect of the story.

We thank the reviewers for this important question regarding the biophysical classification of NDF-FACT condensates. We appreciate the opportunity to provide additional clarification and data.

Recent advances in the field have established that LLPS can produce assemblies with varied material states, not exclusively liquid droplets. As established in current reviews, 'LLPS does not have to result in liquid, freely fusing assemblies; instead, biomolecular condensates can adopt a continuum of material properties' (PMID: 27554449, PMID: 22682242 and PMID: 30682370). Condensates can undergo transitions from liquid to gel to solid states, particularly when they exist at conditions deep within the two-phase regime. The absence of fusion events in our NDF-FACT condensates indicates they have undergone a gel transition, forming functionally adaptive assemblies that remain legitimate phase-separated structures.

The distinction between gel-like and solid-like condensates is well-established in the literature based on specific biophysical criteria. According to foundational studies [for example, PMID: 28283059 (stress granules) and PMID: 27471966(Balbani body)], solid-like condensates are characterized by minimal FRAP recovery of less than 20% fluorescence recovery even after one hour, and extreme salt resistance where structural integrity is maintained under even 2M NaCl treatment for Balbani bodies.

Our NDF-FACT condensates exhibit distinctly different properties: 20-40% FRAP recovery after 200 seconds (Extended Data Fig. 2j), sensitivity to 0.4M NaCl with **reversible dissolution** (Fig. 2d), and ~20% sensitivity to 1,6-hexanediol treatment (Extended Data Fig. 2h) - comparable to other established phase-separated condensates like BRD-IDR condensates (~30% sensitivity, Sabari et al., PMID: 29930091). Additionally, temperature sensitivity experiments (**NEW Extended Data Fig. 2i**) show that preformed condensates respond to elevated temperature with size reduction while maintaining droplet morphology, indicating dynamic assembly rather than solid aggregation.

These properties clearly position NDF-FACT condensates as gel-like phase-separated structures with intermediate dynamics that support biological function while maintaining structural integrity for transcriptional regulation. The rationale for this classification has been added to lines 176-189.

Condensates were formed with 1 μM proteins and incubated at the indicated temperatures before imaging and size distribution analysis. 4 $^{\circ}\text{C}$ (n=561), 25 $^{\circ}\text{C}$ (n=1,481) and 37 $^{\circ}\text{C}$ (n=3736).

2) Fig. 2H. Exciting data. Time-lapse imaging at higher resolution would provide additional insights into the process.

We thank the reviewer for this excellent suggestion regarding Fig. 2h. We attempted time-lapse imaging at higher resolution as requested. However, nucleosome recruitment into NDF-FACT condensates occurs extremely rapidly (within 10 seconds), making conventional super-resolution time-lapse techniques technically challenging within this timeframe.

To address this limitation, we employed computational deconvolution using DeconvolutionLab2 to significantly enhance the spatial resolution of our epifluorescence images. This approach reassigns out-of-focus light based on the microscope's point spread function, providing much clearer visualization of the nucleosome incorporation process. We captured images at multiple time points and quantified Cy2-nucleosome fluorescence intensity within individual condensates over time.

The enhanced images and quantitative analysis clearly demonstrate rapid nucleosome recruitment and even distribution within NDF-FACT condensates, with substantial incorporation occurring within the first 10 seconds (**NEW Extended Data Fig. 2I-m**). We have also included a short video showing the nucleosome incorporation dynamics to provide additional temporal detail of this process.

These data provide the higher-resolution insights into nucleosome recruitment kinetics that the reviewer requested, revealing the remarkably rapid timescale at which NDF-FACT condensates can concentrate and organize their nucleosomal substrates.

This revised content has been added to the manuscript in lines 190-192.

NEW Extended Data Fig. 2I-m

Nucleosome recruitment dynamics in NDF-FACT condensates. I, Time course of Cy2-nucleosome (green) incorporation into NDF-FACT condensates visualized by deconvolution-enhanced microscopy. Scale bar = 10 μm; m, Quantification of nucleosome fluorescence intensity in condensates over time (n=4 condensates, mean ± SEM).

3) Fig. 2I. Again, exciting data. I wondered whether the transcribed RNA is located inside the condensates, attached to their surface, or exported from them. This might provide mechanistic

insight into the process.

We thank the reviewer for this excellent and mechanistically important question regarding Fig. 2i. To determine the spatial relationship between transcribed RNA and condensates, we performed two complementary experiments.

First, we used STORM (Stochastic Optical Reconstruction Microscopy) with ~20 nm resolution to directly visualize the location of 6-FAM-labeled RNA relative to condensates. We captured condensate positions using the epifluorescence channel, then used STORM to precisely map RNA molecular center positions. After reconstructing ~200 individual molecular localizations, we found that RNA molecules localize within condensates rather than at their periphery or outside (**NEW Fig. 2k-l**). However, this approach has inherent limitations: the 6-FAM-fluorophore photobleaches after approximately 200 image acquisitions, and our 2D-STORM analysis cannot exclude the possibility that RNA localizes to condensate surfaces outside the focal plane.

To address these limitations and provide functional validation, we performed RNase A digestion assays. The key experimental design was that both condensate-associated and non-condensate-associated transcripts were present in the same reaction tube under identical conditions—same RNase A concentration, buffer, temperature, and reaction time. If condensates merely affected global enzyme activity, all RNA should be equally protected or degraded. However, we found that condensate-associated transcripts were significantly more resistant to RNase A digestion, while non-condensate transcripts in the same tube remained sensitive to degradation (**NEW Fig. 2j**). This differential protection in the same reaction demonstrates that condensates create a functionally distinct microenvironment that specifically protects associated RNA.

Together, the STORM localization and differential RNase A protection data provide strong evidence that transcription occurs within NDF-FACT condensates, where RNA is protected in a specialized biochemical environment that facilitates efficient transcription through nucleosomal barriers. This revised content has been added to the manuscript in lines 199-203.

j, Experimental scheme for RNase A (ng/μl) protection assay with condensate-associated (C) and non-condensate transcripts (S) in the same reaction.

k, Overlay example image of STORM super-resolution microscopy of 6-FAM-labeled RNA positions (green/yellow) relative to condensate boundaries (red, captured by epifluorescence microscope, condensates were formed with purified NDF and FACT-mCherry) from ~200 image acquisitions. Scale bar = 0.5 micrometer.

l, Quantitative analysis of RNA localization within condensates. Y-axis shows the ratio of condensate radius to center-to-center distance (distance between condensate center and RNA center). Values > 1 indicate RNA is located inside condensates, while values < 1 would indicate RNA outside condensates (n = 26 condensates).

4) Figs. 3D and S3B. Do small and large condensates exhibit similar transcriptional competency?

We thank the reviewer for this insightful question. To address whether the size of NDF-FACT condensates influence transcriptional competency, we measured nucleosome crossing times as a function of condensate area at the point when the maximum photon count was reached in the confocal scan. As shown in NEW Extended Data Fig. 3g-h, we found a moderate negative correlation between condensate area and nucleosome crossing time, suggesting that larger condensates are associated with more efficient transcription through nucleosomes, however, this correlation did not reach statistical significance, likely due to the limited sample size (N = 9). To further explore this relationship, we classified molecules into two groups based on condensate area: large condensates ($\geq 0.25 \mu\text{m}^2$) and small condensates ($\leq 0.23 \mu\text{m}^2$), and compared their average crossing times to Pol II molecules that successfully transcribed through the nucleosome, but for which no condensates were detected (NEW Extended Data Fig. 3g-h). Although pairwise comparisons did not yield statistically significant differences, we observed a trend of increased crossing times as condensate size decreased, with the longest times corresponding to conditions where no condensates were visible.

Given that the average nucleosome crossing time for Pol II in the absence of both FACT and NDF is around 90.2 seconds, the relationship between nucleosome crossing time and condensate size suggests that even small or undetected condensates are functionally competent to assist Pol II during nucleosome transcription, significantly reducing the average crossing time (large condensates ~36.2 s; small condensates ~44.2 s; no visible condensates ~51.1 s). However, additional data points will be necessary to perform a more rigorous and quantitative analysis of this relationship. Importantly, all nucleosome crossing times associated to condensate formation and measured on the C-Trap Lumicks instrument are consistent with the distribution characterized using our high-resolution dual-trap optical tweezers system (N = 25), providing confidence that the data obtained in the C-Trap platform accurately reflect Pol II nucleosome transcription dynamics in the presence of FACT and NDF.

Finally, While our current dataset (Fig. 3d and Extended Data Fig. 3b) has limited statistical power due to sample size, our ability to directly visualize and follow biochemical activity within individual NDF-FACT condensates at the single-molecule level represents a significant step forward in understanding their mechanistic role during nucleosome transcription. This approach opens new avenues for systematically dissecting how condensate properties, such as size, composition, and dynamics, impact chromatin transcription. Taken together, our limited dataset indicate that there is a trend according to which the larger condensates have shorter associated nucleosomal crossing times. We are now actively investigating these relationships, and we hope that a much larger dataset will help to provide significant insights into the mechanism by which condensates formation promote biological activities. This revised content has been added to the manuscript in lines 232-235.

NEW Extended
Data Fig. 3g-h

Relationship between NDF-FACT condensate size and Pol II nucleosome transcriptional efficiency. (A) Scatter plot showing nucleosome crossing time as a function of condensate area. (B) Comparison of average nucleosome crossing times across three groups: large condensates (N=4), small condensates (N=5), and molecules in which no condensates were detected (N=4). Error bars represent standard deviation; p-values are from unpaired two-tailed t-tests.

5) Fig. 5A. It would be informative to investigate whether Ser2 or Ser5 phosphorylation of RPB1 colocalizes with NDF and Spt16 foci.

We thank the reviewer for this excellent suggestion. We performed immunofluorescence experiments to examine the colocalization of NDF-FACT condensates with Ser2- and Ser5-phosphorylated RNA Pol II (New Extended Data Fig. 6i-k).

We observed that both Ser2- and Ser5-phosphorylated RNA Pol II show a mixture of punctate and diffused nuclear signals, consistent with previous reports (PMID: 31391587 and PMID: 29930094). NDF-FACT condensates partially colocalize with both phosphorylated forms, supporting their association with transcriptionally active regions. This partial colocalization pattern is similar to that reported for other transcription elongation factors with phosphorylated Pol II (PMID: 31391587, PMID: 36629390), supporting the functional relevance of our observations.

The partial rather than complete colocalization is expected given the temporal nature of transcription elongation. Ser5-phosphorylated Pol II is predominantly associated with promoter-proximal pausing and early elongation, while Ser2-phosphorylated Pol II is enriched toward transcription termination sites (PMID: 31391587). Our model suggests that NDF-FACT condensates provide targeted assistance when Pol II encounters nucleosomal barriers during specific phases of elongation, rather than accompanying Pol II throughout the entire transcription process.

While these results support the association between NDF-FACT condensates and active transcription, detailed characterization of Pol II condensate dynamics would require comprehensive studies beyond the scope of the current work, which focuses specifically on NDF-FACT condensate function. We have provided this data to address the reviewer's request

while maintaining focus on our central findings. This revised content has been added to the manuscript in lines 323-325.

NEW Extended Data Fig. 6 i-k

i-k, Immunofluorescence analysis of iPSC cells showing localization of NDF-FACT condensates and ser2- (i-j) and ser5- (k) phosphorylated RNA Pol II.

6) Figs. 5B–C and 6C–D. Since 1,6-hexanediol affects chromatin organization, these results are not very informative. A discussion is needed here (PMID: 33536240).

We appreciate the reviewer's important point regarding the limitations of 1,6-hexanediol as a specific probe for phase separation. We are aware that 1,6-hexanediol can affect chromatin organization and other cellular processes beyond disrupting phase-separated condensates, as highlighted in the referenced study (PMID: 33536240).

We have addressed this concern by explicitly acknowledging the limitations of 1,6-hexanediol and interpreting these results cautiously in the revised manuscript. We state:

“While 1,6-Hex sensitivity supports condensate-like behavior, we interpret these results cautiously, given potential effects on other cellular structures.” (Lines 317-319) and

“ While 1,6-hexanediol treatment is widely used in the phase separation field, it can have nonspecific effects and may target other cellular processes. To investigate NDF-FACT condensate function with greater specificity, we utilized... (Lines 361-363)”

While acknowledging its limitations, 1,6-hexanediol sensitivity remains one of the most widely used correlative assays in the phase separation field, particularly when characterizing newly identified condensates. We include this data as supportive evidence within the broader experimental framework and have cited the important reference (PMID: 33536240). We believe our multi-pronged experimental approach, combined with appropriate interpretation of the 1,6-hexanediol data, provides a robust foundation for our conclusions about NDF-FACT condensate function.

7) Page 12. “...we hypothesized that condensate formation might be critical for efficiently recruiting both factors to chromatin in cells. “

I was a bit confused about the authors' point. Does NDF/FACT leave the condensates during

transcription elongation, or does elongation occur on the surface of the condensates?

We thank the reviewer for pointing out this confusing statement. We agree that our original wording was unclear and will revise this entire section for better clarity. We have revised:

“Our single-molecule studies demonstrated that phase separation recruits large quantities of both proteins to elongating Pol II, synergistically enhancing transcription elongation (**Fig. 3**). To test whether this condensate-mediated recruitment mechanism operates in cells, we first performed... (Lines 337-340)”

Regarding the mechanistic questions about condensate dynamics during elongation, our new data provide the following insights:

First, our FRAP experiments and single-molecule observations (Figs. 2 and 3) suggest that NDF and FACT do not significantly leave the condensates during transcription elongation, indicating that the proteins remain associated within the condensate environment.

Second, based on our new data (**NEW Fig. 2j-l**), transcription elongation appears to occur inside the condensates rather than on their surface. This internal organization would allow the high local concentrations of NDF and FACT within condensates to directly facilitate nucleosome disassembly and RNA polymerase II progression through chromatin barriers.

8) Fig. S8E–I. These are interesting data, but drawing conclusions from the Curaxin experiment is difficult. Curaxin is an intercalator, so it can alter chromatin structure—as seen in the DAPI image in Fig. S8H (condensed?)—which may indirectly affect the behavior of various chromatin-associated proteins.

We thank the reviewer for this important critique and fully agree that our original interpretation of the curaxin data was problematic. As the reviewer correctly notes, curaxin is a DNA intercalator with broad chromatin effects, making it impossible to attribute changes in NDF localization specifically to FACT disruption.

We have made the following changes in response:

- 1, We have removed the misleading section title "FACT is required for NDF recruitment and chromatin occupancy in cells" which overstated our conclusions.
- 2, We have significantly revised the text to acknowledge curaxin's limitations and avoid overinterpretation. The revised section now states: "While curaxin has broader chromatin effects as a DNA intercalator, the coordinated disruption of both FACT and NDF puncta support their interdependent relationship. (Lines 402-404)"
- 3, We have reframed the experimental rationale to clearly explain why we attempted this approach (inadequate Spt16 depletion with degron) while acknowledging its limitations (Lines 395-396).
- 4, We have toned down all conclusions from this experiment to avoid suggesting causality that cannot be established with this tool (Lines 400-404).

We recognize that the curaxin experiment alone cannot provide definitive mechanistic insight due to its non-specificity. However, we believe the coordinated response of both NDF and FACT puncta to curaxin treatment provides some support for their functional relationship, even if the precise mechanism cannot be determined from this experiment alone.

Response to the comments of the reviewers (NCB-A57425A)

The core mechanistic conclusions of our study rest on the more specific approaches, particularly the NDF_K161A mutation experiments that selectively disrupt condensate formation without affecting protein-protein interactions. We appreciate the reviewer's guidance in helping us present the curaxin data with appropriate caveats.

Minor points:

1) For general readers, it might be helpful to briefly mention what NDF is in the Introduction.

We have added a section about NDF in the introduction. Lines 100-106

2) The model scheme in Fig. S10 is important and would be better placed as a main figure.

We have moved Fig. S10 to the main figures as Fig. 8.

Reviewer #3 (Remarks to the Author):

Li et al presents a tour-de-force investigation of how Pol II transcription is facilitated by FACT-NDF condensates. I recommend publication, but I believe the manuscript would be strengthened: (1) if a cartoon of the proposed FACT-NDF-facilitated transcription were included in the introduction – something like a simplified version of Fig. S10B;

Thank you for this suggestion. We have added a simplified conceptual overview (Lines 123-127) in the introduction, and moved the mechanistic model (former Fig. S10) to the main figures as new **Fig. 8**.

and (2) if the manuscript addressed the size of the FACT-NDF condensates involved in in vivo transcription – what limitations on the size of the FACT-NDF condensate droplets observed in vivo are established by imaging of the coincident NDF and FACT punta? It would be useful to specify this information in the manuscript.

We addressed the size measurements of FACT-NDF condensates in vivo and acknowledge important technical limitations in our imaging approach. Our confocal microscopy with Airyscan detected FACT-NDF condensates with diameters of $0.37 \pm 0.14 \mu\text{m}$ ($n=126$) in NDF-GFP expressing cells. However, several factors limit the accuracy of these measurements:

1, Technical Limitations:

1.1 Resolution constraints: Our Airyscan confocal system has a resolution limit of $\sim 120 \text{ nm}$, which likely overestimates the true size of smaller condensates, particularly those approaching the diffraction limit.

1.2 Exposure-dependent artifacts: Dim condensates may appear artificially enlarged during extended imaging exposures, contributing to size overestimation.

1.3 Diffraction-limited detection: A significant population of cellular condensates likely exists below the diffraction limit and remains undetected by conventional fluorescence microscopy.

Response to the comments of the reviewers (NCB-A57425A)

Recent super-resolution microscopy studies reveal that many cellular condensates are substantially smaller than diffraction-limited techniques can accurately measure. For example,

FUS condensates measured by SR-SMLM contain ~100 nm surface nanodomains with some structures as small as tens of nanometers (PMID: 37782826); Chromatin nanodomains measured by STORM are ~50-100 nm in tissue samples (PMID: 35245126). And polycomb protein condensates are ~100 nm in *Drosophila* as measured by SRM (PMID: 36979066).

These findings suggest that our measured 0.37 μm diameter likely represents an upper size distribution of FACT-NDF condensates, while smaller condensates remain unresolved. The heterogeneous size distribution we observed, combined with the known limitations of diffraction-limited microscopy, indicates that the true size range of cellular FACT-NDF condensates may extend significantly below our detection threshold.

In addition, (3) I wonder whether a recent bioRxiv, namely <https://www.biorxiv.org/content/10.1101/2024.08.26.609758v1>, provides a valuable theoretical perspective on the experimental results reported in this manuscript?

We thank the reviewer for suggesting the recent preprint by Bagheri et al. (2024), which presents a theoretical framework for prewetting—a surface-mediated phase transition that enables condensate formation at concentrations below the bulk phase separation threshold. This concept is highly relevant to our findings. In our study, we observed that NDF–FACT condensates do not form at concentrations below 60 nM in solution. However, when we include the Pol II elongation complex, condensates readily form even at these lower concentrations (Extended Data Fig. 2C). This suggests that elongating Pol II serves as a scaffold that promotes local condensation, analogous to the way membrane surfaces enable prewetting.

We have now incorporated this theoretical framework into our manuscript (Lines 164-166) to help readers understand the conceptual link between surface-coupled phase transitions and transcription-coupled condensate formation in chromatin.